# Ligand-specific activation trajectories dictate GPCR signalling in cells

Romy Thomas[1,2,10], Pauline S. Jacoby[1], Chiara De Faveri[3,11], Cécile Derieux[4], Aenne-Dorothea Liebing[5], Barbora Melkes[4], Hans-Joachim Martini[4], Marcel Bermúdez[6], Claudia Stäubert[5], Martin J. Lohse[1,2,7,8], Irene Coin[3 ✉] & Andreas Bock[1,2,4,9 ✉]

G-protein-coupled receptors (GPCRs) are key mediators of cell communication and represent the most important class of drug targets[1,2]. Biophysical studies with purified GPCRs in vitro have suggested that they exist in an equilibrium of distinct inactive and active states, which is modulated by ligands in an efficacy-dependent manner[3–11]. However, how efficacy is encoded and whether multiple receptor states occur in living cells remain unclear. Here we use genetic code expansion[12] and bioorthogonal labelling[13–16] to generate a panel of fluorescence-based biosensors for a prototypical GPCR, the $M_2$ muscarinic acetylcholine receptor ($M_2R$). These biosensors enable real-time monitoring of agonist-promoted conformational changes across the receptor's extracellular surface in intact cells. We demonstrate that different agonists produce equilibria of at least four distinct active states of the G-protein-bound $M_2R$, each with a different ability to activate G proteins. The formation of these $M_2R$–G-protein complexes occurs over 0.2–5 s along trajectories that involve both common and ligand-specific conformational changes and appear to determine G-protein selectivity. These observations reveal the molecular nature of ligand efficacy in intact cells. Selectively exploiting such different GPCR activation trajectories and conformational equilibria may open new avenues for GPCR drug discovery.

Activation of cell surface receptors by extracellular ligands is the hallmark of cell–cell communication and controls most physiological functions in humans. GPCRs are the largest class of such receptors[1,2]. After ligand activation, GPCRs communicate their message into cells by recruiting and activating intracellular G proteins[17]. GPCR ligands can activate (agonist) or inactivate (antagonist/inverse agonist) receptors to various extents, and this diversity in ligand efficacy is exploited in drug therapy[18]. However, the molecular nature of ligand efficacy and the mechanisms of GPCR activation in living cells remain largely unclear.

Fluorescence spectroscopy studies with purified $β_2$-adrenergic receptors[19–22], and more recent nuclear magnetic resonance (NMR) and double electron–electron resonance (DEER) spectroscopy studies, have established that GPCRs do not operate as simple on/off switches but exist in a dynamic equilibrium of multiple inactive and active states[3,4,23–25]. Partial agonists are believed to stabilize conformational states that are structurally different from those stabilized by full agonists[6,11,26,27]. Likewise, single-molecule Förster resonance energy transfer (FRET) and advanced NMR spectroscopy studies have demonstrated that GPCRs can form distinct signalling complexes with G proteins[28,29], in which the G proteins may possess different nucleotide affinities[5,7]. Partial agonists have been proposed to stabilize GPCR–G-protein complexes with reduced efficacy

towards nucleotide exchange[5,7,30,31]. However, all of these studies were performed in isolated systems using purified receptors reconstituted in detergent micelles or nanodiscs. Whether GPCRs in the physiological environment of a living cell also adopt different conformations and form distinct signalling complexes is essentially unclear.

Here we develop a new type of conformational GPCR biosensors to probe the existence and ligand modulation of such GPCR–G-protein-signalling complexes in intact cells. Using the $M_2R$ as a model, we demonstrate that agonist activation leads to formation of an equilibrium of distinct GPCR signalling complexes along ligand-specific activation trajectories. These distinct activation trajectories and the relative abundances of these distinct receptor–G-protein complexes dictate the type and extent to which specific G proteins are activated.

## Development of GPCR biosensors

GPCR activation results in large-scale receptor conformational changes[1,32–35]. Most prominently, outward movement of the intracellular part of transmembrane domain 6 enables coupling to and activation of G proteins. In turn, G-protein binding promotes conformational changes in the receptor, including its extracellular domains and the

[1]Rudolf Boehm Institute of Pharmacology and Toxicology, Medical Faculty, Leipzig University, Leipzig, Germany. [2]Max-Delbrück-Center for Molecular Medicine in the Helmholtz Association (MDC), Berlin, Germany. [3]Institute of Biochemistry, Leipzig University, Leipzig, Germany. [4]Institute of Pharmacology, University Medical Center of the Johannes Gutenberg-University Mainz, Mainz, Germany. [5]Rudolf-Schönheimer Institute of Biochemistry, Medical Faculty, Leipzig University, Leipzig, Germany. [6]Institute for Pharmaceutical and Medical Chemistry, Faculty of Chemistry and Pharmacy, University of Münster, Münster, Germany. [7]ISAR Bioscience Institute, Planegg, Germany. [8]Institute of Regenerative Medicine in Cardiology—Technical University of Munich, Munich, Germany. [9]Research Center for Immunotherapy (FZI), University Medical Center of the Johannes Gutenberg-University Mainz, Mainz, Germany. [10]Present address: Department of Molecular and Cellular Physiology, Stanford University, Stanford, CA, USA. [11]Present address: Department of Molecular Environmental Biotechnology, Helmholtz Centre for Environmental Research–UFZ, Leipzig, Germany. ✉e-mail: irene.coin@uni-leipzig.de; andreas.bock@uni-mainz.de

ligand-binding pocket, which stabilize agonist binding[1,32,36]. This communication between extra- and intracellular receptor domains represents the principle of allosteric coupling.

To track conformational changes at the extracellular surface of GPCRs at a high spatial resolution, we sought to develop a new type of biosensor. These biosensors should be genetically encoded, equipped with minimally sized labels (similar to those used in NMR, DEER and single-molecule FRET studies on isolated receptors) and retain an unmodified intracellular surface to preserve G-protein coupling. We chose the $M_2R$ as a model because structural studies have shown that its extracellular conformational changes on activation, including closure of the binding pocket, are the most pronounced among all class A GPCRs of which the structures have been solved[36–39].

The least invasive way to attach probes to a GPCR at the single-residue resolution in living cells is by bioorthogonal chemistry on genetically encoded chemical anchors[12–15]. In brief, a non-canonical amino acid (ncAA), also known as an unnatural amino acid, carrying an anchor for rapid catalyst-free labelling is incorporated into the receptor using genetic code expansion technology (GCE). The label is then attached post-translationally by ultrarapid strain-promoted inverse electron-demand Diels–Alder cycloaddition, which occurs within minutes without interfering with native functional groups. Using this strategy, we previously demonstrated quantitative labelling of GPCRs on the live cell surface[40].

We screened the entire extracellular surface of the $M_2R$ to identify positions at which the click-ncAA trans-cyclooct-2-ene lysine (TCO*K)[41] was efficiently incorporated and yielded robust labelling with a cell-impermeable, tetrazine-conjugated cyanine dye (Tet–Cy3) (Fig. 1a). Of 72 receptor mutants, 25 displayed good cell surface expression and labelling (Extended Data Fig. 1). The other 47 constructs were not expressed, not trafficked to the cell surface or showed no labelling at all (Fig. 1b and Supplementary Fig. 1). As a control, labelling in the absence of TCO*K produced no fluorescence (Supplementary Fig. 2).

Next, we applied the endogenous $M_2R$ agonist acetylcholine (ACh) to single cells expressing one labelled construct each using a superfusion device (Fig. 1a). Seven of the 25 Cy3-TCO*K-$M_2R$ constructs exhibited robust and reproducible changes in their fluorescence emission intensities (Fig. 1c,d), therefore serving as reporters for $M_2R$ activation. We quantified the labelling efficiency at these $M_2R$ biosensors using fluorescence correlation spectroscopy[40,42]. Four $M_2R$ biosensors showed quantitative Cy3 labelling (positions Thr84 ($M_2R^{84}$), Glu175, Ala414 and Pro415) and three reached labelling efficiencies of 60–80% (positions Phe181, Phe188 and Asn419; Extended Data Fig. 1). Importantly, all seven $M_2R$ biosensors robustly activated G proteins and internalized after ACh exposure, indicating full functionality (Extended Data Fig. 2). Notably, labelling position Ala414 with either the rhodamine dye TAMRA or the cyanine dye Cy5 also resulted in $M_2R$ activation biosensors (Supplementary Fig. 3).

Cyanine fluorophores such as Cy3 are environmentally sensitive[23,42,43]. We propose that the observed fluorescence changes arise from local microenvironmental changes (such as transitions to either more hydrophobic or to more polar microenvironments) caused by ligand-promoted receptor conformational changes. Mapping the position of ACh-sensitive labels onto high-resolution structures of the inactive and active $M_2R$ (Fig. 1e) shows that all positions featuring ACh-promoted fluorescence changes lie in extracellular receptor domains that move during receptor activation[37–39]. All seven biosensors retain the ability to respond to the positive allosteric modulator LY2119620 (ref. 38), which binds at the extracellular allosteric binding site (Extended Data Fig. 3). Thus, we infer that the $M_2R$ biosensors report on activation-related conformational changes of the receptor. Supporting this, binding of the antagonist N-methylscopolamine did not induce fluorescence changes at any of these positions (Extended Data Fig. 2).

ACh-stimulated fluorescence changes varied in both direction (fluorescence increase or decrease) and amplitude ($\Delta F/F_0$, −16% to +10%)

(Fig. 1c,d and Supplementary Table 1). All changes were strictly dependent on the presence of ACh and returned to the baseline after ACh washout (Fig. 1c). Moreover, ACh-mediated changes in fluorescence were also concentration dependent, with potencies matching reported ACh affinity values (around 1–6 μM; Extended Data Fig. 4), which further supports the specificity of the observed effects.

## Ligand-unique conformational states of the $M_2R$

The $M_2R$ biosensor panel enables monitoring receptor conformational changes across the entire extracellular surface (Fig. 1e) while leaving the intracellular domains, which couple to G proteins, untagged. This makes it an ideal tool for investigating the existence of distinct, ligand-specific GPCR active states and their coupling to G proteins in intact cells. To study agonist effects, we selected $M_2R$ agonists differing in their reported efficacies in classical pharmacological assays: the endogenous full agonist ACh, the superagonist iperoxo[39,44–46], and the partial agonists arecoline and pilocarpine. All of the agonists stimulated G-protein activation at all seven biosensors in a concentration-dependent manner, similar to wild-type (WT) $M_2$ receptors (Extended Data Fig. 5). We noted that higher agonist potency tended to correlate with a smaller dynamic range of the functional response (Extended Data Fig. 4i). This would be compatible with the notion that some biosensors might display slightly different spontaneous activity compared with WT $M_2$ receptors. Only at the $M_2R^{181}$ biosensor were all of the agonists less potent and less efficacious. Nonetheless, the rank order of agonist potencies and efficacies at all biosensors matched WT receptors (Extended Data Fig. 5 and Supplementary Table 2). This demonstrates that conformational data obtained with the panel of biosensors can be projected to WT receptors.

Stimulation of the $M_2R$ biosensor family with a saturating concentration of iperoxo changed fluorescence emission at six out of the seven ACh-responsive $M_2R$ biosensors (Fig. 2 and Extended Data Fig. 6). Compared with ACh, iperoxo produced larger responses at five biosensors ($M_2R^{84}$, $M_2R^{175}$, $M_2R^{414}$, $M_2R^{415}$ and $M_2R^{419}$), but smaller fluorescence changes at the $M_2R^{188}$ biosensor. Notably, the $M_2R^{181}$ biosensor did not respond at all (Fig. 2a,b and Extended Data Fig. 6) at any of the sequences of agonist addition (Supplementary Fig. 4). Absolute $\Delta F/F_0$ amplitudes after iperoxo exposure (Fig. 2b and Supplementary Table 1) were normalized to ACh at each $M_2R$ biosensor and plotted in a radar plot (Fig. 2c), irrespective of signal direction. We define this visualization of agonist-specific conformational changes of the receptor as the ligand's conformational fingerprint (Fig. 2c).

The conformational fingerprints of arecoline (Fig. 2f) and pilocarpine (Fig. 2i) differed markedly from those of ACh and iperoxo. Arecoline induced smaller fluorescence changes compared with ACh at all positions (Fig. 2d,e, Extended Data Fig. 6 and Supplementary Table 1). Notably, arecoline activation of $M_2R^{84}$ and $M_2R^{175}$ biosensors caused a decrease in fluorescence, in contrast to the increase observed with ACh (Fig. 2d,e and Extended Data Fig. 6). For pilocarpine, $M_2R^{181}$ and $M_2R^{188}$ showed more-pronounced changes in fluorescence compared with ACh, while all of the other biosensors reported significantly smaller changes than with ACh (Fig. 2g,h and Extended Data Fig. 6).

Comparison of the radar plot conformational fingerprints (Fig. 2c,f,i) enables general conclusions: conformational changes at most agonist-sensitive receptor positions (84, 175, 414, 415, 419) scale directly with agonist efficacies, and are largest with the superagonist iperoxo. By contrast, conformational changes at $M_2R^{181}$ and $M_2R^{188}$ biosensors are inversely correlated with agonist efficacies, and are largest with the low-efficacy partial agonist pilocarpine (Fig. 2i). At $M_2R^{181}$, the superagonist iperoxo does not induce changes in fluorescence intensity at all (Fig. 2c).

These data cannot be explained by a simple two-state model of one inactive and one active receptor conformation. Instead, they suggest the existence of an ensemble of distinct active states of the $M_2R$ in intact cells, consistent with a recent NMR study on purified $M_2R$ receptors[24].

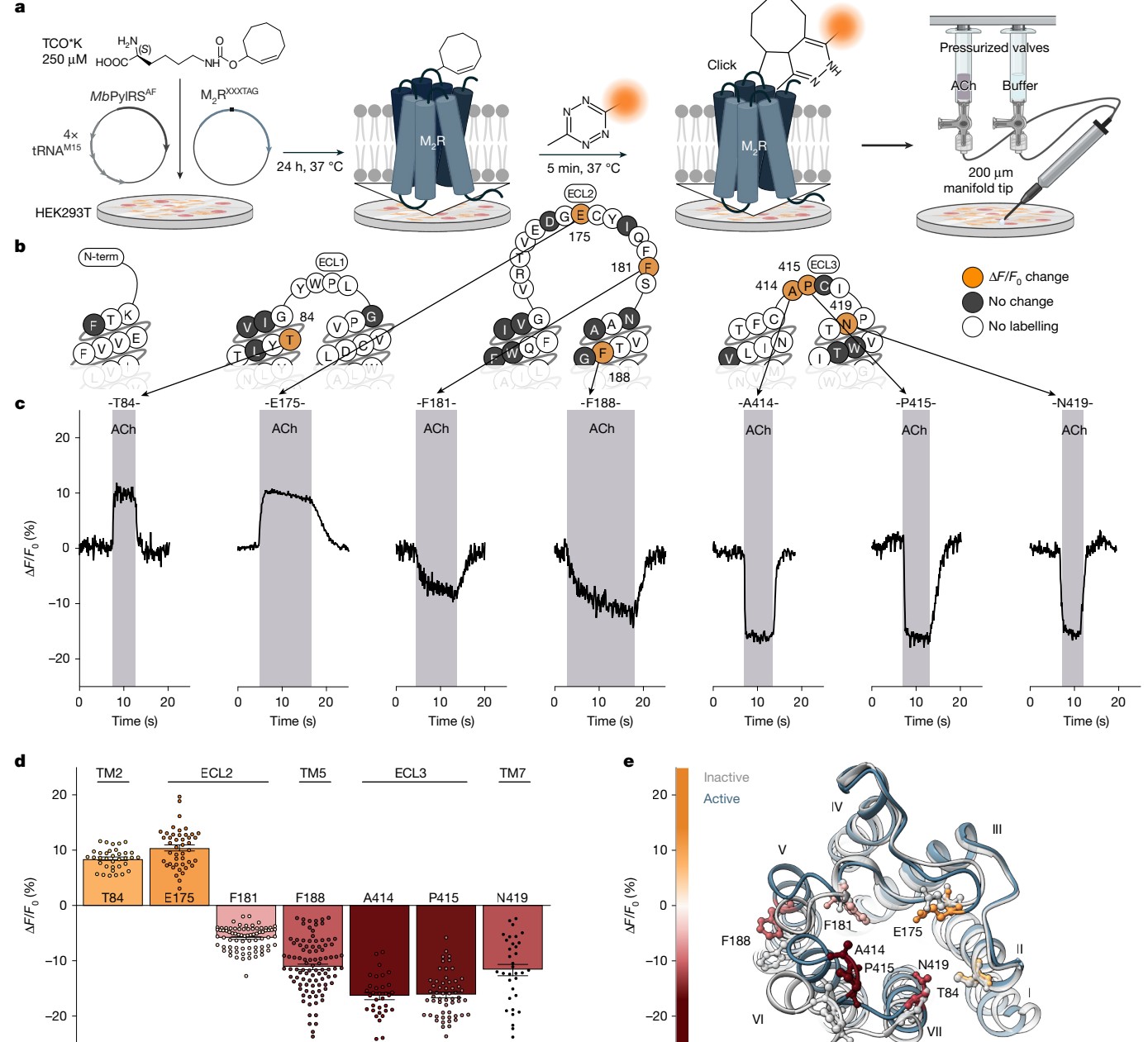

**Fig. 1 | An extracellular single-colour conformational GPCR biosensor panel. a**, Genetic incorporation of a ncAA (TCO*K) and bioorthogonal labelling of $M_2R$ with Tet–Cy3 (Methods). Cells expressing biosensors are stimulated by continuous pressurized application of agonist (ACh) or buffer through a manifold tip. Created in BioRender. Thomas, R. (2025) https://BioRender.com/loxqlqf. **b**, Snake plot of $M_2R$ indicating all positions that were robustly labelled (orange and grey) and showed activation-related changes in fluorescence intensity (orange). The numbers are the residue numbers. Positions that could not be labelled are indicated in white. ECL1, extracellular loop domain 1; ECL2, extracellular loop domain 2; ECL3, extracellular loop domain 3. **c**, Representative changes in fluorescence intensity ($\Delta F/F_0$) were recorded over time from several individual HEK293T cells expressing $M_2R$ biosensors labelled at the indicated amino acid positions. Cells were superfused with 1 mM ACh. The shaded areas indicate the duration of agonist addition, and the unshaded areas indicate

agonist washout with buffer. **d**, Mean fluorescence intensity changes ($\Delta F/F_0$) of all seven agonist-sensitive biosensors after activation with 1 mM ACh. Positive values indicate an increase in fluorescence; negative values indicate a decrease after ACh superfusion. Data are mean ± s.e.m., with each datapoint representing a single cell. $M_2R^{84}$ (35 cells examined over 17 independent experiments), $M_2R^{175}$ (44, 11), $M_2R^{181}$ (76, 9), $M_2R^{188}$ (96, 15), $M_2R^{414}$ (33, 7), $M_2R^{415}$ (56, 9), $M_2R^{419}$ (34, 12). TM2, transmembrane domain 2; TM5, transmembrane domain 5; TM7, transmembrane domain 7. **e**, Top view of the X-ray crystal structure of the active $M_2R$ (blue; Protein Data Bank (PDB): 4MQS). The positions of incorporated TCO*K yielding GPCR biosensors are colour coded according to the gradient, representing the mean ACh-induced changes in fluorescence intensity ($\Delta F/F_0$). For comparison, the X-ray crystal structure of the inactive $M_2R$ (PDB: 3UON) is shown in grey. The roman numerals indicate the number of the transmembrane helix. The constructs used were SP-$M_2R^{XXXTAG}$ (Methods).

## Distinct receptor–G-protein complexes

The observation that the extent of conformational changes elicited by agonist activation scales directly with agonist efficacy at some receptor positions but inversely at others suggests that the $M_2R$ adopts multiple

active states in living cells, which differ in their ability to stimulate G-protein signalling. To probe the existence of such distinct $M_2R$–G-protein complexes, we manipulated the equilibrium between different states by increasing G-protein coupling. To shift the receptor equilibrium toward a high-efficacy, high-affinity $M_2R$–G-protein complex,

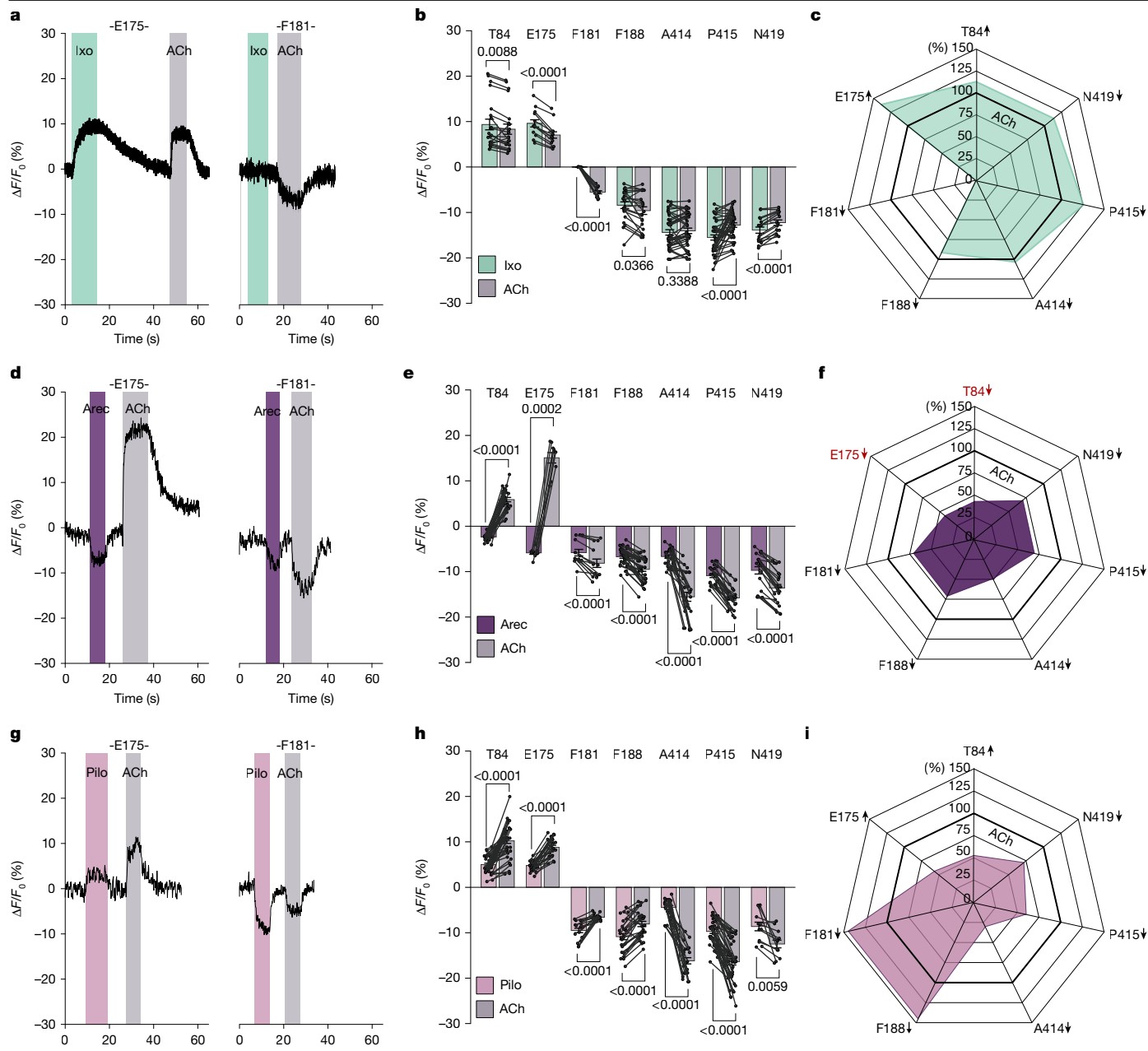

**Fig. 2 | The M₂R biosensor panel uncovers agonist-specific conformational fingerprints in intact cells. a,d,g,** Representative changes in fluorescence intensity ($\Delta F/F_0$) recorded in real-time from single HEK293T cells expressing the indicated M₂R biosensors superfused with 100 μM iperoxo (Ixo; **a**), 1 mM arecoline (Arec; **d**) and 10 mM pilocarpine (Pilo; **g**), followed by 1 mM ACh after wash out with buffer, respectively. Application of different agonists is indicated by shaded areas in different colours. Non-shaded areas indicate buffer application. **b,e,h,** The mean changes in fluorescence intensity ($\Delta F/F_0$) of all seven biosensors after activation with the indicated ligand (iperoxo (green; **b**), arecoline (purple; **e**) and pilocarpine (pink; **h**)) after wash out with buffer. Positive values indicate an increase in fluorescence; negative values indicate a decrease after agonist superfusion. Connected datapoints represent data retrieved from the same cell. Data are mean ± s.e.m., with each datapoint representing a single cell (Supplementary Table 1). M₂R⁸⁴ (iperoxo (21 cells

examined over 7 independent experiments), arecoline (25, 6), pilocarpine (39, 7), M₂R¹⁷⁵ (iperoxo (13, 7), arecoline (9, 4), pilocarpine (25, 5)), M₂R¹⁸¹ (iperoxo (12, 5), arecoline (13, 7), pilocarpine (17, 7)), M₂R¹⁸⁸ (iperoxo (21, 5), arecoline (35, 6), pilocarpine (27, 6)), M₂R⁴¹⁴ (iperoxo (38, 12), arecoline (26, 11), pilocarpine (22, 3)), M₂R⁴¹⁵ (iperoxo (40, 9), arecoline (26, 5), pilocarpine (43, 10)) and M₂R⁴¹⁹ (iperoxo (19, 9), arecoline (22, 6), pilocarpine (12, 6)). For **b**, **e** and **h**, P values were calculated using two-tailed paired *t*-tests. **c,f,i,** The mean changes in fluorescence intensity ($\Delta F/F_0$) in response to iperoxo (green; **c**), arecoline (purple; **f**) and pilocarpine (pink; **i**), normalized to $\Delta F/F_0$ of 1 mM ACh in the same cell. The bold line indicates ACh (set to 100%). The direction of $\Delta F/F_0$ is indicated by arrows: increase (up), decrease (down). For **f**, the change of direction in fluorescence emission for M₂R⁸⁴ and M₂R¹⁷⁵ after arecoline stimulation is highlighted in red. The constructs used were SP-M₂Rˣˣˣᵀᴬᴳ (Methods).

we overexpressed the Gα_oA(G203T) mutant. This Gα mutant features low affinity towards both GDP and GTP[47,48], and its coupling to agonist-bound receptors yields stable nucleotide-free receptor–Gα_oA-protein complexes[48]. Indeed, comparison of maximal fluorescence changes

(relative to ACh) between the individual agonists after Gα_oA(G203T) overexpression reveals differences to the efficacy rank order of agonists in the presence of the endogenous G-protein repertoire of the cell (Supplementary Fig. 6). This is direct evidence that the equilibrium of

different receptor states is altered by the presence of the $G\alpha_{oA}$(G203T) mutant.

We compared relative fluorescence changes across all seven $M_2R$ biosensors in presence of the endogenous G-protein repertoire of HEK293 cells and after $G\alpha_{oA}$(G203T) overexpression (Fig. 3). All of the $M_2R$ biosensors except for the biosensor $M_2R^{419}$ (Supplementary Fig. 5) were sensitive to $G\alpha_{oA}$(G203T) overexpression, but the responses strongly depended on both the ligand and the tested biosensor (Fig. 3). According to the principle of allosteric coupling, the conformational changes of the $M_2R$ induced by coupling to $G\alpha_{oA}$(G203T) are expected to mirror those initiated by agonist binding from the extracellular side. With G-protein overexpression, these changes occur before agonist addition. As a result, agonist stimulation of biosensors indicating the formation of the high-efficacy $M_2R$–G-protein complex would result in reduced or no changes in fluorescence emission intensities.

Most prominently, and for all agonists tested, $G\alpha_{oA}$(G203T) overexpression resulted in a complete loss of agonist-induced changes in fluorescence at the $M_2R^{175}$ biosensor (Fig. 3a,c). At many class A GPCRs, formation of the high-affinity complex between a nucleotide-free G protein and the receptor entails conformational rearrangements of extracellular receptor domains[1,8,32,36]. In the extreme case, this can result in a complete closure of the ligand-binding pocket, which limits agonist association and dissociation, therefore increasing the lifetime of the high-affinity complex[36–39]. Although not providing direct atomic-level structural insights, our data strongly suggest that the biosensor $M_2R^{175}$ indicates conformational changes of the extracellular loops involved in the formation of the high-affinity receptor–G-protein complex. In support of this conclusion, Ala mutation of Tyr426, a key residue involved in this lid closure, led to a complete loss of ACh-induced fluorescence changes in the $M_2R^{175}$ biosensor (Extended Data Fig. 7). Furthermore, this is corroborated by the fact that, under endogenous G-protein levels, the fluorescence changes at the $M_2R^{175}$ biosensor increase with the efficacy of the agonist (Fig. 2). Similar to the loss of signal at the $M_2R^{175}$ biosensor, the changes in fluorescence at the $M_2R^{415}$ (Fig. 3b) were reduced by around 30–80% in the presence of overexpressed $G\alpha_{oA}$(G203T) (Fig. 3b,c).

By contrast, for most ligands, $G\alpha_{oA}$(G203T) overexpression resulted in a significant increase in fluorescence emission at $M_2R^{181}$ (Fig. 3d) and $M_2R^{188}$ (Fig. 3e,f) compared with the endogenous G-protein condition. This suggests that the conformational changes induced by receptor coupling to $G\alpha_{oA}$(G203T) at these positions are different from those induced by the agonists alone. This effect was most pronounced for the high-efficacy agonists than for the partial agonists (Fig. 3f). Notably, although iperoxo did not induce any response at the $M_2R^{181}$ biosensor at endogenous G-protein levels (Fig. 2a–c), it did so in the presence of $G\alpha_{oA}$(G203T) (Fig. 3d–f). Thus, the increase in agonist-promoted changes in fluorescence at $M_2R^{181}$ (Fig. 3d) and $M_2R^{188}$ (Fig. 3e) in the presence of overexpressed $G\alpha_{oA}$(G203T) likely unmasks the stabilization of a $M_2R$–G-protein complex that is distinct from the one revealed by the loss or decrease of responses at the $M_2R^{175}$ and $M_2R^{415}$ biosensors.

To determine whether this distinct complex may have low signalling efficacy, we increased the abundance of GDP-bound $M_2R$–G-protein complexes by pretreating the cells with pertussis toxin (PTX) overnight. Through ADP-ribosylation of $G\alpha_{i/o}$ subunits, PTX locks the $\alpha$ subunits of endogenous $G_{i/o}$ proteins into an inactive GDP-bound state[49], which hampers productive coupling between receptors and G proteins. In contrast to the effects of $G\alpha_{oA}$(G203T) overexpression, pretreatment with PTX completely abolished the fluorescence intensity changes at the $M_2R^{181}$ biosensor induced by ACh and arecoline and significantly decreased the pilocarpine-promoted effect (Extended Data Fig. 8). On the basis of these findings, we propose that these biosensors indicate a low-efficacy, likely GDP bound, $M_2R$–G-protein signalling complex. This is consistent with biophysical experiments using purified proteins in vitro, which suggest the existence of such low-efficacy, GDP-bound receptor complexes at agonist-bound $\beta_2ARs^{[5,11,50]}$, $A_{2A}$ receptors[6,7,31] and $\mu ORs^{[51]}$.

The effects of $G\alpha_{oA}$(G203T) overexpression on two other biosensors, $M_2R^{84}$ and $M_2R^{414}$, were more heterogeneous and varied strongly with the efficacy of the ligand. Specifically, at $M_2R^{84}$, $G\alpha_{oA}$(G203T) overexpression reduced ligand-promoted fluorescence changes compared with endogenous G-protein levels for all of the tested agonists except arecoline (Fig. 3g,i). The unique behaviour of arecoline is further highlighted by the experiments at endogenous G-protein levels (Fig. 2d,e), where arecoline stimulation resulted in fluorescence intensity changes in the opposite direction compared with all of the other tested ligands.

The $M_2R^{414}$ biosensor responded to agonist exposure in a strongly ligand efficacy-dependent manner when the $G\alpha_{oA}$(G203T) mutant was overexpressed (Fig. 3h,i). Similar to the effects observed with the $M_2R^{84}$, $M_2R^{175}$ and $M_2R^{415}$ biosensors, ACh- and iperoxo-promoted fluorescence changes were diminished relative to endogenous G-protein levels, whereas those of the partial agonists arecoline and pilocarpine were increased (Fig. 3h,i).

In summary, the $G\alpha_{oA}$(G203T) overexpression data suggest the existence of at least two functionally distinct $M_2R$–G-protein complexes at steady state.

## Kinetics of the formation and dynamics of $M_2R$–G-protein complexes

To assess whether evidence for distinct $M_2R$–G-protein complexes can also be inferred from the kinetics of the various biosensors, we analysed the time course of their agonist-promoted fluorescence changes. The apparent on-rates of the fluorescence changes promoted by the high-efficacy agonists ACh and iperoxo at the six G-protein sensitive $M_2R$ biosensors lie in the range of 170–2,700 ms (Extended Data Fig. 9a) and 340–3,200 ms (Extended Data Fig. 9b), respectively (Supplementary Table 3). These on-rates are much slower than those obtained with previous intracellular GPCR biosensors (on-rates ≈ 30–50 ms) that were used to monitor activation kinetics[52]. Importantly, the on-rates for the extracellular GPCR biosensors described here agree very well with the kinetics of GPCR-mediated G-protein activation (on-rates ≈ 500–1,000 ms)[52]. Consistent with this, overexpression of $G\alpha_{oA}$(G203T) further increases the apparent on-rates for the vast majority of agonists and biosensors (Extended Data Fig. 9 and Supplementary Table 3). Thus, the conformational changes that we record at the extracellular surface of the receptor probably result from intracellular G-protein coupling, further corroborating that our panel of GPCR biosensors monitors conformational changes in GPCR–G-protein signalling complexes.

By inspecting these rates more closely, we can distinguish two groups of biosensors: one group comprising the $M_2R^{84}$, $M_2R^{414}$, $M_2R^{415}$ and $M_2R^{175}$ biosensors, which displayed the fastest on-kinetics between 160–530 ms (for ACh) and 340–1,000 ms (for iperoxo), and the other group including the $M_2R^{181}$ and $M_2R^{188}$ biosensors, with apparent on-rates of more than 1 s (for ACh) and approximately 3 s (for iperoxo at $M_2R^{188}$) (Extended Data Fig. 9a,b). Notably, iperoxo appeared to divide the group of $M_2R^{84}$, $M_2R^{414}$, $M_2R^{415}$ and $M_2R^{175}$ biosensors further into two subgroups (Extended Data Fig. 9b).

Different kinetic patterns were observed when the biosensors were activated with the partial agonists arecoline and pilocarpine. Specifically, on-rates after activation with arecoline yielded three groups of $M_2R$ biosensors: fast biosensors ($M_2R^{84}$, $M_2R^{175}$ and $M_2R^{415}$; on-rate ≈ 300 ms), an intermediate biosensor ($M_2R^{414}$; on-rate ≈ 900 ms) and slow biosensors ($M_2R^{181}$ and $M_2R^{188}$; on-rate ≈ 2 s) (Extended Data Fig. 9c and Supplementary Table 3). Finally, kinetic analysis of pilocarpine-stimulated conformational changes revealed two groups of $M_2R$ biosensors, which reported receptor conformational changes on significantly different timescales. As seen with ACh and iperoxo, pilocarpine triggered fast conformational changes at the $M_2R^{175}$, $M_2R^{414}$ and $M_2R^{415}$ biosensors with apparent on-rates between 180 and 330 ms (Extended Data Fig. 9d). However, in contrast to what was observed with all other agonists, pilocarpine-stimulated conformational changes reported by $M_2R^{84}$

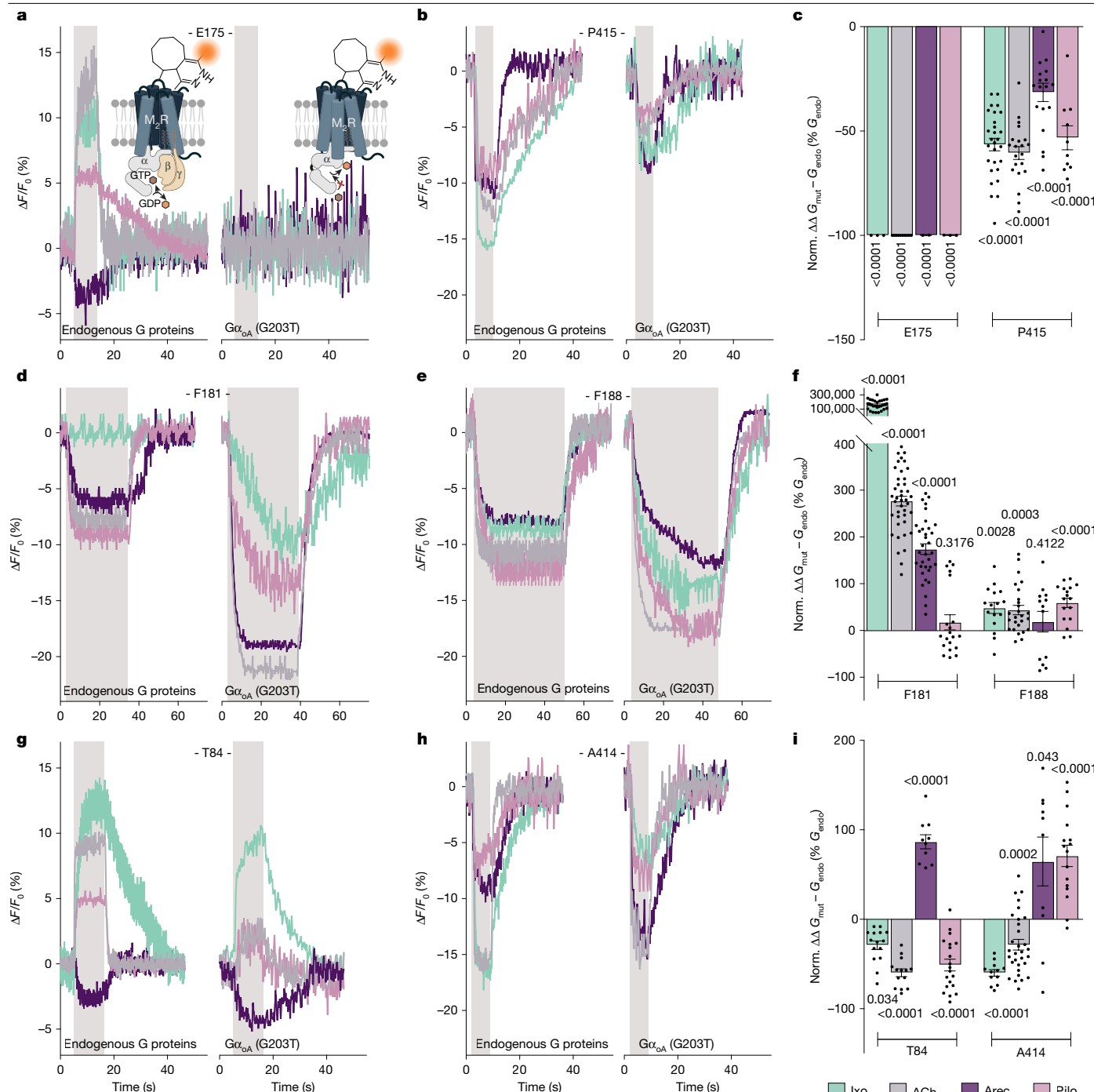

**Fig. 3 | M₂ receptor activation results in an equilibrium of distinct M₂R–G-protein signalling complexes in intact cells.** Comparison of ligand-promoted changes in fluorescence intensity ($\Delta F/F_0$) of the M₂R biosensors labelled with Tet–Cy3 at the indicated positions and stimulated with 1 mM ACh (grey), 100 µM iperoxo (green), 1 mM arecoline (purple) or 10 mM pilocarpine (pink) at endogenous G-protein levels or after Gα_oA(G203T) overexpression. **a,b,d,e,g,h**, Representative traces of the $\Delta F/F_0$ of single cells expressing the indicated M₂R biosensors (M₂R[175] (**a**), M₂R[415] (**b**), M₂R[181] (**d**), M₂R[188] (**e**), M₂R[84] (**g**) and M₂R[414] (**h**)) at endogenous G-protein levels (left) or after Gα_oA (G203T) overexpression (right). The shaded areas indicate the duration of agonist superfusion, and the unshaded areas represent buffer application. **c,f,i**, Statistical summary of the normalized (norm.) changes in $\Delta F/F_0$ obtained from experiments in **a** and **b** (**c**), **d** and **e** (**f**), and **g** and **h** (**i**). Shown are the ligand-dependent differences in $\Delta F/F_0$ after Gα_oA(G203T) overexpression

($G_{mut}$) normalized to the mean $\Delta F/F_0$ of endogenous G-protein levels ($G_{endo}$, set to 0%). Negative values indicate decreases in $\Delta F/F_0$ after Gα_oA(G203T) overexpression, and positive values indicate increases in $\Delta F/F_0$ after Gα_oA(G203T) overexpression. Data are mean ± s.e.m., with each datapoint representing a single cell. M₂R[84] (iperoxo (15 cells examined over 5 independent experiments), ACh (14, 7), arecoline (10, 3), pilocarpine (20, 4)), M₂R[175] (iperoxo (11, 3), ACh (10, 3), arecoline (9, 3), pilocarpine (11, 3)), M₂R[181] (iperoxo (29, 4), ACh (41, 7), arecoline (34, 4), pilocarpine (20, 3)), M₂R[188] (iperoxo (16, 3), ACh (26, 4), arecoline (13, 3), pilocarpine (17, 3)), M₂R[414] (iperoxo (12, 4), ACh (31, 9), arecoline (10, 3), pilocarpine (16, 3)), M₂R[415] (iperoxo (28, 5), ACh (21, 6), arecoline (19, 5), pilocarpine (10, 3)). For **c**, **f** and **i**, P values were calculated using unpaired two-tailed t-tests. The constructs used were SP-M₂R^XXXTAG (Methods). The diagram in **a** was created in BioRender. Thomas, R. (2025) https://BioRender.com/loxqlqf.

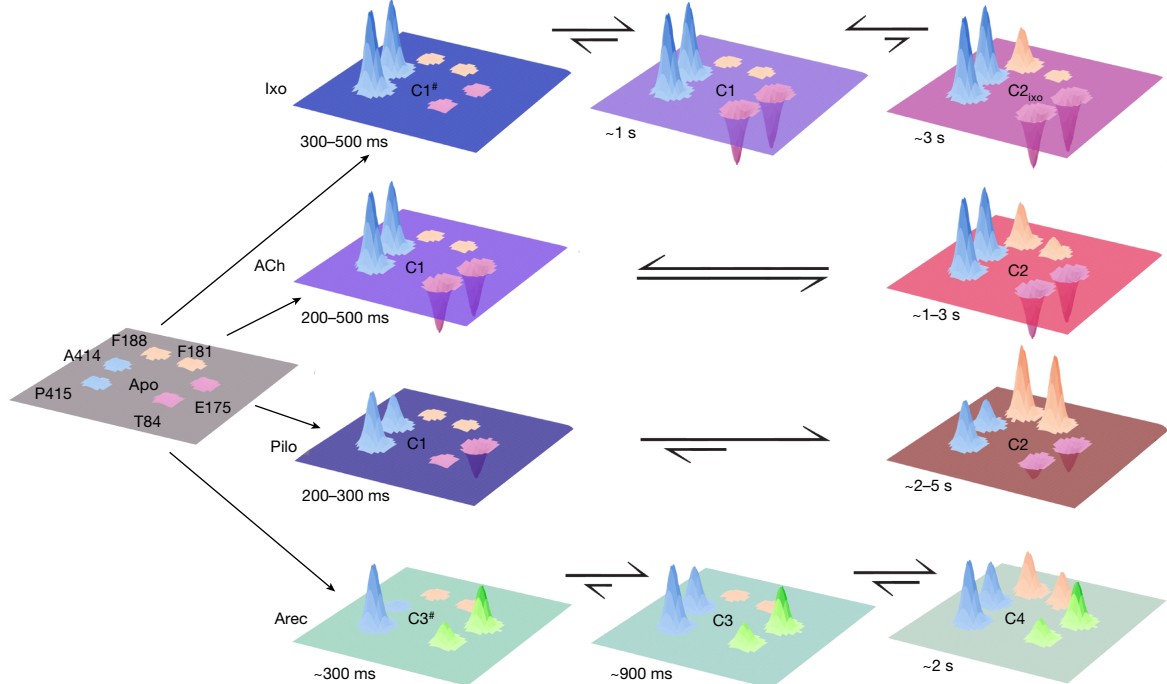

**Fig. 4 | Ligand-specific activation trajectories and equilibria of M₂R–G-protein signalling complexes.** Schematic three-dimensional representation of the time-resolved, agonist-mediated formation of GPCR signalling complexes in intact cells. The planes illustrate the receptor's extracellular surface, and the numbered positions indicate the six biosensors that were sensitive to G-protein modulation (Fig. 3). Agonist-promoted changes in fluorescence are shown as peaks. The mean fluorescence changes (Fig. 2) define the height of each peak. For better visualization, the direction of fluorescence changes was inverted (that is, fluorescence decreases are shown as increasing peaks). Peaks from the same group of biosensors are shown in the same colour. The colour of each

plane is the sum of the colours of all peaks in that plane. This representation yields unique colours that enable visual discrimination between ligand-specific receptor–G protein signalling complexes. For simplification, the apo state is depicted in grey and does not result from the sum of peak colours. The positions of the conformational equilibria at steady state are indicated by the equilibrium arrows. Indicated times are apparent on-rates of agonist-specific fluorescence changes (Extended Data Fig. 9 and Supplementary Table 3). The superscript hash symbols indicate intermediate complexes. The C in the planes stands for complex.

occurred on the same slow timescale as those reported by M₂R[181] and M₂R[188] (Extended Data Fig. 9d).

Overall, the kinetic analysis of receptor conformational changes (Extended Data Fig. 9) yielded the same groups that we identified by modulating the activity state of the G protein (Fig. 3): the fast responders include M₂R[175] and M₂R[415] sensors, which report on the formation of the high-efficacy complex, whereas M₂R[181] and M₂R[188], which indicate the formation of the low-efficacy complex, responded slower. This strongly corroborates the notion that the activated M₂R can form a minimum of two G-protein signalling complexes in intact cells that differ in their signalling efficacies and suggests that these complexes form at different timepoints. Moreover, the patterns of on-rates suggest that these complexes form along different activation trajectories depending on the type of agonist. In the case of ACh, iperoxo and pilocarpine, activation of the receptor results first in the formation of a high-efficacy GPCR signalling complex (complex 1, C1), primarily indicated by increases in fluorescence at position 175 (and fluorescence decreases at positions 414 and 415) (Figs. 2–4). This process takes about 200 ms to 1 s, depending on the agonist (Fig. 4 and Extended Data Fig. 9). A second, low-efficacy, GPCR signalling complex (C2) forms then after around 2–5 s and is indicated by decreases in fluorescence at positions 181 and 188 (Fig. 4). Modulation of the activity state of the G protein (Fig. 3) has clearly demonstrated that C1 and C2 co-exist in an equilibrium; however, it remains unclear whether C2 develops from C1 or whether C1 and C2 form in parallel.

In addition to the two common receptor signalling complexes C1 and C2, there are important ligand-specific differences: compared with ACh, iperoxo stabilizes a greater fraction of C1 (Figs. 2 and 4).

Moreover, iperoxo-activated receptors appear to reach C1 through an intermediate state that forms earlier (300–500 ms) and is indicated by fluorescence changes exclusively at positions 414 and 415 (C1#). Furthermore, the iperoxo-stabilized complex 2 (C2ᵢₓₒ) forms to a much smaller extent compared with ACh; and C2ᵢₓₒ is highly distinct as its stabilization does not involve any conformational rearrangements at position 181. Pilocarpine stabilizes C1 to a much smaller extent than ACh and its formation does not involve fast conformational changes at position 84 (Fig. 4). Furthermore, pilocarpine stabilizes C2 most effectively, which is indicated by the largest conformational changes at positions 181 and 188 among all of the tested agonists (Figs. 2 and 4).

By contrast, arecoline-mediated receptor activation follows a unique trajectory that is highly distinct from the other three agonists (Fig. 4). Arecoline activation stabilizes first the formation of a high efficacy signalling complex (C3) that is primarily characterized by decreases, rather than increases, in fluorescence at positions 84 and 175 (Figs. 2 and 4). This complex forms in about 900 ms through an intermediate state C3# (Fig. 4). Formation of the low-efficacy signalling complex (C4) takes approximately 2 s and involves fluorescence decreases at positions 181 and 188 that are significantly larger than the ones resulting from ACh activation but smaller than those induced by pilocarpine (Fig. 4).

## Equilibria of complexes define ligand efficacy

Our data (Figs. 2–4) have collectively demonstrated that all agonists stabilize the formation of at least two distinct M₂R–G-protein signalling complexes in intact cells at steady state. Common to all agonists is the observation that these signalling complexes co-exist in an equilibrium

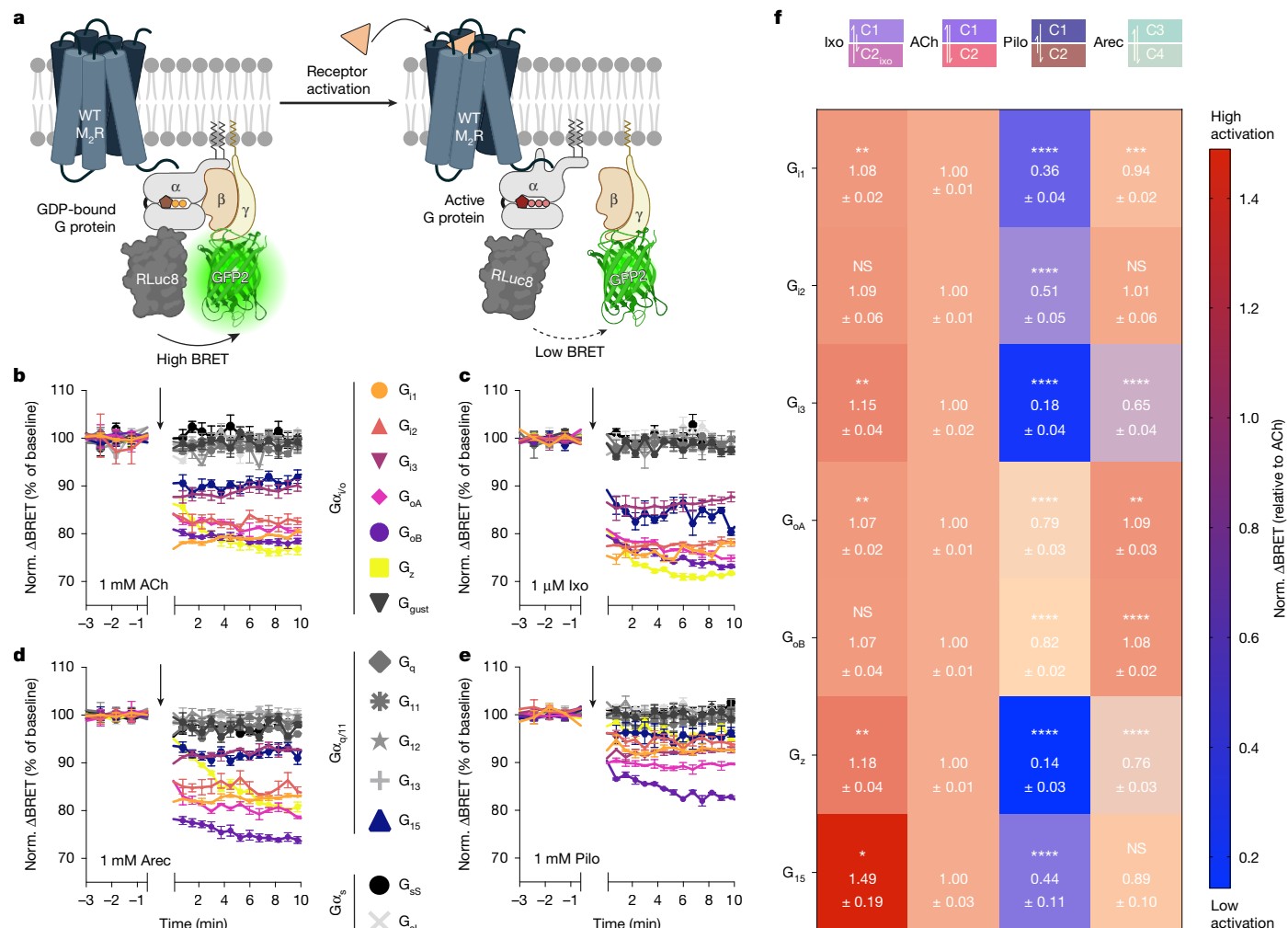

**Fig. 5 | Activation trajectories and conformational equilibria define ligand efficacy in living cells. a**, TRUPATH assay principle. The BRET-based $G\alpha\beta\gamma$ biosensors sense the conformational rearrangement during receptor-promoted G-protein activation as a decrease in BRET due to the increased distance between $G\alpha$ and $G\gamma$ subunits[53]. Created in BioRender. Thomas, R. (2025) https://BioRender.com/loxqlqf. **b–e**, Representative traces of agonist-promoted changes in normalized $\Delta$BRET (%) for all G-protein biosensors after activation of WT $M_2R$ (SP-$M_2R$-WT) with the indicated agonists. Agonists were applied (black arrows) at saturating concentrations of ACh (**b**), iperoxo (**c**), arecoline (**d**) and pilocarpine (**e**). Number of independent experiments for ACh, iperoxo, arecoline and pilocarpine, respectively: 4, 6, 4 and 6 ($G_{i1}$); 4, 7, 4 and 6 ($G_{i2}$);

4, 6, 4 and 5 ($G_{i3}$); 6, 5, 3 and 6 ($G_{oA}$); 5, 8, 3 and 5 ($G_{oB}$); 5, 6, 4 and 4 ($G_z$); and 6, 6, 4 and 5 ($G_{15}$). **f**, G protein-coupling selectivity profile of the $M_2R$. Overview of agonist-promoted $\Delta$BRET normalized to ACh (set to 1) obtained after agonist-promoted, $M_2R$-mediated stimulation of BRET-based G-protein biosensors (TRUPATH). The coloured boxes (using the colour code from the planes in Fig. 4) on top of each column represent the equilibrium of receptor–G-protein complexes stabilized by the indicated ligand (Fig. 4). Heat-map values represent the mean ± s.e.m. (Supplementary Table 4). Statistical analysis was performed using one-way analysis of variance (ANOVA) with Dunnett's post-hoc test for multiple comparisons (ACh as reference); ****$P < 0.0001$, ***$P < 0.001$, **$P < 0.01$, *$P < 0.05$; NS, not significant.

and feature different efficacies towards G-protein signalling (Fig. 3). However, depending on the specific agonist, the formation of the GPCR signalling complexes occurs at different timescales and follows distinct trajectories (Fig. 4 and Extended Data Fig. 9) that involve ligand-unique conformational changes of the receptor (Fig. 2). Moreover, the position of the signalling-complex equilibrium is highly ligand dependent (Fig. 4).

To assess whether this conformational complexity dictates which specific set of G proteins is activated by each ligand, we profiled all agonists in BRET-based G-protein-activation assays using the TRUPATH biosensor platform[53], which comprises 14 different $G\alpha$ subunits (Fig. 5a). All agonists showed strong selectivity for activating the $G_{i/o}$ family of G proteins (Fig. 5b–e) and displayed similar potencies across the $G_{i/o}$-family members ($G_{i1}$, $G_{i2}$, $G_{i3}$, $G_{oA}$, $G_{oB}$, $G_z$) (iperoxo, ~0.2–2 nM; ACh, ~30–300 nM; arecoline, ~0.2–2 μM; pilocarpine, ~2–10 μM; Extended Data Fig. 10 and Supplementary Table 3). Moreover, some small but significant activation of $G\alpha_{15}$ was observed (Fig. 5b–e).

Importantly, comparing the maximal responses of agonists across the seven G proteins clearly reveals a rich texture of ligand efficacies (Fig. 5f), whereby the partial agonists appear to display more efficacy differences between the individual G-protein subunits than the full agonists. In particular, iperoxo behaved as a superagonist for almost all G proteins and pilocarpine elicited a partial response in all G protein-activation assays. However, arecoline displayed a unique efficacy profile exerting superagonism for $G_{oA}$ and $G_{oB}$ while behaving as a partial agonist for all of the other members of the $G_{i/o}$-protein family (Fig. 5f).

Linking a ligand's conformational equilibrium (Fig. 4) to its G-protein signalling profile (Fig. 5) suggests that the position of this equilibrium (C1 versus C2) dictates the strength of agonism across the G-protein subtypes while the ligand-specific trajectory of complex formation has an important role in discriminating between different G-protein subtypes. Specifically, iperoxo stabilizes the high-efficacy complex C1 to a greater extent than ACh (Figs. 2 and 4). However, and in contrast

to ACh, it hardly forms any low-efficacy complex C2 (Figs. 2 and 4). Conversely, pilocarpine strongly favours stabilization of C2 over C1 (Figs. 2 and 4). Finally, the unique trajectory of arecoline-mediated complex formation (Fig. 4) results in preferential activation of $G_o$ over $G_i$ proteins (Fig. 5).

## Conclusions

Overall, our study reveals that the activation of a GPCR in intact cells may be far more complex than previous biophysical studies with isolated receptors have suggested. By attaching minimal-sized fluorescent labels to a set of activation- and G-protein coupling-sensitive positions on the extracellular receptor surface, we were able to track a receptor's activation trajectory in real-time with considerable conformational detail directly within the native membrane environment of an intact cell. In contrast to earlier fluorescent GPCR activation biosensors that rely on fusion to large fluorescent proteins, self-labelling tags or, at best, oligomeric epitopes[52], our sensors report on the movement of single receptor positions labelled with a fluorophore anchored through an ncAA, while leaving the intracellular surface of the receptor completely untouched. Although the method does not reach the atomic-level resolution of biophysical techniques such as NMR or DEER—as fluorophores are attached to the receptor backbone through a relatively large and flexible linker—it offers to our knowledge the highest spatial and temporal resolution currently achievable for tracking conformational changes in receptors within live cells.

Taken together, our data demonstrate that agonist activation of a GPCR results in the formation of an equilibrium of distinct active GPCR states and signalling complexes with various G proteins that differ in their efficacies. We propose that distinct agonists form such signalling complexes along different activation trajectories, involving both common and ligand-specific conformational changes in the receptor that evolve over time in a ligand-dependent manner. These individual activation trajectories may form the molecular basis for G-protein-subtype selectivity, while the position of this signalling complex equilibrium at steady state may define the overall strength of agonism. Thus, our study elucidates the molecular nature of ligand efficacy in intact cells.

It will be interesting to investigate whether and how these GPCR activation trajectories and the resultant formation of signalling complex equilibria can be exploited to expand our ability to manipulate receptors and achieve specific downstream responses and, ultimately, superior therapeutic effects. Moreover, we anticipate that our single-colour conformational sensor technology will be broadly applicable to other receptors, enabling the temporal dissection of conformational changes elicited by structurally distinct ligands, ranging from small molecules to larger peptides with diverse pharmacological profiles. This unique information, derived from the live-cell context, will enhance development of GPCR drug candidates with unique signalling profiles and expand our understanding of the general principles underlying GPCR structural dynamics.

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

# Methods

## Biosensor construction

**Gibson assembly.** All cloning, except for site-directed mutagenesis, was conducted using Gibson assembly (NEBuilder HiFi DNA Assembly Master Mix, New England Biolabs)[54]. All primers were synthesized by BioTeZ. A list of all of the primer sequences is provided in Supplementary Table 5. Primers 1 and 3 were used to cut within the pcDNA3.0 backbone to decrease fragment sizes and increase yields for PCR amplification. All PCR products were obtained using Q5 High-Fidelity DNA Polymerase (New England Biolabs). All constructs were verified by Sanger sequencing (LGC Genomics).

For SP-$M_2$R-WT, the cDNA of human WT $M_2$R was cloned into pcDNA3.0 and a cleavable signal peptide (SP)[55] was cloned N-terminally of WT human $M_2$R using primers 1 and 2, 3 and 4, and 5 and 6. To obtain SP-$M_2$R-WT-eGFP, eGFP was fused to the C terminus of SP-$M_2$R-WT using primers 1 and 7, 3 and 8, and 9 and 10. For the ELISA assay, an HA-tag was cloned N-terminally to the WT receptor, resulting in SP-HA-$M_2$R-WT, using primers 3 and 11, and 1 and 12.

**Site-directed mutagenesis.** Receptor mutants were cloned by introducing an amber stop codon (TAG) at the desired positions by site-directed mutagenesis using the AAscan primer design tool[56]. A list of the primer sequences is provided in Supplementary Table 6. Mutagenesis was performed using SP-$M_2$R-WT, SP-HA-$M_2$R-WT and SP-$M_2$R-WT-eGFP as templates. Point mutations were introduced by PCR using Thermo Fisher Scientific Phusion High-Fidelity DNA Polymerase (New England Biolabs). PCR products for QuickChange mutagenesis were incubated for 1 h at 37 °C with 1 µl DpnI restriction enzyme (New England Biolabs) before transformation. The resulting mutants are referred to as SP-$M_2$R$^{XXXTAG}$, SP-HA-SP-$M_2$R$^{XXXTAG}$ or SP-$M_2$R$^{XXXTAG}$-eGFP throughout. To introduce the mutation disrupting the tyrosine lid closure (Y426A), the primers 13 and 14, and SP-$M_2$R$^{175TAG}$ as a template were used, resulting in SP-$M_2$R$^{175TAG}$-Y426A. The point mutation was introduced using the Q5 Site-Directed Mutagenesis Kit (New England Biolabs).

## Cell culture

HEK-tsA201 (Sigma-Aldrich; referred to as HEK293T cells throughout) cells were cultured in T75 flasks at 37 °C, 5% $CO_2$ in complete DMEM with 4.5 g l$^{-1}$ glucose (PAN-Biotech). Culture media was supplemented with 10% (v/v) FBS (Biochrom), 100 U ml$^{-1}$ penicillin, 100 mg ml$^{-1}$ streptomycin (Biochrom) and 2 mM L-glutamine (PAN-Biotech). Cells were passaged every 2–3 days when reaching a confluency of 80–90%. For passaging and seeding, the culture medium was aspirated, cells were washed with 5 ml Dulbecco's PBS solution (Sigma-Aldrich), detached with 2 ml trypsin/EDTA (PAN-Biotech), resuspended in 5 ml DMEM and transferred to a new T75 flask. All cell lines were routinely tested for mycoplasma contamination using MycoAlert Mycoplasma Detection Kit (Lonza Group) and were not contaminated with mycoplasma. For the qualitative screen of labelled mutants at the accessible extracellular receptor surface, HEK293T cells were seeded on glass-bottomed 8-well µ-slides (Ibidi) at a density of approximately $7 \times 10^4$ cells per well in 300 µl DMEM. For single-cell fluorescence microscopy experiments, HEK293T cells were seeded on 24 mm glass coverslips (Paul Marienfeld) in 6-well plates at a density of approximately $2.5–3 \times 10^5$ cells per well in 1.5 ml culture medium. Coverslips and 8-well µ-slides were coated with poly-D-lysine (PDL; Sigma-Aldrich; 25 µg ml$^{-1}$ in PBS) for 30 min at room temperature and washed with PBS twice before seeding cells. For the quantification of labelling efficiency by temporal brightness experiments, HEK-293AD (BioCat; referred to as HEK-AD cells throughout) cells were seeded on uncoated 24 mm glass coverslips in 6-well plates at a density of approximately $3 \times 10^5$ cells per well in 1.5 ml culture medium. To determine the cell-surface expression of $M_2$R biosensors using a cell-surface enzyme-linked immunosorbent assay (ELISA), $1.6 \times 10^6$ HEK293T cells were seeded into a T25 flask and grown

for 24 h at 37 °C. For the TRUPATH G-protein activation experiments of WT $M_2$ receptors, HEK293T cells were seeded into 6-well plates at a density of $3 \times 10^5$ cells per well.

For the TRUPATH Gα$_{oA}$-activation assay of WT $M_2$R and each of the seven $M_2$R biosensors as well as SP-$M_2$R$^{175TAG}$-Y426A, HEK293T cells were seeded into T75 flasks and grown for 24 h at 37 °C to a confluency of 80–85%.

For the internalization experiments, HEK293T cells were seeded onto glass-bottomed 4-well µ-slides (Ibidi) at a density of approximately $5 \times 10^4$ cells per well in 300 µl DMEM.

## Transfection and ncAA incorporation

**Principle and rationale of ncAA incorporation.** To genetically encode a ncAA, an orthogonal aminoacyl-tRNA synthetase (AARS)–tRNA pair must be introduced into the host cell. This pair does not crosstalk with the endogenous synthetase–tRNA pairs responsible for incorporating canonical amino acids. Typically, the orthogonal pair is derived from a different organism—for example, bacterial pairs are commonly used for GCE in eukaryotic cells. The ncAARS specifically recognizes the ncAA, while the orthogonal tRNA functions as an amber suppressor: it carries an anticodon complementary to a stop codon (usually the amber stop codon UAG) and competes with termination factors to reassign this natural nonsense codon as a sense codon. The ncAARS charges the orthogonal tRNA with the ncAA, which is then delivered to the ribosome for regular incorporation into the nascent protein. Each ncAA requires a specific AARS, although some AARSs can accommodate more than one amino acid.

In our laboratory, we have established a two-plasmid system to incorporate ncAA into proteins of interest (POIs). One plasmid (typically pcDNA3) carries the gene encoding the POI, in which a TAG stop codon replaces the natural codon at the position targeted for ncAA incorporation. The second plasmid is a bicistronic construct that encodes the translational machinery: the aminoacyl-tRNA synthetase (AARS) and the corresponding tRNA. To ensure proper expression and processing of the procaryotic tRNA in mammalian cells, the tRNA gene—lacking the 3′-CCA sequence—is placed under the control of external Pol III promoters (H1 or U6) and followed by an appropriate trailer. To achieve the high tRNA concentrations required to outcompete the release factor, the tRNA expression cassette is typically repeated in tandem.

The orthogonal pair used for TCO*K incorporation is derived from the system that naturally incorporates pyrrolysine (Pyl) in methanogenic archaea in response to the amber codon (UAG)[57]. Specifically, the plasmid contains one copy of the *Methanococcus barkeri* pyrrolysyl-tRNA synthetase (*Mb*PylRS) under the control of a CMV promoter, along with four tandem copies of the gene encoding the enhanced M15 tRNA for expression in mammalian cells[58]. The plasmid, which was generated in our lab, is deposited in Addgene, where the complete map and additional information can be found (105830, https://www.addgene.org/105830/).

**Protocols. Live-cell epifluorescence microscopy.** Cells grown on coverslips in 6-well plates were transfected 12–24 h after seeding when reaching a confluency of 40–60%. Before transfection, the culture medium was changed to DMEM without supplemented antibiotics, FBS and L-glutamine. For bioorthogonal labelling of *trans*-cyclooct-4-*en*-lysine (TCO*K, SiChem) a premix of HEPES buffer (1 M, pH 7.4, Sigma-Aldrich) and TCO*K stock solution (100 mM TCO*K in ncAA storage buffer, 0.2 M NaOH, 15% DMSO) was added to a final concentration of 0.25 mM TCO*K per well 1 h before transfection. Cells on coverslips were transfected using Lipofectamine 2000 transfection reagent (Thermo Fisher Scientific) as follows: per well a total amount of 1.5 µg cDNA was diluted with 150 µl Opti-MEM (Thermo Fisher Scientific) and combined after 5 min incubation at room temperature with 3.75 µl Lipofectamine 2000 transfection reagent, diluted in 150 µl Opti-MEM. After 20 min incubation at room temperature, the transfection mixture was added dropwise to each well. To maintain cell viability and to remove

remaining excess TCO*K, the medium was changed to complete DMEM 4–6 h after transfection. Cells were grown an additional 18–20 h before single-cell microscopy experiments were conducted.

For ncAA incorporation and subsequent bioorthogonal labelling, the cDNA of SP-M$_2$R$^{XXXTAG}$ or SP-M$_2$R$^{175TAG}$-Y426A (for biosensor activation experiments) or SP-M$_2$R$^{XXXTAG}$-eGFP (for expression analysis and quantification of bioorthogonal labelling) and the MbPylRS$^{AF}$/4xtRNA$^{M15}$ were transfected at a 1:1 ratio. For the labelled control without ncAA incorporation, cells were transfected with SP-M$_2$R$^{414TAG}$/MbPylRS$^{AF}$/4xtRNA$^{M15}$ as described in the previous section (total amount of cDNA, 1.5 µg) but then not loaded with TCO*K. When overexpressing the Gα$_{oA}$(G203T) mutant, a cDNA ratio (total amount of cDNA, 1.5 µg) of 1:1:1 (SP-M$_2$R$^{XXXTAG}$:Gα$_{oA}$(G203T):MbPylRS$^{AF}$/4xtRNA$^{M15}$) was used. For the Gα$_{i3}$-FRET activation assay, the cDNA (total amount of cDNA, 1.5 µg) of the SP-M$_2$R$^{XXXTAG}$ or SP-M$_2$R-WT was transiently transfected with MbPylRS$^{AF}$/4xtRNA$^{M15}$ and the G$_{i3}$-FRET biosensor at a ratio of 10:10:1. For the internalization experiments, the cDNA (total amount of cDNA, 1.5 µg) of SP-M$_2$R$^{XXXTAG}$ or SP-M$_2$R-WT-eGFP, MbPylRS$^{AF}$/4xtRNA$^{M15}$ and GRK3 were transfected at a 5:5:1 ratio.

**Cell-surface ELISA.** For determining the cell-surface expression of M$_2$R biosensors using ELISA, the cells were supplemented with TCO*K as described above and transfected with 10 µl Lipofectamine 2000, using a cDNA ratio of 1:1 (SP-HA-M$_2$R$^{XXXTAG}$ or SP-HA-M$_2$R-WT, MbPylRS$^{AF}$/4xtRNA$^{M15}$) of 4 µg total plasmid 24 h after seeding in a T25 cell culture flask.

**BRET-based G-protein activation experiments (TRUPATH).** To assess the G protein-activation profile of M$_2$ WT receptors, cells were transfected with SP-M$_2$R-WT, Gα-RLuc8, Gβ and GFP$^2$-Gγ at a ratio of 1:1:1:1 in a total amount of 1.5 µg cDNA per well of a 6-well plate. For each Gα subunit, the combination of Gβ and Gγ used was the one optimized previously[53], as listed in Supplementary Table 7. Transfection was performed using the Effectene Transfection Kit (Qiagen) 18–24 h after seeding. For the transfection per well, the cDNA was premixed with 66 µl DNA-condensation buffer (EC-buffer) and 12 µl enhancer was added, mixed and incubated at room temperature for 2 min. After adding 6 µl Effectene, the mix was incubated at room temperature for 20 min. The culture medium was renewed, 350 µl DMEM was added to the transfection mix and the resulting solution applied dropwise to the cells.

For TRUPATH Gα$_{oA}$-activation experiments of M$_2$R WT and each of the seven M$_2$R biosensors as well as the double mutant SP-M$_2$R$^{175TAG}$-Y426A, per T75 flask cells were transfected with the cDNA of either SP-M$_2$R-WT (1.5 µg) or SP-M$_2$R$^{XXXTAG}$ (3.5 µg) together with MbPylRS$^{AF}$/4xtRNA$^{M15}$ (3.5 µg), Gα$_{oA}$-RLuc8 (1.5 µg), Gβ$_8$ (1.5 µg) and GFP$^2$-Gγ$_3$ (1.5 µg). In the case of SP-M$_2$R-WT the amount of cDNA was lowered to reduce expected expression differences compared with the M$_2$R biosensors. To ensure equal levels of transfected cDNA among all samples, 2 µg of empty pcDNA3.1 vector was added to the samples containing SP-M$_2$R-WT. Transfections were performed using Lipofectamine 2000 transfection reagent according to manufacturer recommendations for 16–24 h. In brief, cDNAs were dissolved in 1,200 µl OptiMEM (per flask), incubated for 5 min and 1,200 µl of OptiMEM containing 30 µl of Lipofectamine was added per transfection mixture and incubated with the cDNAs for 20 min at room temperature. The resulting solution was then added into freshly exchanged DMEM containing 100 mM TCO*K, according to the same principle as described above. To avoid transfection variability in between the different M$_2$Rs, stock solutions containing MbPylRS$^{AF}$/4xtRNA$^{M15}$, Gα$_{oA}$-RLuc8, Gβ$_8$ and GFP$^2$-Gγ$_3$ with appropriate cDNAs quantities, as well as Lipofectamine were prepared and divided subsequently for each construct.

### Bioorthogonal labelling of M$_2$R biosensors

Fluorescent labels were attached to the receptor using ultra-rapid click chemistry between dye–tetrazine derivatives and the ncAA TCO*K. While we have experience with other labelling chemistries[13]

in our laboratory, such as copper-catalysed azide–alkyne cycloaddition (CuAAC) on both azide- and alkyne-containing ncAAs, and strain-promoted azide–alkyne cycloaddition (SPAAC) on ncAAs bearing strained alkynes (for example, BCNK)[14], we have consistently achieved the best results in terms of labelling speed, efficiency, cell viability and reproducibility using ultrarapid strain-promoted inverse electron-demand Diels–Alder cycloaddition on TCO*K. This ncAA carries a selected isomer of cyclooctene (*trans*-2-cyclooctene), which is highly reactive with both tetrazine and methyl-tetrazine dye derivatives yet sufficiently stable over the duration of the experiment (maximum of 2 min)[41].

Cells expressing SP-M$_2$R$^{XXXTAG}$ or SP-M$_2$R$^{XXXTAG}$-eGFP, as well as SP-M$_2$R$^{175TAG}$-Y426A, were labelled 30 min to 1 h before microscopy. If not indicated otherwise, cells were labelled with Tet–Cy3-conjugated dye (JenaBioscience) according to our previously published protocol[59]. In brief, Tet-conjugated dyes were dissolved in imaging buffer (144 mM NaCl, 5.4 mM KCl, 1 mM MgCl$_2$, 2 mM CaCl$_2$, 10 mM HEPES, pH 7.3), from 0.5 mM stock solutions in DMSO to a final concentration of 1.5 µM. The culture medium was removed from the cells and, subsequently, 0.5 ml (coverslips) or 150 µl (µ-slides) of the solution was applied to the cells and removed after 5 min incubation at 37 °C. Cells were kept in imaging buffer at 37 °C until imaging. For the G$_{i3}$-FRET activation assay cells were labelled using Tet-Cy5-conjugated dye (Lumiprobe) according to the same protocol. Cells were labelled with Cy5 (instead of Cy3) to overcome the spectral overlap of Cy3 with the acceptor fluorophore of the G$_{i3}$-FRET biosensor (that is, cpVenus). To test the transferability of the biosensor approach using structurally different dyes, cells were labelled with tetrazine-5-TAMRA (JenaBioscience) using the same protocol.

### Labelling screen of M$_2$R mutants

For these experiments, the SP-M$_2$R$^{XXXTAG}$-eGFP constructs were used. For the qualitative evaluation of full-length receptor expression and membrane localization, all cloned constructs of SP-M$_2$R$^{XXXTAG}$-eGFP were expressed in HEK293T cells and labelled at the respective site as described before. Confocal images of the cells expressing SP-M$_2$R$^{XXXTAG}$ were taken before and after labelling with Tet–Cy3 to ensure cell viability before labelling. The screen for labelled receptor mutants was done using a LEICA TCS SP8 laser-scanning microscope with an oil-immersion objective (HC PL APO ×63/1.40 NA, oil). A 554 nm laser was used at 5% power to excite Cy3 fluorophores and the respective emission was measured within 590–650 nm. To excite eGFP fluorophores, a 488 nm laser was used at 5% power and the respective emission was measured within 500–555 nm. Images were acquired with a hybrid detector in sequential scan mode to avoid bleedthrough (1,024 × 1,024 pixel, line average 4, 400 Hz, gating 0.3–6 ms) using the Leica Application Suite X (LASX) software (v.3.5.7.23225). Labelling was assessed by considering full-length receptor expression (reflected by C-terminal eGFP) and Cy3-staining of the same cells. M$_2$R mutants were evaluated as being labelled when Cy3-labelling and eGFP membrane staining could be observed robustly for cells from at least three independent experiments.

### Temporal brightness analyses through quantification of labelling efficiencies

For these experiments, the SP-M$_2$R$^{XXXTAG}$-eGFP constructs were used. The labelling efficiency of the selected M$_2$R biosensors was quantified using molecular brightness analyses according to a previously published protocol[59]. In brief, the seven different SP-M$_2$R$^{XXXTAG}$-eGFP constructs were transfected in HEK-AD cells and bioorthogonally labelled with Tet–Cy3 as described in the 'Bioorthogonal labelling of M$_2$R biosensors' section above. Temporal brightness experiments were conducted on the LEICA TCS SP8 laser-scanning microscope, using the same laser lines and detector settings as described in the 'Labelling screen of M$_2$R mutants' section above. To reduce possible photobleaching, the laser power was reduced to 1% while identifying

appropriate cells. Cells were imaged at their basolateral membranes. Suitable cells for the analysis exhibited a homogenous morphology of the basolateral membrane and distribution of fluorescent spots. For each cell, 100 consecutive frames were acquired (256 × 256 pixel, line average 1, zoom factor 22.8).

Temporal brightness analysis was performed in ImageJ (v.1.5.4f) using the Number & Brightness analysis plugin of J. Unruh[60]. Image stack files were converted to 16-bit and appropriate regions with homogenous intensity distribution were selected from single fluorescence channels, opened as interactive 2D histogram. The number of emitters per pixel was extracted from the intensity of that pixel divided by the molecular brightness of the emitters as follows:

$N = x_{avg}/y_{avg} - 1$, where $y_{avg}$ represents the average apparent brightness of all selected pixels over time and $x_{avg}$ represents the average intensity of all selected pixels over time. The data were plotted as a scatter dot plot of $N$ obtained from the eGFP channel and $N$ obtained from the Cy3 channel. To determine the labelling efficiency, data were fitted to a linear regression with constraints at $x,y = 0$. The slope of the regression indicates the resulting labelling efficiency.

### Live-cell epifluorescence microscopy
**Single-cell kinetic experiments with $M_2R$ biosensors.** For these experiments, the SP-$M_2R^{XXXTAG}$ constructs, as well as SP-$M_2R^{175TAG}$-Y426A, were used. Kinetic single-cell fluorescence microscopy experiments using the $M_2R$ biosensors were conducted using an inverted DMi8 epifluorescence microscope (Leica Microsystems), equipped with an oil-immersion objective (HC PL APO ×63/1.40–0.60, oil), a high-speed polychromator (VisiChrome, Visitron Systems), a Xenon-Lamp (75 W, 5.7 A, Hamamatsu Photonics) or a CoolLED pE-800 (40% illumination, CoolLED) for the labelled control without ncAA incorporation and the kinetic experiments with the SP-$M_2R^{175TAG}$-Y426A mutant, a Photometrics Prime 95B sCMOS camera (Visitron Systems) with a Optosplit II dual emission image splitter (Cairn research), and the Visiview v.4.0 imaging software (Visitron Systems). A DAPI/FITC/Cy3/Cy5 ET Quadband Filter (ChromaTechnology) was used for imaging Cy3-labelled cells at 555/10 nm. Emission was recorded using a T590lpxr dichroic mirror (ChromaTechnology) and a 595/50 nm emission filter (ChromaTechnology). Coverslips with transfected cells were transferred to imaging chambers (Attofluor, Thermo Fisher Scientific) and washed once with imaging buffer. Cells were kept in imaging buffer throughout the experiment. All single-cell imaging was performed at room temperature. For ligand application, a solenoid valve perfusion system with a 200-µm inner diameter manifold-tip (Octaflow II, ALA Scientific Instruments) was used. Ligands were applied in direct vicinity of the cells after superfusion with imaging buffer for 5–10 s. The superfusion was conducted at a pressure of 50 mbar. Image sequences were recorded at 100-ms excitation time and acquisition intervals or 50 ms when using the CoolLED pE-800 lightsource. Image processing was performed with ImageJ[61]. Each cell was analysed individually. Cell membranes were selected as regions of interest using the drawing tool. An area without cells was defined as background. Fluorescence intensity over time of all regions was extracted for each emission channel. The raw data were processed by subtracting the background fluorescence at every timepoint for all recorded emission channels.

Changes in the fluorescence emission intensity were normalized to baseline according to the following formula: $\Delta F/F_0 = ((F - F_0)/F_0) \times 100\%$, where $F_0$ is the mean emission intensity of the first ten datapoints of the time series, and $F$ is the mean emission intensity of ten datapoints at the stable plateau of emission intensity changes reached after ligand application. To retrieve the apparent on-rates $\tau$, a plateau followed by one-phase association/decay ordinary fitting was performed as follows: $r(t) = F_0 + (\text{Plateau}-F_0) \times (1 - e^{-K \times (t-t_0)})$, if $t < t_0$, where $t$ is time (s), $t_0$ is the respective timepoint of ligand application and Plateau-$F_0$ is the amplitude of emission intensity change. From this, $\tau$ has been

calculated as $\tau = 1/K$. Concentration–response curves were fit using a four-parametric variable slope fit (log(agonist) versus response) by calculating:

$Y = \text{Bottom} + (\text{Top} - \text{Bottom})/(1 + 10^{\log(EC_{50} - X)} \times \text{HillSlope})$, where $X$ is the log of dose or concentration, $Y$ is the response (Plateau) and Top is the maximum efficacy $E_{max}$.

**$G_{i3}$-FRET activation assay.** For the $G_{i3}$-FRET activation assay, the SP-$M_2R^{XXXTAG}$ constructs were used. The FRET assay was performed using an Olympus IX83 Inverted Microscope equipped with an oil-immersion objective (UPLAPO60XOHR ×60/1.5 NA, oil) with an ORCA-Fusion C14440-20UP camera (Hamamatsu Photonics). A Spectra III-LCR-8S-A21 light engine (Lumencore) at 50 mW light intensity and an additional 20% transmission neutral-density filter (Qioptiq Photonics) were used for excitation with the following band-pass filters: CFP, 438/29 nm; YFP, 511/16 nm; Cy5 637/12 nm. The emission light was split into two channels using an OPTOSPLIT II (Cairn research) equipped with the following band-pass filters: CFP, 475/28 nm; YFP, 542/27 nm; and Cy5, 700/75 nm. For FRET measurements only Tet–Cy5-labelled cells were imaged. Sequences of images were acquired with camera scan mode 2, a 30 ms excitation time, 100 ms frame interval and 2 × 2 camera binning, resulting in 1,152 × 1,152 pixel resolution. After each individual FRET-experiment, YFP direct excitation was recorded with the same imaging settings, enabling comparisons of sensor expression. Ligands were applied using a solenoid valve perfusion system with a 100-µm inner diameter manifold-tip (Octaflow II, ALA Scientific Instruments) at a pressure of 350 mbar. In addition to background correction for both donor and acceptor emission, the obtained acceptor emission was corrected for spectral bleedthrough ($B$) as $\text{Acceptor}_{emission} - B \times \text{Donor}_{emission}$ (ref. 62). The spectral bleedthrough was experimentally determined from FRET measurements of HEK293T cells expressing the donor fluorophore mTurquoise2 as the ratios of acceptor/donor emission. A correction factor of 0.22 was determined from three independent experiments. The FRET-ratios were calculated as the ratio of corrected acceptor emission (referred to as FRET) over corrected donor emission as $\text{Acceptor}_{emission}/\text{Donor}_{emission}$. All datapoints were plotted as $\Delta\text{FRET} (\%) = (\text{FRET} - \text{FRET}_0/\text{FRET}_0) \times 100\%$, where $\text{FRET}_0$ is the baseline, which was determined from the average of ten datapoints before ligand application.

**Inhibition of endogenous $G_{i/o}$ proteins with PTX.** For these experiments, the SP-$M_2R^{XXXTAG}$ constructs were used. To inhibit the activation of endogenous $G_{i/o}$ proteins, cells have been treated overnight with the exotoxin PTX (Tocris Bioscience). To this end, HEK293T cells were seeded and the $M_2R$ biosensors were transfected as described above. After 4–6 h of transfection, the culture medium was exchanged for complete DMEM, and PTX was added to the cells at a final concentration of 0.1 µg µl$^{-1}$. Cells were grown an additional 18–20 h before single-cell microscopy experiments were conducted. Labelling, fluorescence microscopy and image processing were carried out as described above. For suitable comparability, single-cell microscopy experiments were conducted for cells with and without PTX pretreatment on the same day for the same $M_2R$ biosensor.

### Cell-surface expression of $M_2R$ biosensors
For these experiments, the SP-HA-$M_2R^{XXXTAG}$ and SP-HA-$M_2R$-WT constructs were used. Cell-surface expression of SP-$M_2R$-WT and the $M_2R$ biosensors in HEK293T cells was assessed using an indirect cellular ELISA assay as described in a previously published protocol[63]. In brief, 24 h after transfection, cells were detached using Versene, reseeded into 48-well plates at a density of $1.2 \times 10^5$ cells per well and grown at 37 °C for another 24 h. The ELISA assay was performed by fixing the cells with 4% formaldehyde, washing twice with PBS, blocking with DMEM supplemented with 10% FBS. The medium was removed and 50 mU ml$^{-1}$ anti-HA-peroxidase antibody (1:1,000, Sigma-Aldrich) was applied to

the cells. The cells were washed three times with PBS. Subsequently, PBS was removed and detection solution (3.7 mM $o$-phenylenediamine, 22.65 mM citric acid, 51.4 mM $Na_2HPO_4$) including $H_2O_2$ were applied. The reaction was stopped by adding stopping reagent (0.12 M $Na_2SO_3$, 1 M HCl). Per well, the supernatant was transferred into a clear 96-well plate and the absorbance was measured at 492 nm (620 nm reference wavelength) in an EnVision microplate reader. The data were corrected to the reference wavelength and normalized to the absorbance of $M_2R$ WT (set to 100%).

**BRET-based G protein-activation experiments (TRUPATH)**

**G-protein-activation profile of the $M_2R$ WT.** To survey which G proteins are activated by the SP-$M_2R$-WT, we used the TRUPATH library (Addgene, 1000000163) according to protocols that were published recently[53,64]. In brief, 24 h after transfection cells were detached, collected in fresh supplemented DMEM and reseeded into PDL-coated white 96-well plates at a density of $5 \times 10^4$ cells per well. Cells were grown at 37 °C for another 24 h. The plates were stored at room temperature for 10 min before starting the assay. The culture medium was removed, and the cells in every well were washed twice with imaging buffer. Cells were kept in 60 µl imaging buffer and 10 µl of freshly prepared Prolume Purple working solution (NanoLight Technology) was added to each well at a final concentration of 5 µM per well. The plates were incubated for 5 min at room temperature in the dark. BRET measurements were performed using the Synergy Neo2 plate reader (Agilent Technologies), equipped with the BRET2 Filter cube (Agilent Technologies, 400/510 nm emission) and using a 50 ms integration time. Before ligand application of 30 µl per well, a baseline was recorded for 5 min. Negative controls were obtained by the addition of 30 µl imaging buffer per well instead of ligand. The BRET measurement was continued for 20 min with a time interval of 45 s between each datapoint. The BRET ratios 510/400 of each well were normalized to the baseline of the negative control (set to 100%). For the G-protein-activation heat map, all datapoints were normalized to the mean plateau value of the wells stimulated with 1 mM ACh. The data were analysed using Microsoft Excel 2016.

**Functional characterization of $M_2R$ biosensors.** For the TRUPATH $G\alpha_{oA}$-activation assay of WT $M_2R$ and the seven $M_2R$ biosensors, the same batch of cells expressing one of the $M_2R$ biosensors or the WT receptor was seeded into the same plate, and all of the ligands were tested within the same read. BRET measurements were performed using a PHERAstar fsx plate reader (BMG Labtech), equipped with a BRET2 filter cube (BMG Labtech, 410/530 nm emission). The baseline was recorded for 5 min followed by 7 min stimulation with the different ligands with a time interval of 60 s between each datapoint. The inverse BRET ratios 410/530 of each well were normalized to the baseline of negative control (set 0%) and ACh maximum concentration (set 100%) for each mutant. The data were analysed using Microsoft Excel 2016.

**Internalization assay**

The SP-$M_2R$-WT-eGFP and the SP-$M_2R^{XXXTAG}$ constructs were expressed in HEK293T cells and labelled with Tet–Cy3 as described in the previous sections. After addition of ACh (400 µM), cells were incubated for 1 h at 37 °C. Images were acquired using the Axio Observer.Z1/7 microscope (Zeiss), equipped with a C-Apochromat ×63/1.2 NA W Korr DICII objective, an AxioCam 705 mono with Duolink camera and a LED Colibri 5 light source. Filters used (bandpass/bandstop in nm): eGFP, 450–490/495 (excitation), and 500/550 nm (emission); Cy3, 545–565/575 (excitation) and 579–604 nm (emission). Images were acquired using Zen blue 2.3 lite software (Zeiss).

**Data analysis and statistics**

Plotting, curve fitting and statistical analyses were performed using Prism v.7.0 or newer (GraphPad). Drift corrections in fluorescence imaging was performed using Origin 2022 (OriginLab). A Gaussian distribution of all datapoints was tested using the D'Agostino–Pearson omnibus normality test in Prism 7.0.

Statistic differences of two groups were assessed using two-tailed unpaired Student's $t$-tests. A two-tailed paired $t$-test was performed when comparing various data obtained from the same single cell. In the case of normal distribution, a parametric test was used. Welch's correction was further performed in the case of unequal variance. Statistic differences of three or more populations have been assessed using a parametric one-way ANOVA. When comparing to a reference group of data, a Tukey's post hoc test was performed. In the case of multiple comparisons, a Dunnett's test was used. For all statistical analyses, the confidence interval was set to 95% ($P = 0.05$). All experiments were performed with samples from independent experiments and repeated at least three times. The number of individually analysed cells is referred to as $n$. Details about $n$ and the statistical test performed are stated in the appropriate figure legend.

Molecular models of receptors and analyses thereof were performed with UCSF ChimeraX, developed by the Resource for Biocomputing, Visualization, and Informatics at the University of California, San Francisco, with support from National Institutes of Health R01-GM129325 and the Office of Cyber Infrastructure and Computational Biology, National Institute of Allergy and Infectious Diseases[65]. Schematics were created using BioRender.

For Fig. 2, the $\Delta F/F_0$ of each condition was normalized to the $\Delta F/F_0$ obtained for ACh within the same cell. For Fig. 3c,f,i and Supplementary Fig. 5, data from overexpression of the $G\alpha_{oA}$(G203T) mutant were normalized to data obtained at endogenous G-protein levels using the following equation: Norm $\Delta\Delta G_{mut} = ((G_{mut} - mean_{endo})/mean_{endo}) \times 100\%$, where $G_{mut}$ represents the $\Delta F/F_0$ of the respective $M_2R$ biosensor when co-expressed with $G\alpha_{oA}$(G203T), and $mean_{endo}$ refers to the mean $\Delta F/F_0$ at endogenous G-protein levels (Fig. 2 and Supplementary Table 1). For Supplementary Fig. 6, the $\Delta F/F_0$ of each datapoint obtained from activation of the seven $M_2R$ biosensors with the ligands Ixo, Arec and Pilo was normalized to the mean $\Delta F/F_0$ of ACh at endogenous G-protein expression levels or after overexpression of $G\alpha_{oA}$(G203T).

For Fig. 4, three-dimensional surface plots (Fig. 4) were generated using the NumPy and matplotlib libraries of Python (v.3.7.9)[66,67]. The plots were generated as abstract Gaussian peaks modelled using the Gaussian function: $z(x, y) = A \times e^{(-((x-x_0)2+(y-y_0)2)/2\sigma 2)}$ with $A$ as the mean $\Delta F/F_0$ relative to the plane $z = 0$. $M_2R$ biosensors indicating the same receptor–G-protein complex were assigned similar colours. For each ligand-dependent receptor/G-protein complex at a given timepoint after agonist activation, the colour intensities from all positions were summed, and the respective plane was labelled in that colour.

**Reporting summary**

Further information on research design is available in the Nature Portfolio Reporting Summary linked to this article.

## Data availability

All data supporting the findings of this study are available in the Article and its Supplementary Information. Raw videos and images from microscopy experiments are available from the corresponding authors on request. The atomic coordinates used to generate the receptor model in Fig. 1 are available from the Protein Data Bank under accession codes 3UON and 4MQS. Source data are provided with this paper.

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

**Acknowledgements** We thank J.-P. Pin for providing the plasmid encoding the Ga$_o$ mutant and B. Kobilka for discussions. This study was funded by the Deutsche Forschungsgemeinschaft (German Research Foundation) through SFB1423, project number 421152132, subprojects C05 (to I.C. and A.B.), C06 (to C.S.) and C08 (to M.J.L. and A.B.); and through project 407707190 (to C.S.) and CO822/5-1 (671228, NanoBelt) (to I.C.). R.T. is supported by an Add-on Fellowship for Interdisciplinary Life Science from the Joachim Herz Foundation; and A.-D.L. by the Pre-Doc Award of Leipzig University. A.B. acknowledges funding by the Else Kröner-Fresenius-Stiftung. Figures 1, 3 and 5 and Extended Data Figs. 1 and 8 were created in part using BioRender with the publication licence LO27JFUF6I.

**Author contributions** I.C. and A.B. conceptualized the study. R.T. engineered and characterized the biosensor panel and performed most of the experiments. P.S.J. and C.D. established and performed TRUPATH assays. P.S.J. performed G$_i$-activation assays in single cells. C.D.F. performed the initial expression and labelling screen of M$_2$R mutants and internalization assays. A.-D.L. and C.S. performed cell-surface expression experiments with the help of P.S.J.; C.D. and B.M. established the biosensor technology in the new laboratory of A.B. in Mainz, Germany during revision of this paper. H.-J.M. performed experiments with the tyrosine lid mutant. R.T., P.S.J., C.D.F., C.D., H.-J.M. and A.B. analysed data and generated figures. M.B. provided valuable support in the structural interpretation and visualization of the biosensor dataset. M.J.L. provided critical ideas and contributed extensively to discussions. I.C. and A.B. supervised the project. A.B. wrote the paper with input from M.J.L. and I.C.

**Funding** Open access funding provided by Johannes Gutenberg-Universität Mainz.

**Competing interests** The authors declare no competing interests.

**Additional information**
**Correspondence and requests for materials** should be addressed to Irene Coin or Andreas Bock.

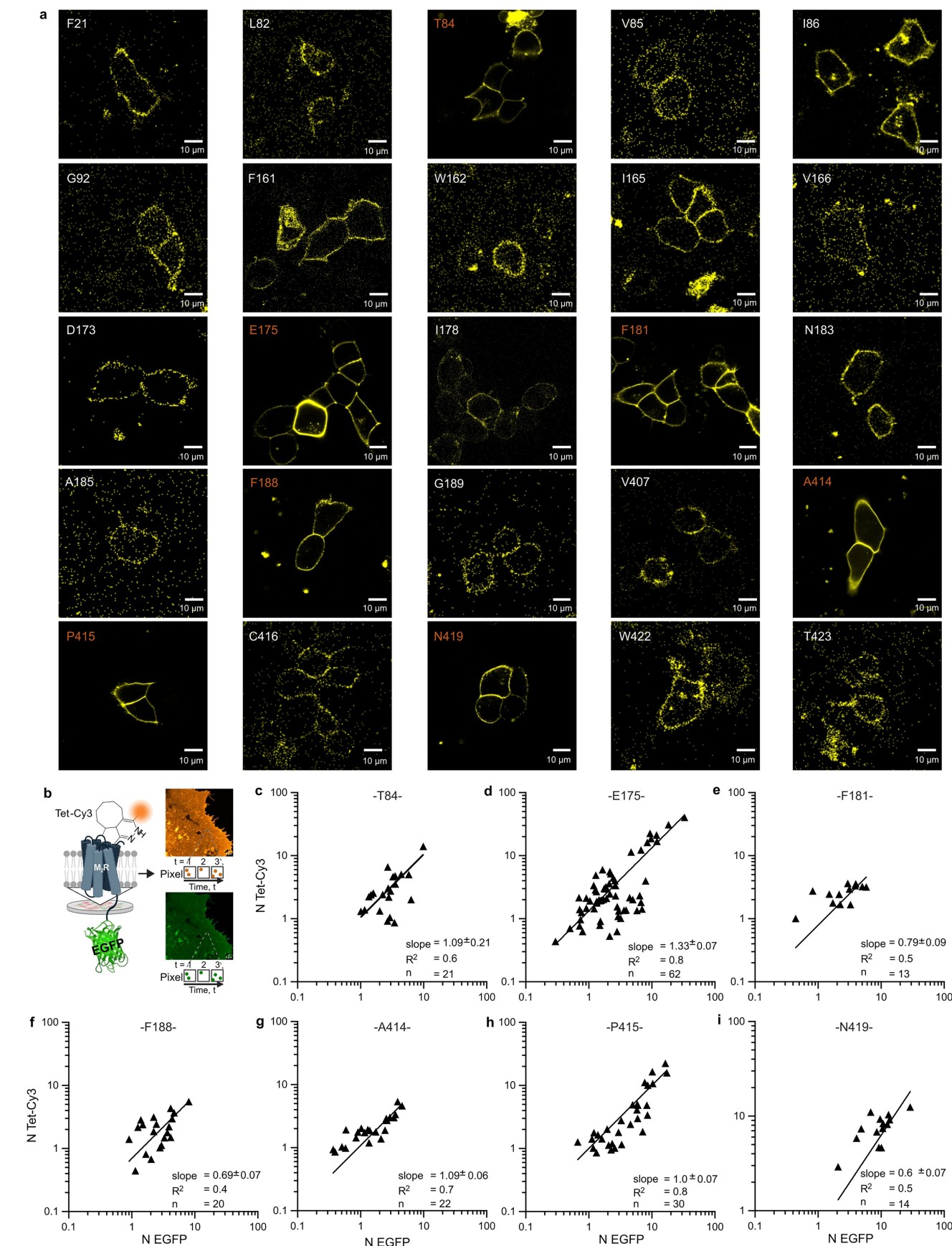

**Extended Data Fig. 1** | See next page for caption.

**Extended Data Fig. 1 | Labelling efficiency of M$_2$ receptor biosensors.**
(a) Shown are representative confocal images obtained from the M$_2$R labelling screen. M$_2$R constructs (*SP-M$_2$R$^{XXXTAG}$*) incorporating the non-canonical amino acid TCO*K at the indicated positions (amino acid numbering of the native human M$_2$ receptor) could be labelled with Tet-Cy3 on at least 3 independent days of experiments. Labelled M$_2$R constructs that show agonist-induced fluorescence intensity changes (see Fig. 1c), are highlighted in orange. (b) For quantification of labelling efficiencies, M$_2$R constructs, each comprising a C-terminal EGFP, were transiently expressed in HEK-AD cells (*SP-M$_2$R$^{XXXTAG}$-EGFP*). The average numbers (N) of counted Tet-Cy3 labelled molecules were obtained using molecular brightness analysis[41]. The number of Tet-Cy3 labelled molecules was then compared to the number of molecules counted in the EGFP channel. Created in BioRender. Thomas, R. (2025) https://BioRender.com/loxqlqf. (c-i) The average number N of counted Tet-Cy3 labelled molecules (y-axis) and EGFP (x-axis), obtained from approx. 100 images, is plotted in a log-log-scale. Each data point indicates a single HEK-AD cell analysed from overall 3 independent experiments. The data were fitted by linear regression with constraints at X,Y = 0. The fits are shown as slope ± sd. n, number of cells. $R^2$, goodness of fit.

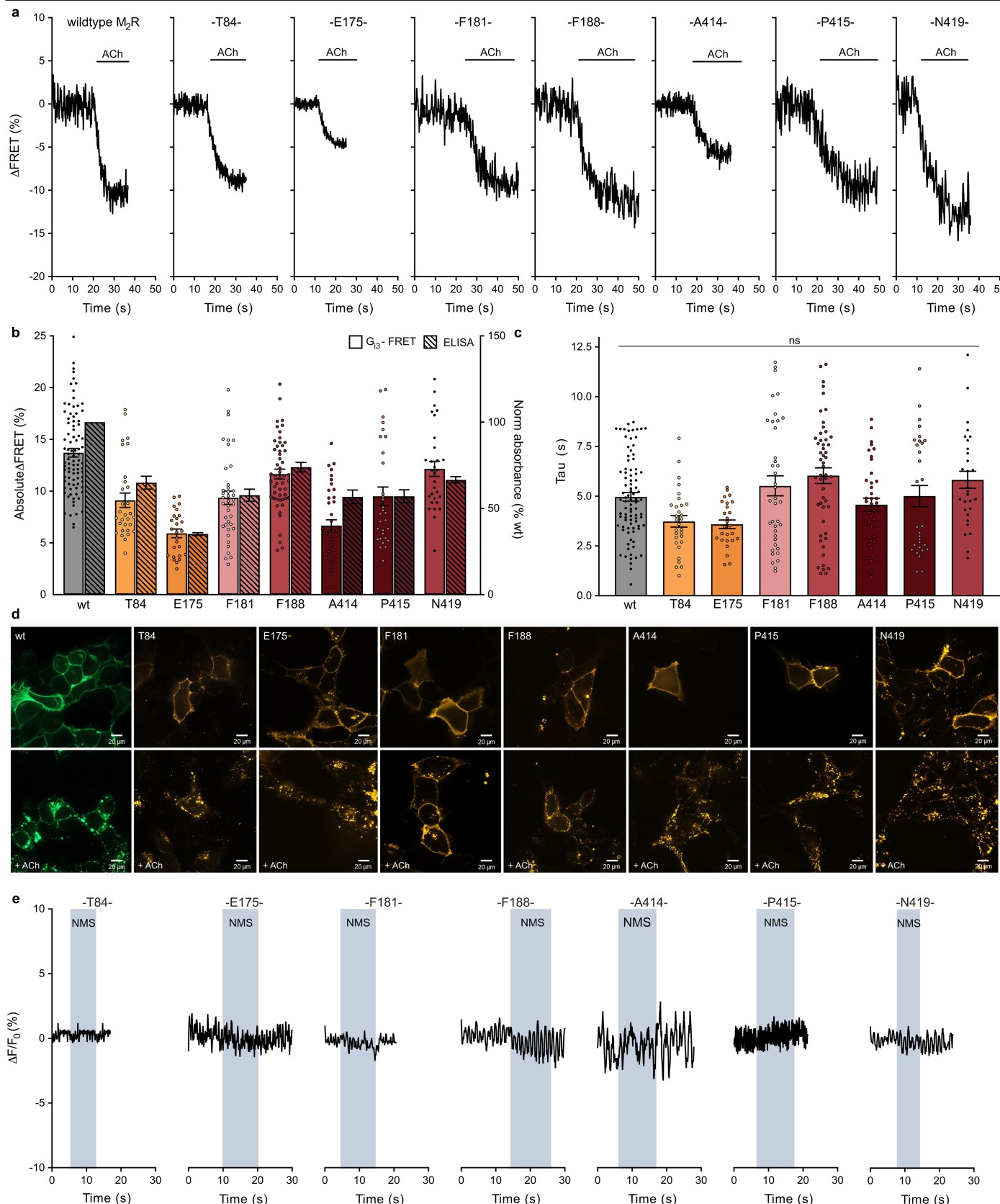

**Extended Data Fig. 2 |** See next page for caption.

**Extended Data Fig. 2 | Functional characterization of M₂R biosensors.**

(a) Representative ΔFRET traces from single HEK293T cells transiently expressing the Gα$_{i3}$-FRET biosensor with wild-type (*SP-M₂R-wt*) or mutant receptors (*SP-M₂R$^{XXXTAG}$*) labelled with Tet-Cy5 at the indicated positions before and after activation with 300 μM ACh. (b) Comparison of mean ΔFRET responses (left y-axis) and receptor surface expression (right y-axis) for all seven M₂R biosensors and M₂R-wt. Surface expression was determined by ELISA in 4 independent experiments using N-terminal HA-tagged constructs (*SP-HA-M₂R-wt* or *SP-HA-M₂R$^{XXXTAG}$*). Unshaded bars indicate absolute ΔFRET from the Gα$_{i3}$-FRET assay, dashed bars the ELISA absorbance normalized to wild-type (100%). Reduced receptor expression correlates with decreased maximal G-protein activation. (c) Apparent activation kinetics (Tau (τ), s) derived from the Gα$_{i3}$-FRET assay. Bars show mean ± s.e.m.; each point represents one cell. ns, not significant (one-way ANOVA with Dunnett's post-hoc test vs. M₂R-wt). M₂R$^{XXX}$ (n = X number of individual cells examined over Y number of independent experiments): M₂R-wt (89; 5), M₂R$^{84}$ (30; 6), M₂R$^{175}$ (26; 6), M₂R$^{181}$ (39; 5), M₂R$^{188}$ (52; 5), M₂R$^{414}$ (39; 4), M₂R$^{415}$ (30; 5), M₂R$^{419}$ (30; 4). (d) All seven biosensors internalize upon agonist addition. Representative epifluorescence images of *SP-HA-M₂R$^{XXXTAG}$* constructs labelled with Tet-Cy3 before (top row) and after (bottom row) addition of 400 μM ACh. *SP-M₂R-wt-EGFP* served as a positive control. Green, GFP channel; orange, Cy3 channel. (e) M₂R biosensors do not respond to the inverse agonist NMS. Representative ΔF/F$_0$ traces of the indicated *SP-HA-M₂R$^{XXXTAG}$* biosensors labelled with Tet-Cy3 and stimulated with 100 μM NMS. Shaded areas denote NMS exposure; unshaded areas indicate buffer application. M₂R$^{XXX}$ (n = X number of individual cells examined over Y number of independent experiments): M₂R$^{84}$ (20; 6), M₂R$^{175}$ (21; 6), M₂R$^{181}$ (19; 4), M₂R$^{188}$ (25; 3), M₂R$^{414}$ (30; 7), M₂R$^{415}$ (16; 4), M₂R$^{419}$ (20; 5).

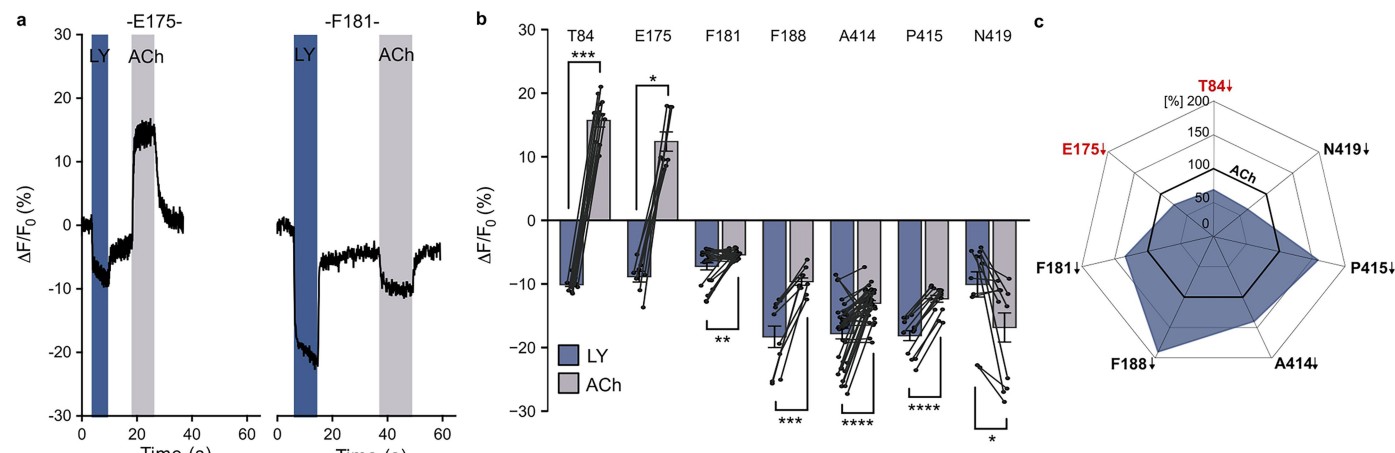

**Extended Data Fig. 3 | Conformational fingerprint of a positive allosteric modulator.** (a) Representative fluorescence intensity changes ($\Delta F/F_0$) recorded in real-time from single HEK293T cells expressing indicated $M_2R$ biosensors superfused with 30 μM LY2119620 (LY), followed by 1 mM ACh after wash out with buffer. Ligand application is indicated by shaded areas in different colours. Non-shaded areas indicate buffer wash-out. (b) Mean fluorescence intensity changes ($\Delta F/F_0$) at all seven biosensors after addition of LY and ACh. Positive values indicate an increase in fluorescence, negative values indicate a fluorescence decrease. Bars indicate means ± sem, each data point represents a single cell. $M_2R^{xxx}$ (n = X number of individual cells examined over Y number of independent experiments): $M_2R^{84}$ (12; 3), $M_2R^{175}$ (8; 3), $M_2R^{181}$ (23; 3), $M_2R^{188}$ (10; 3), $M_2R^{414}$ (35; 8), $M_2R^{415}$ (13; 4), $M_2R^{419}$ (11; 4). ****$p < 0.0001$, ***$p < 0.001$ (T84: 0.0006, F188: 0.0002), **$p < 0.01$ (F181: 0.0041), *$p < 0.05$ (E175: 0.0147, N419: 0.0166), according to a two-tailed paired t-test. (c) Radar plot of mean fluorescence intensity changes $\Delta F/F_0$ in response to LY, normalized to $\Delta F/F_0$ of 1 mM ACh in the same cell. The bold line indicates ACh (set to 100%). The direction of $\Delta F/F_0$ is indicated with arrows (decrease, down). The change of direction in fluorescence emission for $M_2R^{84}$ and $M_2R^{175}$ upon LY stimulation compared to ACh stimulation is highlighted in red.

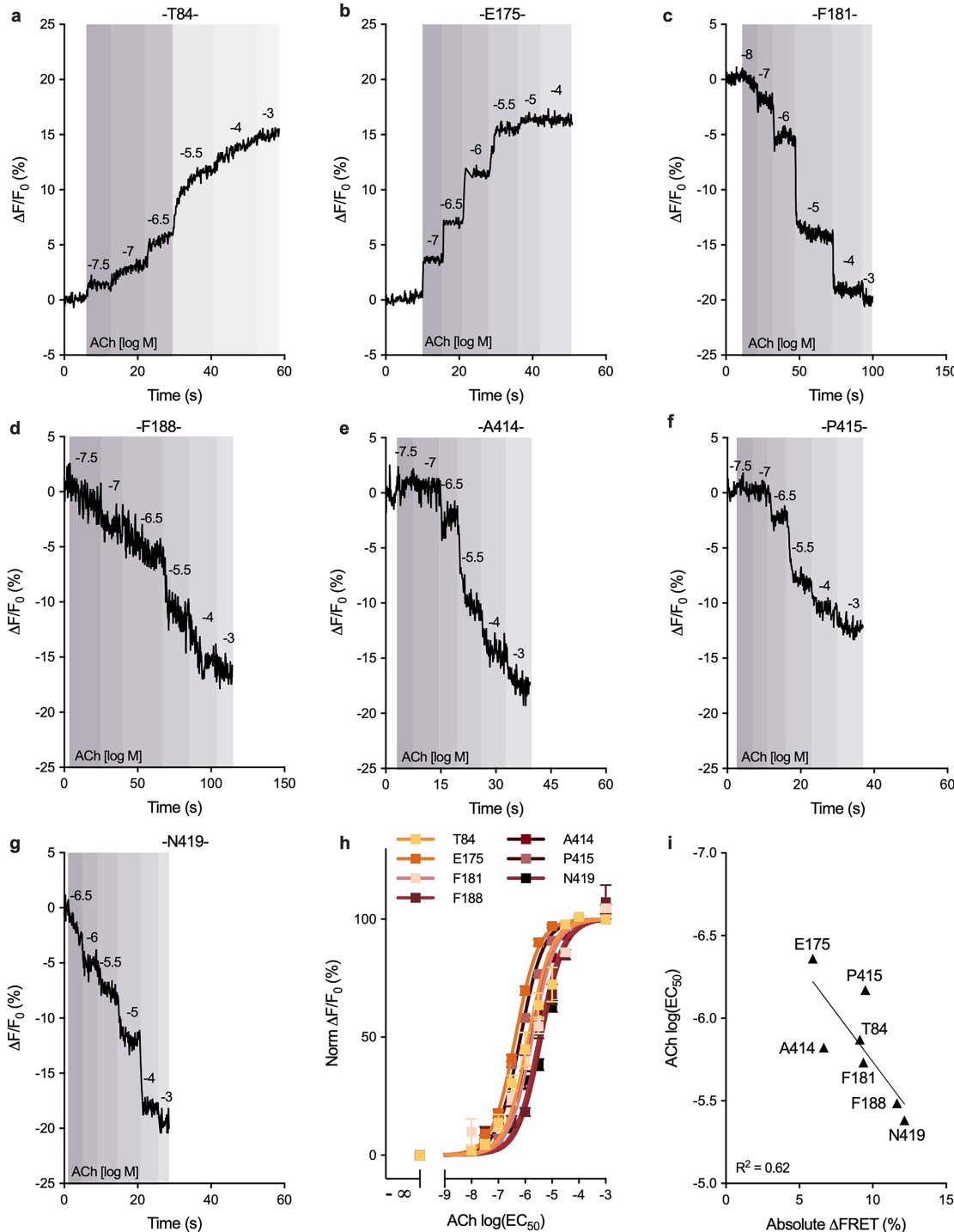

**Extended Data Fig. 4 | ACh potencies are similar across all seven M₂R biosensors.** (a-g) Representative traces of concentration-dependent changes of fluorescence intensity ($\Delta F/F_0$) of a single cell expressing the indicated M₂R biosensor (*SP-M₂R^XXXTAG*), stimulated with increasing concentrations of ACh, indicated as log (M) and by a grey colour gradient. (h) Concentration-effect-curves of normalized, mean ACh-mediated fluorescence intensity changes in response to increasing concentrations of ACh. ACh potencies ($\log_{EC50} \pm$ sem):

T84, $-5.9 \pm 0.05$; E175, $-6.4 \pm 0.01$; F181, $-5.7 \pm 0.03$; F188, $-5.5 \pm 0.03$; A414, $-5.8 \pm 0.02$; P415, $-6.2 \pm 0.02$; N419, $-5.4 \pm 0.05$. n = 10 (T84), n = 19 (E175), n = 21 (F181), n = 20, n = 20 (188), n = 29 (A414), n = 15 (P415), n = 13 (N419). Data obtained from 3 independent experiments. (i) Plot of maximal $\Delta$FRET responses of the $G\alpha_{i3}$-FRET assay (see Extended Data Fig. 2) and ACh potencies at the seven biosensors. The data were fitted by linear regression. $R^2$, goodness of fit.

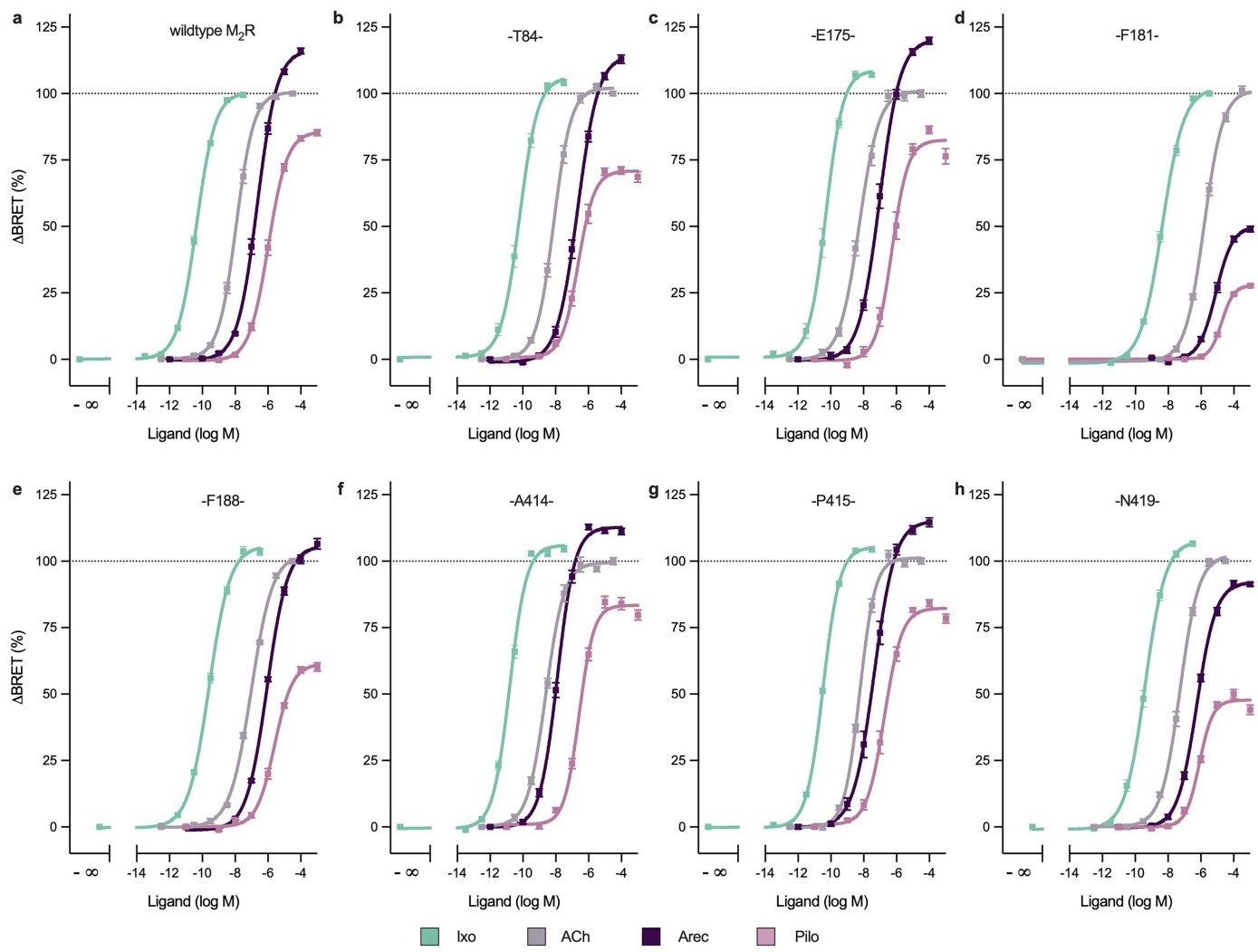

**Extended Data Fig. 5 | Ligand-dependent, M₂R biosensors-mediated activation of G$_{oA}$ proteins.** (a-h) Concentration-dependent activation of G$_{oA}$ proteins determined by the TRUPATH BRET assay upon activation of the indicated M₂R variant (*SP-M₂R-wt*) or (*SP-M₂R$^{XXXXTAG}$*) by Ixo (green), ACh (grey), Arec (purple), and Pilo (pink). Concentration-dependent BRET changes (ΔBRET) were normalized to buffer (set to 0%) and the maximal response of ACh (set to 100%), indicated by the dotted line. Shown are the means ± sem of 3 (M₂R biosensors) and 4 (M₂R wildtype) independent experiments (see Supp. Table 2).

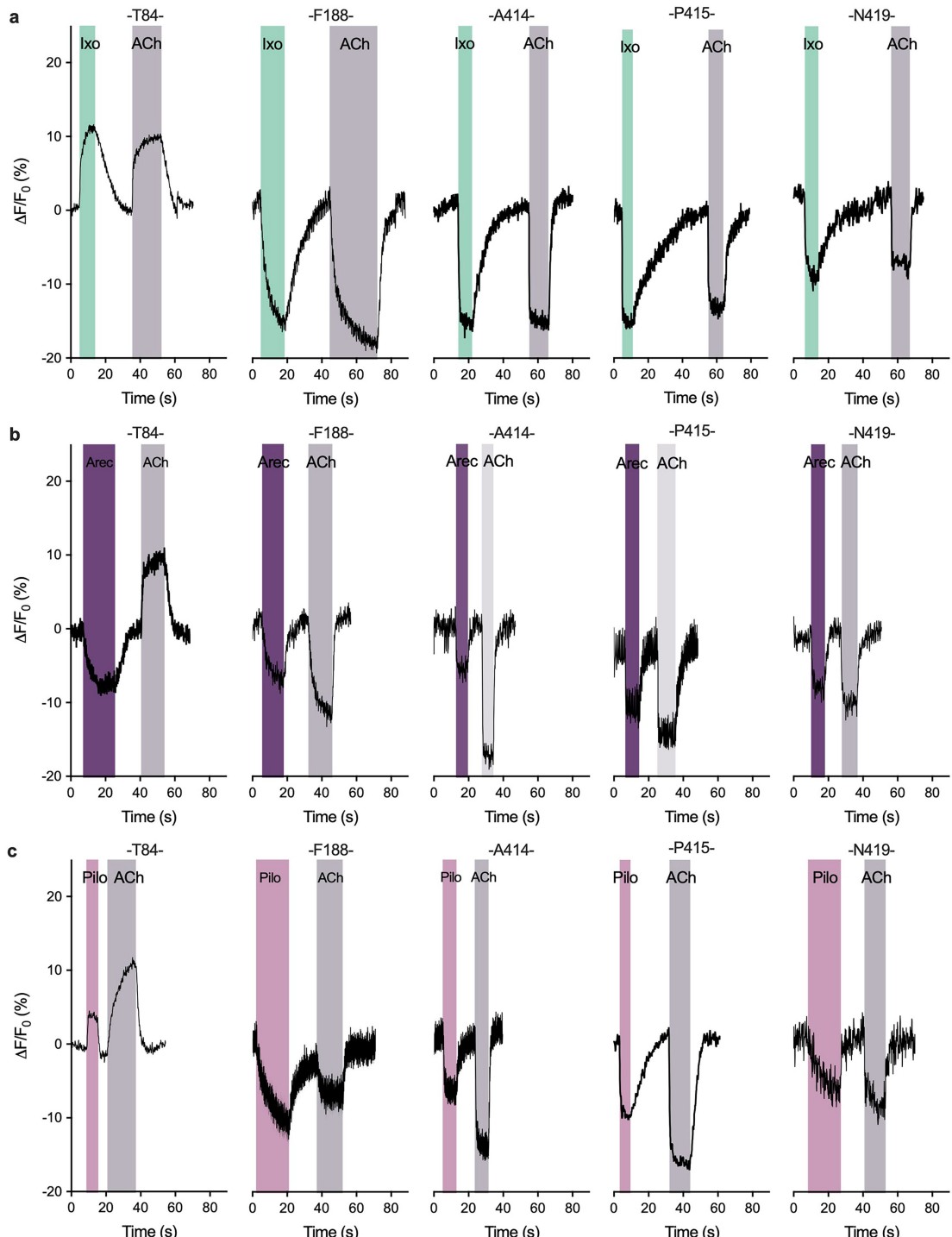

**Extended Data Fig. 6 | M₂R biosensors respond to different agonists to various degrees.** Representative traces of $\Delta F/F_0$ of a single cell expressing the indicated M₂R biosensor ($SP\text{-}M_2R^{XXXTAG}$) and stimulated with (a) 100 µM Ixo (green), (b) 1 mM Arec (purple), or (c) 10 mM Pilo (pink) and 1 mM ACh (grey) after washout with buffer, respectively (see Supp. Table 1). Shaded areas indicate the addition of the respective agonist, unshaded areas represent buffer application.

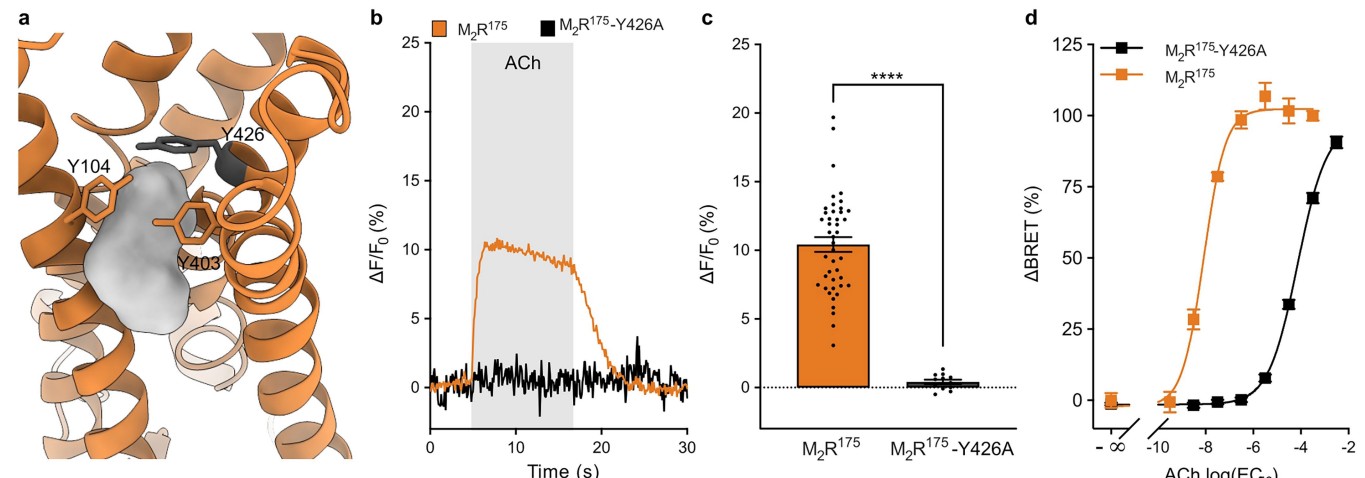

**Extended Data Fig. 7 | A tyrosine-lid mutation impedes the M₂R¹⁷⁵ biosensor-response to Ach.** (a) Side-view on the X-ray crystal structure of the active M₂R (orange, PDB: 4MQS). A tyrosine lid is formed on top of iperoxo (light grey). The mutated tyrosine residue (Y426$^{7.39}$) is highlighted in black. (b) Representative fluorescence intensity changes (ΔF/F₀) recorded in real-time from single HEK293T cells expressing the M₂R¹⁷⁵ (orange) or the M₂R¹⁷⁵-Y426A mutant (black) biosensors superfused with 1 mM ACh (shaded area). Non-shaded areas indicate buffer application. (c) ACh-induced mean fluorescence intensity changes (ΔF/F₀) of the M₂R¹⁷⁵ (orange) or the M₂R¹⁷⁵-Y426A mutant (black) biosensors. Bars indicate means ± sem, each data point represents a single cell. M₂Rˣˣˣ (n = X number of cells examined over Y number of independent experiments): M₂R¹⁷⁵ (44; 11), M₂R¹⁷⁵-Y426A (16; 3). ****p < 0.0001, according to an two-tailed unpaired t-test with Welch's correction. The data for the M₂R¹⁷⁵ biosensor are replotted from Fig. 1c and d. (d) Concentration-response curves of agonist-induced G-protein activation of the M₂R¹⁷⁵ (orange) or the M₂R¹⁷⁵-Y426A mutant (black) biosensors as determined by the TRUPATH BRET-based assay. Concentration-dependent BRET changes (ΔBRET) were normalized to buffer (set to 0 %) and the maximal response of ACh (set to 100%) at the M₂R¹⁷⁵ biosensor. Shown are the means ± sem of 3 independent experiments (see Supp. Table 2). Constructs used: *SP-M₂R$^{E175TAG}$* and *SP-M₂R$^{E175TAG}$-Y426A*.

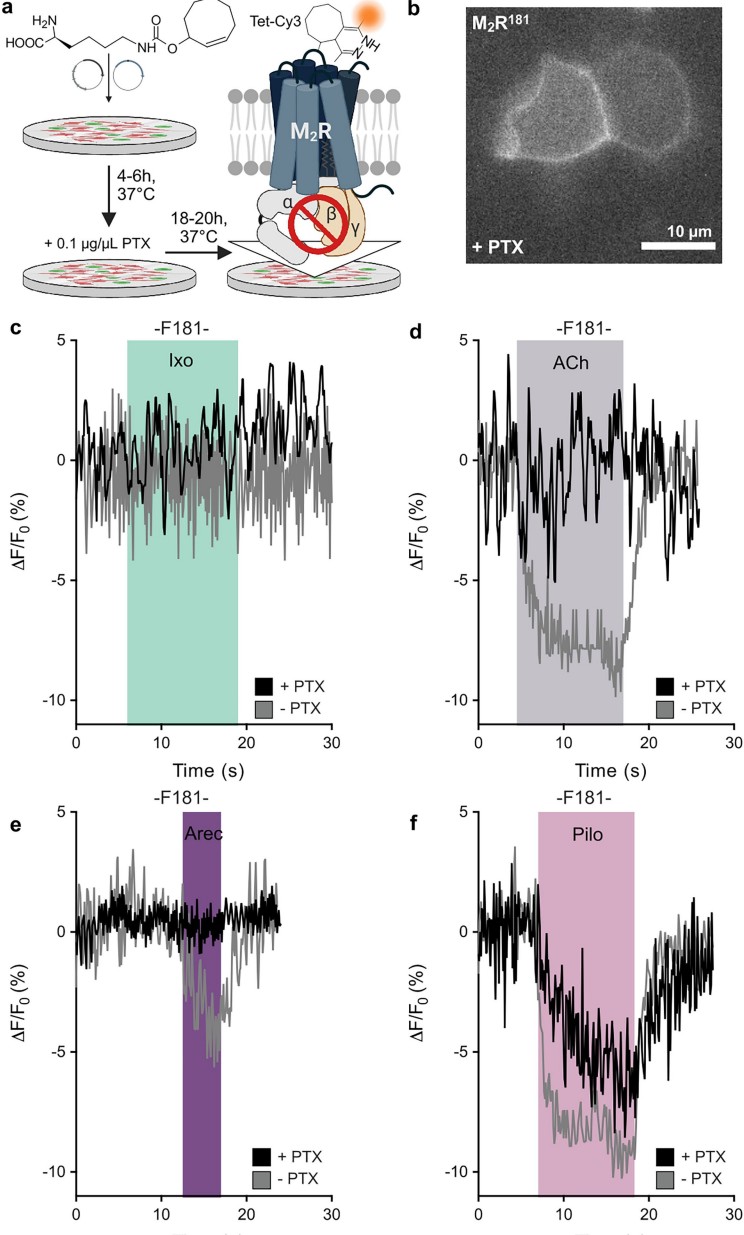

**Extended Data Fig. 8 | PTX-pretreatment decreases or abolishes agonist-promoted fluorescence intensity changes.** (a) Sketch of the experimental procedure: 4–6 h after transfection with *SP-M₂R^F181TAG*, HEK293T cells were treated with 0.1 µg/µL PTX and grown at 37 °C for 18–20 h before labelling with Tet-Cy3. Created in BioRender. Thomas, R. (2025) https://BioRender.com/f79d908. (b) Epifluorescence image of two representative HEK293T cells expressing the M₂R^181 biosensor, treated with PTX, and labelled with Tet-Cy3.

(c-f) Representative traces of M₂R^181 expressed in HEK293T cells, treated with PTX (black, see Fig. 2) or without PTX-treatment (grey), and stimulated with 100 µM Ixo (c), 1 mM Ach (d), 1 mM Arec (e), or 10 mM Pilo (f). Shaded areas indicate agonist application, unshaded areas represent buffer application. Data were obtained from 3 independent experiments. n (number of single cells, -PTX, +PTX); n = 7, 9 (Ixo), n = 18, 13 (ACh), n = 10, 8 (Arec), n = 10, 13 (Pilo).

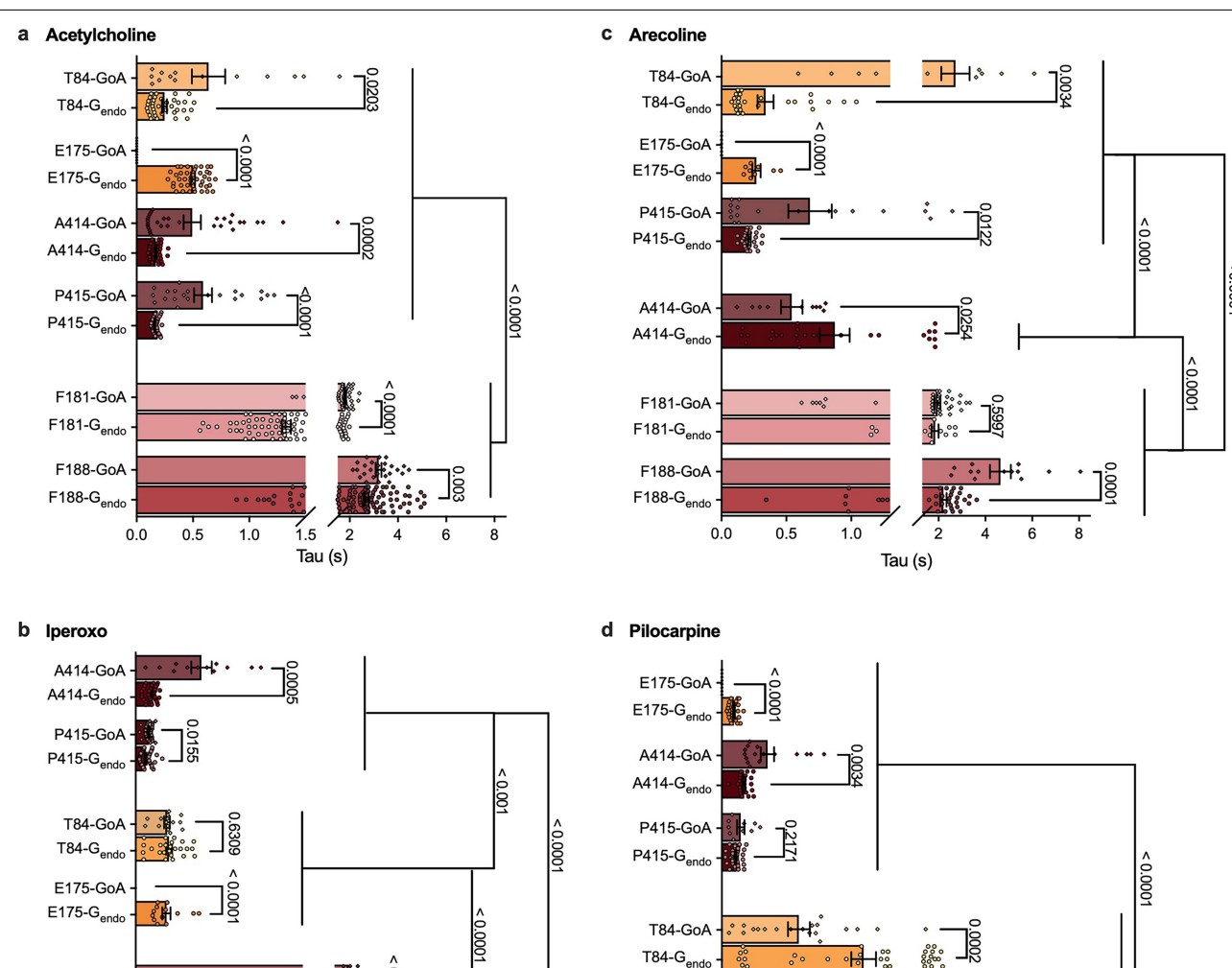

**Extended Data Fig. 9 | Time-resolved dynamics of M₂R/G protein-signalling complexes.** Comparison of apparent on-rates (Tau in s) obtained from six biosensors (*SP-M₂R^XXXTAG*) after activation with the indicated ligands 1 mM ACh (a), 100 µM Ixo (b), 1 mM Arec (c), and 10 mM Pilo (d) at endogenous G-protein levels (endo) and after Gα_oA(G203T) overexpression. Bars indicate means ± sem, each data point represents a single cell (see Supp. Table 3). For Gα_oA(G203T) overexpression: M₂R^XXX (per agonist: n = X number of cells examined over Y number of independent experiments): M₂R^84 (Ixo: 15; 5, ACh: 14; 7, Arec: 10; 3, Pilo: 20; 4), M₂R^175 (Ixo: 11; 3, ACh: 10; 3, Arec: 9; 3, Pilo: 11; 3), M₂R^181 (Ixo: 29; 4, ACh: 41; 7, Arec: 34; 4, Pilo: 20; 3), M₂R^188 (Ixo: 16; 3, Arec: 13; 3, Pilo: 17; 3),

M₂R^414 (Ixo: 12; 4, ACh: 31; 9, Arec: 10; 3, Pilo: 16; 3), M₂R^415 (Ixo: 28; 5, ACh: 21; 6, Arec: 19; 5, Pilo: 10; 3). The number of individual cells and independent experiments for the endogenous G-protein condition are listed in the legends of Figs. 1 and 2. P values, according to according to a two-tailed unpaired t-test with Welch's correction are plotted in (a-d). The biosensors are classified into statistically different groups, according to a one-way analysis of variance (ANOVA multiple comparisons test) with Tukey's posthoc test; ns, not significantly different. Biosensors that are grouped are not significantly different from each other.

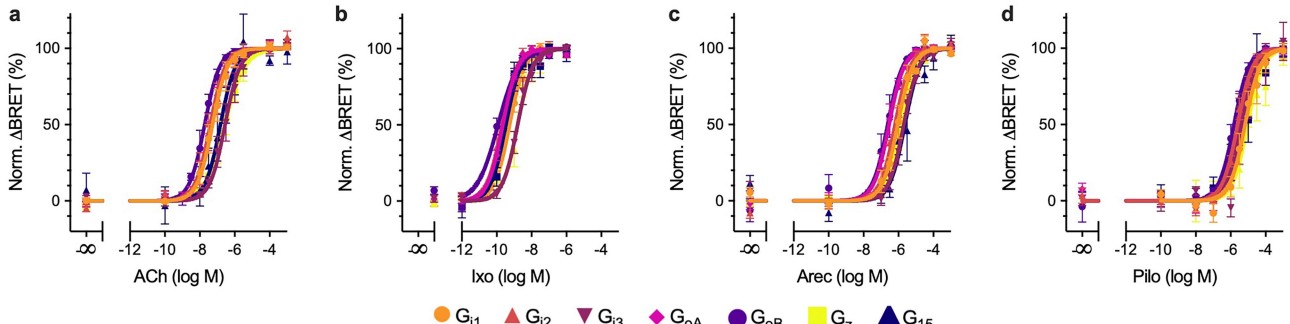

**Extended Data Fig. 10 | Concentration-effect curves of agonist-mediated G-protein activation at M₂R-*wt*.** The concentration-effect curves were generated from experiments as performed in Fig. 5. Concentration-dependent ligand-dependent BRET changes (ΔBRET) were normalized to buffer (set to 0 %) and the agonists' maximal effects (set to 100 %). (a) ACh, (b) Ixo, (c) Arec, and (d) Pilo. Shown are the means ± sem. n= number of independent experiments (for Ixo, ACh, Arec, Pilo): 4, 6, 4, 6 (G$_{i1}$), 4, 7, 4, 6 (G$_{i2}$), 4, 6, 4, 5 (G$_{i3}$), 6, 5, 3, 6 (G$_{oA}$), 5, 8, 3, 5 (G$_{oB}$), 5, 6, 4, 4 (G$_z$), 6, 6, 4, 5 (G$_{15}$), see Supp. Table 4.

Andreas Bock

# Reporting Summary

## Statistics

For all statistical analyses, confirm that the following items are present in the figure legend, table legend, main text, or Methods section.

| n/a | Confirmed | |
|---|---|---|
| ☐ | ☒ | The exact sample size (*n*) for each experimental group/condition, given as a discrete number and unit of measurement |
| ☐ | ☒ | A statement on whether measurements were taken from distinct samples or whether the same sample was measured repeatedly |
| ☐ | ☒ | The statistical test(s) used AND whether they are one- or two-sided<br>*Only common tests should be described solely by name; describe more complex techniques in the Methods section.* |
| ☒ | ☐ | A description of all covariates tested |
| ☐ | ☒ | A description of any assumptions or corrections, such as tests of normality and adjustment for multiple comparisons |
| ☐ | ☒ | A full description of the statistical parameters including central tendency (e.g. means) or other basic estimates (e.g. regression coefficient) AND variation (e.g. standard deviation) or associated estimates of uncertainty (e.g. confidence intervals) |
| ☐ | ☒ | For null hypothesis testing, the test statistic (e.g. *F*, *t*, *r*) with confidence intervals, effect sizes, degrees of freedom and *P* value noted<br>*Give P values as exact values whenever suitable.* |
| ☒ | ☐ | For Bayesian analysis, information on the choice of priors and Markov chain Monte Carlo settings |
| ☒ | ☐ | For hierarchical and complex designs, identification of the appropriate level for tests and full reporting of outcomes |
| ☒ | ☐ | Estimates of effect sizes (e.g. Cohen's *d*, Pearson's *r*), indicating how they were calculated |

*Our web collection on statistics for biologists contains articles on many of the points above.*

## Software and code

Policy information about availability of computer code

| Data collection | Single-cell fluorescence microscopy data were collected with VisiView 4.0 software (Visitron Systems), temporal brightness data with LASX software 3.5.7.23225 (Leica Biosystems), Images from internalisation experiments were collected with the Zen blue 2.3 lite software (Zeiss), the Gi3-FRET activation with MicroManager 2.0, TRUPATH data with Gen5 3.11 software (Agilent Technologies), and ELISA data were collected with EnVision Manager 1.13.3. |
|---|---|
| Data analysis | The following published software was used to analyse the data as referenced in Methods: Origin2022b 9.9.9.171 (OriginLab); ImageJ 1.5.4f, (NIH), GraphPad Prism 7.0 or newer (Graphpad Software), Microsoft Excel 2016, Python 3.7.9 (Python software foundation). |

For manuscripts utilizing custom algorithms or software that are central to the research but not yet described in published literature, software must be made available to editors and reviewers. We strongly encourage code deposition in a community repository (e.g. GitHub). See the Nature Portfolio guidelines for submitting code & software for further information.

## Data

Policy information about availability of data

All manuscripts must include a data availability statement. This statement should provide the following information, where applicable:

- Accession codes, unique identifiers, or web links for publicly available datasets
- A description of any restrictions on data availability
- For clinical datasets or third party data, please ensure that the statement adheres to our policy

All data that support the findings of this study are available in the manuscript, Extended Data and Supplementary Information. Raw movies and images from microscopy experiments are available from the corresponding authors upon request. The atomic coordinates used to generate the receptor model in Figure 1 are available from the Protein Data Bank under PDB ID:3UON and ID:4MQS. Source data are provided with this paper.

## Research involving human participants, their data, or biological material

Policy information about studies with human participants or human data. See also policy information about sex, gender (identity/presentation), and sexual orientation and race, ethnicity and racism.

| | |
|---|---|
| Reporting on sex and gender | N/A |
| Reporting on race, ethnicity, or other socially relevant groupings | NA |
| Population characteristics | N/A |
| Recruitment | N/A |
| Ethics oversight | N/A |

Note that full information on the approval of the study protocol must also be provided in the manuscript.

# Field-specific reporting

Please select the one below that is the best fit for your research. If you are not sure, read the appropriate sections before making your selection.

☒ Life sciences ☐ Behavioural & social sciences ☐ Ecological, evolutionary & environmental sciences

For a reference copy of the document with all sections, see nature.com/documents/nr-reporting-summary-flat.pdf

# Life sciences study design

All studies must disclose on these points even when the disclosure is negative.

| | |
|---|---|
| Sample size | Single-cell fluorescence imaging: For single-cell fluorescence imaging, sample size corresponded to the number of individual cells analyzed. No formal sample-size calculation was performed because the experiments involve large, independently measured cell populations where increasing the number of cells primarily reduces measurement noise rather than altering effect size estimation. Sample sizes were determined based on established practice in the field with a minimum of 3 independent experiments. Other experiments (plate-reader functional assays): For all functional assays performed in the plate reader, no statistical sample-size calculation was performed either. Instead, sample sizes were chosen based on prior experience with these well-established assays, where effect sizes are robust and biological variability is low under controlled culture conditions. We routinely use at least three independent biological replicates, each measured with technical replication, because these replicate numbers have historically been sufficient to detect the expected differences with appropriate statistical power and to yield consistent results across independent experiments. The chosen sample sizes therefore provide reliable estimation of mean values and variance and are standard for this experimental system. |
| Data exclusions | No data were excluded from the analysis. |
| Replication | All experiments were successfully reproduced through at least 3 independent biological replicates. |
| Randomization | Randomization was not relevant to this study because all experimental conditions were predefined by the study design (e.g., specific treatments such as different agonists, G proteins etc. and specific biosensor constructs) and were performed on parallel cultures of the same cell line under tightly controlled laboratory conditions. Samples were not "assigned" from a heterogeneous population but were generated directly by applying the prescribed experimental manipulation to each group. As a result, there were no covariates or sources of allocation bias that randomization would mitigate. All samples within each experiment were processed on the same day using identical protocols, which effectively controls for technical variability and ensures that group differences arise solely from the intended experimental manipulation. |
| Blinding | Blinding was not relevant to this study because the experimental readouts were objective, quantitative measurements (e.g., plate-reader |

| Blinding | signals, fluorescence imaging, or software-based analysis) that do not involve subjective assessment by the experimenter. Samples were processed in parallel using standardized protocols, and data acquisition and quantification were performed using controlled instrument settings and predefined analysis parameters. As a result, experimenter knowledge of sample identity could not influence the measurement or interpretation of the results, and blinding would not have altered the outcome of the analysis. |
|---|---|

# Reporting for specific materials, systems and methods

We require information from authors about some types of materials, experimental systems and methods used in many studies. Here, indicate whether each material, system or method listed is relevant to your study. If you are not sure if a list item applies to your research, read the appropriate section before selecting a response.

## Materials & experimental systems

| n/a | Involved in the study |
|---|---|
| ☐ | ☒ Antibodies |
| ☐ | ☒ Eukaryotic cell lines |
| ☒ | ☐ Palaeontology and archaeology |
| ☒ | ☐ Animals and other organisms |
| ☒ | ☐ Clinical data |
| ☒ | ☐ Dual use research of concern |
| ☒ | ☐ Plants |

## Methods

| n/a | Involved in the study |
|---|---|
| ☒ | ☐ ChIP-seq |
| ☒ | ☐ Flow cytometry |
| ☒ | ☐ MRI-based neuroimaging |

## Antibodies

| Antibodies used | An Anti-HA-Peroxidase, High Affinity from rat IgG1 monoclonal (1:1000) (Clone 3F10, Sigma-Aldrich) was used in this study. Antibody Identifier: 12013819001 RRID: AB_390917 |
|---|---|
| Validation | The Antibody validation statement can be found on the website: https://www.sigmaaldrich.com/DE/en/product/roche/12013819001 |

## Eukaryotic cell lines

Policy information about cell lines and Sex and Gender in Research

| Cell line source(s) | HEK-tsA201 cells (labeled HEK293T cells in the manuscript) were purchased from Sigma-Aldrich (ECACC Cat#96121229) and HEK293AD cells were purchased from Biocat, cat. no. AD-100-GVO-CB. |
|---|---|
| Authentication | Cell lines have not been authenticated after purchase. Early passages were consistently used. Cells were undergoing verification of cell morphology during cell-culture routine. |
| Mycoplasma contamination | Cells were routinely tested for mycoplasma contamination using MycoAlertTM mycoplasma Detection Kit from Lonza (Basel, Switzerland). Cells have been tested negative from contamination. |
| Commonly misidentified lines (See ICLAC register) | No commonly misidentified cell lines were used in the study. |

## Plants

| Seed stocks | Not relevant for this study. |
|---|---|
| Novel plant genotypes | Not relevant for this study. |
| Authentication | Not relevant for this study. |

