## [Peer Review file · Nature]

Ligand-specific activation trajectories dictate GPCR signalling in cells

Corresponding Author: Professor Andreas Bock

Version 0:

Reviewer comments:

Referee #1

(Remarks to the Author)

In their study by Thomas et al, the authors describe the conformational landscape of the M2-muscarnic acetylcholine receptor (M2AChR) upon agonist and G protein binding. The authors utilize unnatural amino acid incorporation of environmentally sensitive fluorescent probes on the extracellular face (extracellular loops) of the receptor on intact cells. A subset of these biorthogonal, fluorophore-labeled receptors displayed significance changes in their fluorescence spectra, evaluated in this case as alterations in fluorescence emission, upon agonist binding or G protein coupling. With these biosensors the authors are able to assess agonist efficacy in the absence or presence of G proteins. Subtle differences in specific subsets of biosensor fluorescence in response to full and partial agonist with or without G proteins have allowed the authors to define four distinct receptor conformations. These data are in good agreement with previous analysis of GPCR conformation using NMR, DEER and fluorescence spectroscopy in excellulo studies with purified, reconstituted protein, but in this case the studies are performed in intact cells.

This is a very interesting study using a novel approach to measure GPCR in living cells. For the most part many of the conformations observed in this study have been observed using in vitro studies with purified components in a phospholipid bilayer. In some sense the data reported in this study validates the in vitro data, rather than visa-versa, as the conformational changes occurring here are in a more native cellular environment. These data should be of significant value to the field but will eventually need to be tested in other GPCRs, at least with respect to the proposed four active state conformations.

There are only a couple of comments.

The ACh dose-response curves presented in the extended data (ext data fig 5) are very nice but should really be extended to the other agonists (iperoxo, Pilo and arecoline). In addition, the differential response of the various biosensors to agonist stimulation in the presence overexpressed Gao (mutant) is very interesting and consistent with the notion that the nucleotide-free G protein stabilizes the active conformation in the absence of agonist. Of the biosensors that do respond to agonist stimulation it would be of great interest to know if the dose response is comparable to G protein recruitment BRET data. Of minor note is that that the dose-response curve should be replotted in a more reasonable choice of x-axis. Perhaps the x-axis should be from 10^{-9} to 10^{-3} so that we can see any differences, if any, in the EC50s.

The kinetic data are very nice and useful to define the distinct conformational states but need some statistical analysis. None-the-less the rates biosensor responses to agonists using endogenous levels of Galpha are comparable to the previously reported G protein recruitment/activation data and much slower than intracellularly-labeled receptor biosensors (50 ms). In reading the manuscript it seems that the authors argue that extracellular biosensors reflect more the G protein-stabilized conformation since the kinetics are similar to G protein recruitment data. Thus it would be important to know if the receptor biosensors that do respond to agonist in the presence of overexpressed Gao (mutant), has fast or slow kinetics.

There are some typographic and grammatical errors but should be picked up in future editorial processes.

Referee #2

(Remarks to the Author)

Thomas et al develop a fluorescence biosensor assay to monitor activation dependent conformational changes of the M2R in live cells. Using their panel of biosensors, they can detect differences in the conformational dynamics of the extracellular surface between agonists of different efficacies. Using a panel of agonists, they found not only absolute differences in the conformational ensemble occupied by the receptor, but differences in the equilibrium between different activation states induced by a given ligand (termed low and high efficacy states). They then delve into the functional implications of this, finding that the activation states induced by a given ligand directly influence the receptor's preference (or lack thereof) for specific subtypes of G protein.

The novelty of the technology, as well as the contribution of the results to the overarching controversy between ligand efficacy and ligand bias, merit publication in Nature pending revision. The most important revision would be more precise determination of what conformational changes the assay system is reporting on. How do the biosensors respond to ECL binding allosteric modulators? Is there mutational validation that the authors could perform to test whether their assay is reporting on closing of the extracellular lid over the ligand binding pocket?

Sensor Characterization (Figure 1, Supporting Data F1, Extended Data F1-5) – Minor Comments

I appreciate the systematic nature of the assay development that the authors underwent, as well as the thorough functional characterization of M2R biosensors. The authors point out a valid correlation between sensor expression level and FRET response in Extended Data Figure 3, which helps with the data interpretation. However, I think that it would also be valid to point out the correlation between FRET response in Extended Data Figure 3 and the sensor potency (EC₅₀) in Extended Data Figure 5.

It is also difficult to make out the structural changes from the inactive to active states in Figure 1E as presented.

Figure 2 – Minor Comment

The authors perform a paired comparison between a control ligand, Ach, and three new ligands with their panel of biosensors. To facilitate data interpretation from the reader, I would suggest pairing individual data points via line segments in the data presentation.

I also think it is important, given the paired data comparison, to demonstrate that the order of ligand addition in the assay does not influence the result.

Figure 3 – Minor Comment

The authors compare the response of their biosensor panel from endogenous G proteins to a “nucleotide free” G protein mimic. I think it would be more informative to compare between ligands in panels C, F and I rather than comparing to the null hypothesis. I also think it would be informative to show individual data points on the bar graphs in C, F and I, given a lack in consensus between representative and mean data in certain panels (B and C).

Figure 4 – Minor Comment

The authors map the time scales of the conformational changes observed in Figures 1-3 onto a plane, capturing differences in the activation state of M2R depending on ligand efficacy. I think that this figure would be more powerful if the authors presented a conformational model (rather than the planar images) based upon the solvatochromic properties of the cyanine dyes and the structural differences observed in the extracellular domain between inactive and active state structures of M2R or other GPCRs.

Figure 5 – Minor Comment

The authors correlate their observations of ligand efficacy-dependent conformational changes in the extracellular domain of the receptor with changes in G protein selectivity, wherein lower efficacy ligands appear to preferentially activate G protein isoforms. To make this correlation more effective, I think that the authors should explicitly include their model presented in Figure 4 with the signaling data presented in Figure 5. I think this data provides some important insight into the controversy in the field surrounding ligand efficacy versus ligand bias and merits some further elaboration in the discussion.

There also appears to be some lack of consensus between panel C and panel F, specifically for the Gi3 and G15 data points. I would recommend some other form of data presentation to alleviate this.

Referee #3

(Remarks to the Author)

In this manuscript, Thomas et al., developed biosensors for studying GPCR signaling in intact cells. They focused on the M2 muscarinic acetylcholine receptor (M2R) and utilized these newly created M2R biosensors to study the receptor's response to stimulation in living cells.

While I am not an expert in GPCR biology, technically this manuscript nicely illustrates the power of genetic code expansion and click chemistry. By combining these advanced technologies, the authors were able to design a set of minimally invasive fluorescent M2R biosensors. To create these sensors, genetic code expansion was used to incorporate noncanonical amino acids (ncAAs) site-specifically into the extracellular domain of M2R. Once incorporated, the reactive ncAAs were labelled with a cell-impermeable tetrazine-conjugated fluorescent dye using a bioorthogonal click chemistry reaction. This approach allowed the authors to target various extracellularly accessible residues and study GPCR conformational changes by evaluating fluorescence intensity variations.

I see significant potential and value in using these technologies to address challenges in the studies of GPCRs. However, I

also see some shortcomings that should be addressed prior to publication of this manuscript:

1. This manuscript relies on genetic code expansion and click chemistry to fluorescently label different M2R variants. While these technologies have been previously used by the authors and others for fluorescent protein labelling in various applications, including studies of not only GPCRs but also a wide range of other proteins, they are still not widely adopted. For this reason, it is particularly important to provide a comprehensive description of these techniques in both the main text and the methods section. However, my impression is that the manuscript devotes insufficient attention to these aspects. Furthermore, the current version contains numerous references to "as described previously" in both the main text and methods section, which I believe is inappropriate. I will outline specific examples below:
 - a. It is impressive that the authors tested 72 extracellularly accessible residues for ncAA incorporation in M2R. However, the manuscript should provide a more detailed explanation of what is required to create such biosensors. For example, the authors do not seem to describe the plasmid and genetic elements which are required to incorporate nCAAs. This information is essential for readers who are not familiar with genetic code expansion but are interested in developing or utilizing these biosensors in future studies. It might help to include additional references that deal with ncAA incorporation and click chemistry fluorescent labelling in living cells to provide further context and guidance for interested researchers.
 - b. Similarly, the authors should describe how the ncAA was chosen. Are there any limitations to keep in mind? For example, the click reaction between TCO*-Lysine and tetrazines used in this study is known to potentially undergo a click-to-release mechanism. Discussing such limitations and the rationale behind the choice of ncAA would provide valuable insights for readers, particularly those considering using this approach in their own research.
 - c. The authors primarily used Cy3 and they mention that Cy3 is environmentally sensitive. Is there any published work to back this up? Also, what about other fluorophores in addition to Cy3 and Cy5? Could they be used?
 - d. For proper biosensor variants characterization, click labelled controls without ncAA should also be included.
 - e. According to the methods section, various M2R construct variants were generated, including full-length M2R protein (without additional tags) and variants with HA-tag and GFP reporter. However, it is not entirely clear which variant was used for each specific purpose. This should be better described and clearly indicated in the text and in the corresponding figure legends.
 - f. The Biosensor Construction section mentions the cloning of a signaling peptide, but it is not clear why this step is necessary and if it was used in all of the constructs.
 - g. What makes me wonder is the expression level of these biosensors. If I understand correctly, the biosensor variants will compete with endogenous, unlabeled GPCRs. If the expression levels of the different biosensor variants are not consistent, this could potentially influence the observed responses upon stimulation. This issue should be discussed and it would help to provide some kind of comparative analysis of the expression levels of different variants.
2. Not only related to the genetic code expansion and click chemistry-based labelling, the manuscript and methods section should be carefully reviewed to replace "as previously described" references with more detailed descriptions.
3. In some of their figure legends, the authors use a wording "3 independent days of experiments". At least to me, this sounds confusing.
4. In some of their figure legends, the authors mention individual wavelengths for excitation and emission of Cy3. I do not understand why. It is enough to describe this in the methods sections. Even there, one needs to describe the laser line that was used for the excitation and the individual filter cubes (or spectral range) that were used to collect the emitted light.
5. More generally, it would be helpful to provide an outlook on the broader applicability of this approach. What would be required to create similar biosensors for other GPCRs? Additionally, how might these biosensors be adapted for use in other biological systems or cell types? A discussion of these possibilities would greatly enhance the potential impact and generalizability of the study.
6. What are the main advantages and limitations of this approach, and how will the findings of this study help advance further research in the field? Furthermore, how can researchers best utilize the different biosensor variants developed here? What is their value beyond this specific study, and how might they contribute to future investigations?

Version 1:

Reviewer comments:

Referee #1

(Remarks to the Author)

The revised manuscript by Thomas et al. represents a significant improvement to an already interesting story. The authors have responded mostly well to this and other reviewers' comments in the form of stylistic improvements as well as providing additional experimental data. However, there are some old and new issues that need to be addressed by the authors in the form of nuanced statements and better clarification (see below).

Major comments:

The following comments are listed in order of their appearance in the manuscript, rather than in order of their importance.

1. There is a question of whether there is a difference in labeling efficiency (of the fluorescent moiety) with different agonists. Is there a difference in fluorophore access to the non-canonical tRNA-encoded residue by various conformations stabilized by agonists or antagonist? The authors did show that ligands such as NMS did not interfere directly with the Cy3, either conformationally or through altering the fluorophore's spectral properties. The authors do report differences in labeling efficiency between various positions, which could be directly related to receptor conformation and possibly sensitivity to agonists of varying efficacies. For example for positions 181, 188 and 419, where labeling efficiencies are lower than other residues, could be restricted unless in the active state. Under basal conditions the GPCRs do drift into the active state.

2. The authors did state that Cy3 is environmentally sensitive but really didn't describe how and why. It would be useful for readers to understand that solvent exposure is likely the key player in fluorescence quenching. For example, movement of the fluorescent moiety into a more solvent exposed environment as part of the change in conformation would appear as a decrease in fluorescence emission. So, biosensors that display agonist-stimulated fluorescence decreases likely have their fluorophore, in their basal states, lying in a relatively hydrophobic environment like the vestibule itself. Thus, conformational changes in the ECLs following agonist addition move the fluorophore directly into solvent. Similarly, agonist-promoted increases would put the fluorophore into a more hydrophobic environment.

3. On p. 9 the authors describe the conformational states stabilized by agonists compared to G proteins. Herein are a couple details that need to be clarified. The G203T mutant indeed has poor affinity for nucleotides, which is why it is useful for stabilizing RG complexes for structural work. Hence it is understood that the G203T mutant stabilizes an active conformation similar to agonists. Several biophysical studies have suggested that both G protein and/or agonist addition alter the distribution of conformational states, favoring those of the active states. In this manner agonist and nucleotide-free G proteins act cooperatively to stabilize the fully active state better than G proteins or agonist alone. The question is whether failure of agonists to induce changes in fluorescence with some of the sensors when G302T is present is actually due to the fact that G203T itself adopted something close to the fully active conformation. Furthermore, these ensemble measurements really can't resolve multiple active conformations. This would require single molecule spectroscopy work (also with FRET or Trp quenching) where distributions in fluorescence lifetimes can be measured.

4. In the manuscript the authors refer to the non-G protein-coupled receptor as the 'basal condition' in contrast to WT or mutant Galpha overexpression. This is confusing to readers. Basal conditions should really only refer to non-agonist/inverse agonist treated receptor activity. As stated it is not always clear whether the authors are referring to no G proteins or no agonist, or both. Comparing receptor activity in the absence or presence of G protein should be referred to as the uncoupled or coupled-state, in this context.

5. On p. 12 the authors describe the kinetics of fluorescence changes due to agonist treatment on biosensors 181 and 188 and how they are slower than with other biosensors. Is this due to steric effects of the bulky fluorophores on agonist entry into the orthosteric site? Have the authors tried looking at [3H]NMS on-rates to 181 and 188 compared to other biosensors? This is important since the authors on p. 13 are arguing that this is naturally a slower conformational step that follows other conformational changes, indicated by other biosensors.

Minor comments:

The description of the actions of PTX in p. 10 is a little misleading. The major effects of PTX-mediated ADP ribosylation of the C-terminus of Galpha_{i/o} prevents coupling through sterically occluding access to the receptor core. This is why it affects nucleotide exchange, especially for the use of PTX in this study.

The first use of the term non-canonical amino acid (ncAA) in the introduction should also include 'as known as unnatural amino acid incorporation' or something of the sort.

Referee #2

(Remarks to the Author)

The authors' reluctance to provide a more structural interpretation of biosensor readout are fair, and I think that this viewpoint is clear in the data interpretation throughout the manuscript. However, I think that both sets of experimental data provided in the response letter (Reviewer Figs 2 and 3) provide important context for the study. The demonstration with LY2119620 in Reviewer Figure 2 is important for the significance of this study. For this technology to be transformative for GPCR drug discovery down the line, it is paramount for it to provide some insight as to how drugs influence the dynamics of the extracellular domain of the receptor, regardless of a structural interpretation of the readout. To this end, demonstrating how these biosensors report on an allosteric compound that directly interacts with the extracellular loops serves this purpose. Furthermore, the experimental data in Reviewer Fig. 3 (and interpretation provided by the authors) also provide essential physical context for the biosensor readout, especially given that the biosensor experiments are performed at (near) saturating concentrations that appear to still be able to activate G proteins downstream of the compromised biosensor. I would be curious as to how the chosen mutation influences G protein activation downstream of a non-biosensor version of M2R, which is relevant to my next point.

I maintain that there is a correlation between the response of the Gi13 FRET sensor in Ext. Fig. 3, and the potency of the biosensor response to acetylcholine in Ext. Fig. 5. For clarity, here is a plot of this comparison, pulled from the source data provided by the authors.

This correlation suggests that the more potent the biosensor fluorescence change is to the reference ligand, the larger the

impact there is on G protein activation. This correlation between biosensor readout and function implies that the biosensors could be influencing the activation states occupied by the receptor, instead of passively reporting on the conformational states occupied by the receptor. To this end, there is also a convincing trend between another functional readout, the TRUPATH EC50 of the biosensor for acetylcholine (Ext. Fig. 6), and the ability of a sensor to differentiate between high and low efficacy signaling complexes (Fig. 3).

It is important for the reader to understand these implications, especially as this technology is translated to other receptor systems.

I will also note that in performing these analyses, I found discrepancies between the reported EC50 values in the legend of Ext. Fig. 5 and the source data file. There are also discrepancies between Fig. 3i and the source data file. I would advise a careful review of the source data file for any other discrepancies.

Referee #3

(Remarks to the Author)

I appreciate the effort the authors have put into revising their manuscript. They now provide a detailed description of the underlying GCE technology and bioorthogonal labelling, even more detailed than expected. Overall, I believe this will make the methodology more broadly applicable to other researchers. In this regard, I also recommend that the plasmids be made more freely available to the community, since (if feasible) this would be of great benefit.

I have a few additional minor comments on this version:

1. Choice of ncAA: The authors write: "which is highly reactive with both tetrazine and methyl-tetrazine dye derivatives yet sufficiently stable over the duration of the experiment⁴¹." Please specify in that section what the duration of these experiments was, as this remains a critical question when working with TCO(2-en)-Lys.
2. Control without ncAA incorporation: The authors write: "We fully acknowledge the rationale for including a labelled control without ncAA incorporation. In the course of our thorough biosensor characterisation, we never observed any detectable labelling in cells lacking a properly incorporated ncAA (so that we could not image these samples). This corroborates the specificity of our labelling approach." I still believe it would be useful to show the controls without ncAAs. While I understand that doing so may be more difficult at this stage, the authors should at least clarify in the respective section or figure legends that these control experiments were performed. This clarification would also be important for other researchers intending to apply this technology to other GPCR types.
3. "Up to" phrasing: On p. 5, the authors write: "we have previously demonstrated up to quantitative labelling..." I assume that "up to" is redundant here, or else the sentence needs to be reformulated.
4. Plasmid ratios: In the Methods section, when describing different plasmid ratios (e.g., 1:1:1, 10:10:1, 5:5:1), does the total amount of DNA remain the same as defined at the beginning of that Methods section? Please clarify.
5. Plate format for DNA transfection: The authors mention transfection with 4 µg of total DNA, but it is not clear for which plate format this was done. For example, for determining the cell-surface expression of M2R biosensors with an ELISA, the cells were supplemented with TCO*K as described above and transfected with 10 µl Lipofectamine 2000, using a cDNA ratio of 1:1 (SP-HA-M2RXXXTAG or SP-HA-M2R-wt, MbPyIRSAF/4xtRNAM15) of 4 µg total plasmid 24 h after seeding. Please specify the plate format in this context.
6. Ext. Data Fig. 1 legend: Are those epifluorescence images from living cells? I assume so, since the manuscript otherwise seems to show only live-cell images. If not, please specify in the figure legends and the Methods section.
7. TAMRA-tetrazine source: In the Methods section, TAMRA-tetrazine is also mentioned. Please specify its source.

Version 2:

Reviewer comments:

Referee #2

(Remarks to the Author)

Responses to the reviewers' comments

We would like to thank all three Reviewers for their consistently positive and encouraging comments, which have allowed us to significantly improve the manuscript. Please find our specific answers to all individual comments made by the three Reviewers below.

Referee #1 (Remarks to the Author):

In their study by Thomas et al, the authors describe the conformational landscape of the M2-muscarnic acetylcholine receptor (M2AChR) upon agonist and G protein binding. The authors utilize unnatural amino acid incorporation of environmentally sensitive fluorescent probes on the extracellular face (extracellular loops) of the receptor on intact cells. A subset of these biorthogonal, fluorophore-labeled receptors displayed significance changes in their fluorescence spectra, evaluated in this case as alterations in fluorescence emission, upon agonist binding or G protein coupling. With these biosensors the authors are able to assess agonist efficacy in the absence or presence of G proteins. Subtle differences in specific subsets of biosensor fluorescence in response to full and partial agonist with or without G proteins have allowed the authors to define four distinct receptor conformations. These data are in good agreement with previous analysis of GPCR conformation using NMR, DEER and fluorescence spectroscopy in excellulo studies with purified, reconstituted protein, but in this case the studies are performed in intact cells.

This is a very interesting study using a novel approach to measure GPCR in living cells. For the most part many of the conformations observed in this study have been observed using in vitro studies with purified components in a phospholipid bilayer. In some sense the data reported in this study validates the in vitro data, rather than visa-versa, as the conformational changes occurring here are in a more native cellular environment. These data should be of significant value to the field but will eventually need to be tested in other GPCRs, at least with respect to the proposed four active state conformations. There are only a couple of comments.

Authors' reply: We are grateful for this very positive assessment and the elegant and accurate summary of our manuscript. We fully agree with the Reviewer that – in the future – our newly developed biosensing method needs to be tested for other GPCRs to establish whether several distinct active state conformations are similarly stabilized by

different agonists for all other GPCRs. Obviously, such an endeavor is far beyond the scope of this manuscript and will require a concerted effort of several groups in the field.

The ACh dose-response curves presented in the extended data (ext data fig 5) are very nice but should really be extended to the other agonists (iperoxo, Pilo and arecoline). In addition, the differential response of the various biosensors to agonist stimulation in the presence overexpressed Gao (mutant) is very interesting and consistent with the notion that the nucleotide-free G protein stabilizes the active conformation in the absence of agonist. Of the biosensors that do respond to agonist stimulation it would be of great interest to know if the dose response is comparable to G protein recruitment BRET data.

Authors' reply: We thank the Reviewer for appreciating our concentration-response curves reported in the Extended Data Figure 5 and we fully agree that investigating whether the agonists' potencies and efficacies are consistent across all conformational biosensors for all ligands is an important addition to strengthen our study. In the revised manuscript, we now provide G_{OA} -protein activation data for all 7 biosensors and all 4 agonists (c.f. new **Extended Data Fig. 6**).

The curves were generated using the (more) generally used and accepted TRUPATH assay rather than measuring fluorescence emission changes at different ligand concentration in single-cell microscopy, as we have reported with acetylcholine in the first version of the manuscript. The TRUPATH approach has two advantages: In the first place, we could collect a systematic and complete dataset, which is not accessible by measuring fluorescence changes with our biosensors, as – for instance – iperoxo did not promote fluorescence changes at position 181 (Figure 2) and overexpression of the G-protein mutant abolished agonist-promoted fluorescence changes for all agonists at position 175 (Figure 3). Second, the TRUPATH assay offers a much better signal-to-noise ratio and thus more reliable results. As can be observed in Figure 1d of the revised manuscript, the maximal change in fluorescence intensity at supramaximal concentrations of the full agonist ACh ranges from 5-15 %, depending on the biosensor. Under these premises, it is practically impossible to generate concentration-response relationships for all partial agonists under basal conditions and with G-protein overexpression (which reduces the measurement window for most positions even further) with sufficient accuracy and precision. The signal-to-noise ratio – especially at low agonist concentrations – becomes a limiting factor. Finally, we would like to remark that generating concentration-response curves by measuring fluorescence changes on single cells is technically very cumbersome and is practically feasible within a reasonable investment of time and effort only for small datasets.

The new data, which are included in the revised manuscript as **Extended Data Fig 6** and **Supplementary Table 2**, enable us to draw several major conclusions that significantly strengthen and broaden the interpretation of our findings:

- Every agonist stimulates G-protein coupling by every biosensor in a concentration-dependent manner: This indicates that all biosensors are *bona-fide* functional receptors that couple to and activate G proteins.
- The rank order of both agonist efficacies and potencies at all biosensors remains exactly the same as at wild-type receptors: This demonstrates that results retrieved from all conformational biosensors can be precisely projected to wild-type receptors.
- Six out of seven biosensors essentially behave as wild-type receptors in G-protein activation assays. Only the M₂R¹⁸¹ biosensor is impaired for all agonists. This demonstrates that our technology only minimally perturbs receptor function (please note that the TRUPATH experiments were done by transiently overexpressing 5 different plasmids which can result in variation of receptor and G-protein expression and of their stoichiometry. This is why we only consider a biosensor as functionally impaired when agonist potencies drop by a factor of more than 1 log unit).

NEW Extended Data Figure 6

Extended Data Figure 6: Ligand-dependent, M₂R biosensor-mediated activation of G_{αA} proteins. Concentration-response curves of agonist-induced G-protein activation of M₂R wildtype and all M₂R biosensors. Potencies and efficacies of Ixo (green), ACh (grey), Arec (purple), or Pilo (pink) towards G_{αA} proteins determined by the TRUPATH BRET-based assay upon activation of the indicated M₂R construct, i.e. *SP-M₂R-wt* or *SP-M₂R^{XXXXTAG}*. Concentration-dependent BRET changes (ΔBRET) were normalised to buffer (set to 0 %) and the maximal response of ACh (set to 100%), indicated by a dotted line. Shown are the means ± sem of 3-4 independent experiments (see Supp. Table 2).

Of minor note is that that the dose-response curve should be replotted in a more reasonable choice of x-axis. Perhaps the x-axis should be from 10^{-9} to 10^{-3} so that we can see any differences, if any, in the EC50s.

Authors' reply: We thank the Reviewer for this comment. We have replotted the data accordingly (**cf. revised Extended Data Figure 5**).

Revised Extended Data Figure 5

Extended Data Figure 5: M₂R biosensors react to ACh with similar sensitivities. (a-g) Representative traces of concentration-dependent changes of fluorescence intensity ($\Delta F/F_0$) of a single cell expressing the indicated M₂R biosensor (*SP-M₂R^{XXXTAG}*), stimulated with increasing concentrations of ACh. The addition of increasing ACh concentrations is indicated as log (M) and by a grey color gradient. (h) Concentration-effect-curves of normalised, mean ACh-mediated fluorescence intensity changes in response to increasing concentrations of ACh. ACh potencies ($\log EC_{50} \pm \text{sem}$): T84, -6.2 ± 0.08 ; E175, -6.4 ± 0.03 ; F181, -5.6 ± 0.05 ; F188, -5.4 ± 0.06 ; A414, -5.8 ± 0.02 ; P415, -6.2 ± 0.02 ; N419, -5.2 ± 0.06 . $n = 10$ (T84), $n = 19$ (E175), $n = 21$ (F181), $n = 20$, $n = 20$ (188), $n = 29$ (A414), $n = 15$ (P415), $n = 13$ (N419). Data obtained from 3 independent experiments.

The kinetic data are very nice and useful to define the distinct conformational states but need some statistical analysis. None-the-less the rates biosensor responses to agonists using endogenous levels of Galpha are comparable to the previously reported G protein recruitment/activation data and much slower than intracellularly-labeled receptor biosensors (50 ms). In reading the manuscript it seems that the authors argue that extracellular biosensors reflect more the G protein-stabilized conformation since the kinetics are similar to G protein recruitment data. Thus it would be important to know if the receptor biosensors that do respond to agonist in the presence of overexpressed Gao (mutant), has fast or slow kinetics.

Authors' reply: We are very grateful for the Reviewer's comment. In the revised manuscript, we now include kinetic data of agonist-mediated fluorescence changes of all biosensors under control conditions and upon $G\alpha_{oA}(G203T)$ overexpression (cf. revised **Extended Data Figure 9** and new **Supplementary Table 3** (former Supplementary Table 2)). Statistical testing of the kinetic data, as requested by the Reviewer, clearly demonstrates that the agonist-mediated biosensor responses are significantly slower upon overexpression of $G\alpha_{oA}(G203T)$. These data strengthen our argument that the biosensor responses correlate strongly with timescales previously observed for G-protein activation.

Revised Extended Data Figure 9

Extended Data Figure 9: Time-resolved dynamics of M_2R/G protein-signalling complexes. Comparison of apparent on-rates (Tau in s) obtained from six biosensors ($SP-M_2R^{XXXXTAG}$) after activation with the indicated ligands 1 mM ACh (a), 100 μ M Ixo (b), 1 mM Arc (c), and 10 mM Pilo (d) at endogenous G-protein levels (wt) and after $G\alpha_{oA}(G203T)$ overexpression. Bars indicate means \pm sem of at least 3 independent experiments, each data point represents a single cell (see Supp. Table 3). **** $p < 0.0001$, *** $p < 0.001$, ** $p < 0.01$, * $p < 0.05$, according to a two-tailed unpaired t-test with Welch's correction. The biosensors are classified into statistically different groups, according to a one-way analysis of variance (ANOVA multiple comparisons test) with Tukey's posthoc test. wt, wildtype; GoA, $G\alpha_{oA}(G203T)$; ns, not significantly different. Biosensors that are grouped are not significantly different from each other.

Revised Supplementary Table 3 (former Supp. Table 2)

Supplementary Table 3 (related to Fig. 4 and Ext. Data Fig. 9). Mean Tau-ON (apparent on-rates, s) values of fluorescence intensity changes from M₂R biosensors stimulated with the indicated agonists. N: total number of cells. Exp: independent experiments, SEM: standard error of mean, wt: wildtype (endogenous G-protein level), G α_{oA} : overexpressed G α_{oA} (G203T) mutant.

Agonist	T84 ^{TM2}								E175 ^{ECL2}							
	Ixo		ACh		Arec		Pilo		Ixo		ACh		Arec		Pilo	
	wt	G α_{oA}	wt	G α_{oA}	wt	G α_{oA}	wt	G α_{oA}	wt	G α_{oA}	wt	G α_{oA}	wt	G α_{oA}	wt	G α_{oA}
Mean	1.12	1.06	0.25	0.64	0.34	2.72	2.10	1.14	1.03	NA	0.50	NA	0.27	NA	0.18	NA
SEM	0.11	0.09	0.02	0.15	0.06	0.60	0.18	0.16	0.14	NA	0.02	NA	0.03	NA	0.01	NA
N	25	15	35	14	25	10	39	20	13	11	44	10	9	9	25	11
Exp	7	5	17	7	6	3	7	4	7	3	11	3	4	3	5	3

Agonist	F181 ^{ECL2}								F188 ^{TM5}							
	Ixo		ACh		Arec		Pilo		Ixo		ACh		Arec		Pilo	
	wt	G α_{oA}	wt	G α_{oA}	wt	G α_{oA}	wt	G α_{oA}	wt	G α_{oA}	wt	G α_{oA}	wt	G α_{oA}	wt	G α_{oA}
Mean	NA	18.41	1.33	1.82	1.85	1.96	1.87	1.50	3.21	9.35	2.69	3.20	2.21	4.64	4.73	8.09
SEM	NA	1.48	0.04	0.04	0.16	0.13	0.10	0.32	0.26	0.45	0.11	0.12	0.14	0.45	0.34	0.54
N	NA	29	76	41	13	34	17	20	21	16	96	26	35	13	27	17
Exp	NA	4	9	7	7	4	7	3	5	3	15	4	6	3	6	3

Agonist	A414 ^{ECL3}								P415 ^{ECL3}							
	Ixo		ACh		Arec		Pilo		Ixo		ACh		Arec		Pilo	
	wt	G α_{oA}	wt	G α_{oA}	wt	G α_{oA}	wt	G α_{oA}	wt	G α_{oA}	wt	G α_{oA}	wt	G α_{oA}	wt	G α_{oA}
Mean	0.53	2.21	0.17	0.49	0.87	0.54	0.33	0.68	0.34	0.44	0.16	0.59	0.22	0.68	0.21	0.28
SEM	0.03	0.34	0.01	0.08	0.12	0.08	0.02	0.10	0.03	0.02	0.00	0.08	0.01	0.17	0.01	0.06
N	38	12	33	31	26	10	22	16	40	28	56	21	26	19	43	10
Exp	12	4	7	9	11	3	3	3	9	5	9	6	5	5	10	3

There are some typographic and grammatical errors but should be picked up in future editorial processes.

Authors' reply: Thank you for spotting these. We hope to have eliminated all of these typos.

Referee #2 (Remarks to the Author):

Thomas et al develop a fluorescence biosensor assay to monitor activation dependent conformational changes of the M2R in live cells. Using their panel of biosensors, they can detect differences in the conformational dynamics of the extracellular surface between agonists of different efficacies. Using a panel of agonists, they found not only absolute differences in the conformational ensemble occupied by the receptor, but differences in the equilibrium between different activation states induced by a given ligand (termed low and high efficacy states). They then delve into the functional implications of this, finding that the activation states induced by a given ligand directly influence the receptor's preference (or lack thereof) for specific subtypes of G protein. The novelty of the technology, as well as the contribution of the results to the overarching controversy between ligand efficacy and ligand bias, merit publication in Nature pending revision.

Authors' reply: We are very grateful that the Reviewer highlights both the experimental as well as the conceptual novelty of our work and posits that this merits publication in *Nature*.

The most important revision would be more precise determination of what conformational changes the assay system is reporting on.

Authors' reply: We thank the Reviewer for this insightful comment. We agree that it would be fantastic to derive detailed structural information directly from the receptor embedded in the cell membrane of a living cell. While such information can be accessed through biophysical methods such as X-ray crystallography and - to a smaller extent - NMR and DEER using isolated receptors in artificial lipid environments, no technique currently exists that can achieve comparable resolution in live cells. We can confidently state that our approach provides the highest currently achievable resolution for conformational tracking of receptors in live cells. While previous conformational sensors have relied on the fusion of large fluorescent proteins, or in the best case the introduction of oligomeric epitopes (for instance FLAsH tags), we introduce single-residue modifications that allow us to follow individual domains of the receptor. However, it is intrinsic to the nature of our method that it cannot provide direct atomic-level structural insights. This is because the fluorophore is not closely enough attached to the protein backbone, which is obviously the case of NMR probes and even DEER probes, which are installed *via* short spacers. Instead, the post-translational modification of ncAAs – at least currently (see below) – requires a relatively large and flexible spacer between the fluorophore and the protein backbone, making it impossible to draw detailed structural conclusions from the observed fluorescence changes. This also holds true for recently published single-molecule FRET studies by the Kobilka, Blanchard, and Isacoff labs that used similar cyanine-based dyes as labels to investigate GPCR dynamics (e.g. Latorraca et al., *Nat. Chem. Biol.*, 2025, <http://doi.org/10.1038/s41589-025-01895-3>, Zhao et al. *Nature*, 2024, <https://doi.org/10.1038/s41586-024-07295-2>, Kumar et al., *Nature*, 2024, <https://doi.org/10.1038/s41586-024-07327-x>, Kumar et al., *Cell*, 2023, <https://doi.org/10.1016/j.cell.2023.02.028>, Gregorio et al., *Nature*, 2016, <https://doi.org/10.1038/nature22354>).

While the precise structural interpretation of conformational changes requires complementary information from other methods, just as is generally necessary for the smFRET studies mentioned above, the unique power of our approach lies in its ability to provide **direct access to real-time conformational dynamics of GPCR activation in intact cells**. Based on the comparison of the structural changes during M₂R activation (derived from cryo-EM) with the receptor positions whose fluorescent labeling yields biosensors (Figure 1E), we can interpret the observed conformational changes as changes in the **structural dynamics of the extracellular loops**, whereas we refrain from statements that biosensors indicate lid closure. This interpretation is strongly supported by recent elegant work from Scott Prosser's lab (Picard et al., *Nat. Chem. Biol.*, 2024, <https://doi.org/10.1038/s41589-024-01682-6>). Using a combination of NMR and computational methods at the isolated adenosine A₂A receptor *in vitro*, this work strikingly shows that structural dynamics of ECL2 strongly depend on the type of G protein that is coupled to the intracellular part of the receptor. In the revised manuscript, we place our data in the context of this study.

How do the biosensors respond to ECL binding allosteric modulators?

Authors' reply: Intrigued by the Reviewer's question, we set out to test the effects of the positive allosteric modulator (PAM) LY2119620 across all seven M₂R biosensors. Interestingly, we observed a unique conformational fingerprint of LY2119620 that is clearly distinct from all orthosteric agonists tested. These are fascinating and complex new findings which should form the basis of a new study that, because of the wealth of data and the complexity of its interpretation, we are currently exploring in more detail for a separate publication in the future. Clearly, a complete assessment of the molecular principles of allosteric modulation of GPCR complexes in intact cells is far beyond the scope of the present manuscript. However, as we are very grateful for the Reviewer pointing us into this direction, we would like to share a preview of our preliminary data (c.f. Reviewer Figure 2 below).

Reviewer Fig 2: Conformational fingerprint of a positive allosteric modulator. (a) Representative fluorescence intensity changes ($\Delta F/F_0$) recorded in real-time from single HEK293T cells expressing indicated M₂R biosensors superfused with 30 μ M LY2119620 (LY), followed by 1 mM ACh after wash out with buffer. Application of different agonists is indicated with shaded areas in different colors. Non-shaded areas indicate buffer application. (b) Mean fluorescence intensity changes ($\Delta F/F_0$) of all seven biosensors after activation with LY and ACh after wash out with buffer. Positive values indicate an increase in fluorescence; negative values indicate a decrease upon agonist superfusion. Bars indicate means \pm sem of at least 3 independent experiments, each data point represents a single cell. ****p < 0.0001, ***p < 0.001, **p < 0.01, *p < 0.05, according to a two-tailed paired t-test; ns, not significantly different. (c) Radar plot of mean fluorescence intensity changes $\Delta F/F_0$ in response to LY, normalised to $\Delta F/F_0$ of 1 mM ACh in the same cell. The bold line indicates ACh (set to 100%). The direction of $\Delta F/F_0$ is indicated with arrows: increase (up), decrease (down). The change of direction in fluorescence emission for M₂R⁸⁴ and M₂R¹⁷⁵ upon LY stimulation is highlighted in red.

Is there mutational validation that the authors could perform to test whether their assay is reporting on closing of the extracellular lid over the ligand binding pocket?

Authors' reply: We thank the Reviewer for this comment. In the original version of the manuscript, we have hypothesised that the M₂R¹⁷⁵ biosensor may report on the so-called 'lid closure', which signifies activation-related structural rearrangements of the receptor that trap the agonist in its binding pocket. In the M₂R, three tyrosine residues (i.e. Y, Y, Y) form the so-called 'tyrosine lid' and close off the ligand binding pocket from the extracellular space (Kruse, Valant?). Following the Reviewer's suggestion, we reasoned that mutation of one of those tyrosine residues should result in a receptor mutant in which the 'lid closure' is severely hampered.

We introduced a Y426A mutation into the M₂R¹⁷⁵ biosensor to create the M₂R¹⁷⁵-Y426A biosensor. Interestingly, stimulation of this mutant biosensor with ACh did not result in any fluorescence changes, whereas the M₂R¹⁷⁵ biosensor, in which the tyrosine lid is intact, gives a strong response to ACh (see Reviewer Figure 3 below). The lack of ACh-induced fluorescence changes is not due to a lack of receptor activation because ACh-stimulation of the mutated biosensor results in efficacious G_{oA} activation (albeit with markedly reduced potency). These data support the hypothesis that the M₂R¹⁷⁵ biosensor indeed reports on the 'lid closure'.

Although the new data are compatible with the original hypothesis, in the revised manuscript we prefer to just state that the GPCR biosensors (not only the M₂R¹⁷⁵) are detecting structural rearrangements of the extracellular loops. Given the inherent limitations of our biosensor technology to provide direct, atomic-level structural information (see above our response to the Reviewer's first comment), and the fact that the M₂R¹⁷⁵-Y426A biosensor is severely compromised (ACh has a 10,000x fold lower potency), we feel it is more appropriate to not overinterpret the structural resolution of our method. Therefore, we have decided not to include these data into the revised manuscript.

Reviewer Fig. 3: ACh-stimulation does not result in fluorescence changes in the tyrosine-lid mutated biosensor. (a) Representative fluorescence intensity changes ($\Delta F/F_0$) recorded in real-time from single HEK293T cells expressing the M_2R^{175} (blue) or the $M_2R^{175}\text{-Y426A}$ mutant (green) biosensors superfused with 1 mM ACh (shaded area). Non-shaded areas indicate buffer application. (b) ACh-induced mean fluorescence intensity changes ($\Delta F/F_0$) of the M_2R^{175} (blue) or the $M_2R^{175}\text{-Y426A}$ mutant (green) biosensors. Bars indicate means \pm sem of at least 3 independent experiments, each data point represents a single cell. **** $p < 0.0001$, according to an unpaired t-test. The data for the M_2R^{175} biosensor are replotted from Figure 1c and 1d. (c) Concentration-response curves of agonist-induced G-protein activation of the M_2R^{175} (blue) or the $M_2R^{175}\text{-Y426A}$ mutant (green) biosensors as determined by the TRUPATH BRET-based assay. Concentration-dependent BRET changes ($\Delta BRET$) were normalised to buffer (set to 0%) and the maximal response of ACh (set to 100%) at the M_2R^{175} biosensor. Shown are the means \pm sem of 3-4 independent

Sensor Characterization (Figure 1, Supporting Data F1, Extended Data F1-5) – Minor Comments

I appreciate the systematic nature of the assay development that the authors underwent, as well as the thorough functional characterization of M2R biosensors. The authors point out a valid correlation between sensor expression level and FRET response in Extended Data Figure 3, which helps with the data interpretation. However, I think that it would also be valid to point out the correlation between FRET response in Extended Data Figure 3 and the sensor potency (EC_{50}) in Extended Data Figure 5.

Authors' reply: We thank the Reviewer for this suggestion, and we have performed the correlation analysis accordingly. In the Reviewer Figure 3 below we have plotted the biosensor potencies (pEC_{50}) for ACh from Extended Data Figure 5, the absolute $\Delta FRET$ change from the $G\alpha_i$ activation assay as well as the biosensor's cell-surface expression levels relative to wildtype receptors from the original Extended Data Figure 3. From this figure it is evident that there is no correlation between the amplitude of the FRET response in G-protein activation and the ACh potencies in fluorescent biosensors assays. As we transiently overexpress the biosensors, there is considerable receptor reserve so that small differences in receptor expression would not be expected to be visible neither as changes in agonist potencies or efficacies (amplitude of FRET response). In the interest of clarity and conciseness, we do not show this absence of a correlation in the revised manuscript.

Reviewer Fig 4: Functional characterisation of M₂R biosensors. Comparison of (left y-axis) mean ΔFRET[%] and ACh potencies (pEC₅₀ ± sem), and (right y-axis) biosensor cell-surface expression obtained from cell-surface ELISA assays of all seven biosensors and wildtype M₂R (wt). Bars indicate means ± sem, each data point represents a single cell. The dashed bars represent the ACh potencies obtained from concentration-effect-curves of normalised, mean ACh-mediated fluorescence intensity changes in response to increasing concentrations of ACh; the unshaded bars represent the absolute ΔFRET[%] from the G_α₁₃-FRET assay; the dotted bars represent the absorbance normalised to wt (set to 100%) from the cell-surface ELISA assay. Note that a reduced maximal effect on G-protein activation correlates with reduced receptor expression but not with ACh potencies. wt, wildtype.

It is also difficult to make out the structural changes from the inactive to active states in Figure 1E as presented.

Authors' reply: Upon the Reviewer's request, we have revised **Figure 1E**. We are confident that the structural transitions from inactive to active are now presented in a much clearer way.

Revised Figure 1

Figure 1: A novel extracellular, single-color conformational GPCR biosensor panel.

(a) Genetic incorporation of a non-canonical amino acid (TCO*K) and bioorthogonal labelling of M₂R with tetrazine-cyanine 3 (Tet-Cy3) (see **Methods**). Cells expressing biosensors are stimulated by continuous pressurised application of agonist (ACh) or buffer through a manifold tip. (b) Snake-plot of M₂R indicating all positions that were robustly labelled (orange and grey) and showed activation-related fluorescence intensity changes (orange). Numbers indicate residue numbers. Positions that could not be labelled are indicated in white. (c) Representative fluorescence intensity changes ($\Delta F/F_0$) were recorded over time from several individual HEK293T cells expressing M₂R biosensors at the indicated amino acid positions. Cells were superfused with 1 mM ACh. Shaded areas indicate the duration of agonist addition, and unshaded areas indicate agonist washout with buffer. (d) Mean fluorescence intensity changes ($\Delta F/F_0$) of all seven agonist-sensitive biosensors after activation with 1 mM ACh. Positive values indicate an increase in fluorescence, negative values indicate a decrease upon ACh superfusion. Bars indicate means \pm sem of at least 4 independent experiments, each data point represents a single cell. (e) Top-view on the X-ray crystal structure of the active M₂R (blue, PDB: 4MQS). The positions of incorporated TCO*K to yield GPCR biosensors are color-coded according to the gradient, representing the mean fluorescence intensity changes ($\Delta F/F_0$). For comparison, the X-ray crystal structure of the inactive M₂R (PDB: 3UON) is shown in grey. Roman numerals indicate the number of the transmembrane helix. Constructs used: *SP-M₂R*^{XXXTAG}

Figure 2 – Minor Comment

The authors perform a paired comparison between a control ligand, Ach, and three new ligands with their panel of biosensors. To facilitate data interpretation from the reader, I would suggest pairing individual data points via line segments in the data presentation. I also think it is important, given the paired data comparison, to demonstrate that the order of ligand addition in the assay does not influence the result.

Authors' reply: We thank the Reviewer for these valid suggestions. To highlight such paired data comparisons, in the revised manuscript we have improved data presentation in Figure 2 which now displays line segments connecting individual data points (c.f. revised Figure 2).

Revised Figure 2

Figure 2: The M₂R biosensor panel uncovers ligand-specific conformational fingerprints in intact cells. (a, d, g) Representative fluorescence intensity changes ($\Delta F/F_0$) recorded in real-time from single HEK293T cells expressing indicated M₂R biosensors superfused with (a) 100 μ M iperoxo (Ixo), (d) 1 mM arecoline (Arec), (g) 10 mM pilocarpine (Pilo), followed by 1 mM ACh after wash out with buffer, respectively. Application of different agonists is indicated with shaded areas in different colors. Non-shaded areas indicate buffer application. (b, e, h) Mean fluorescence intensity changes ($\Delta F/F_0$) of all seven biosensors after activation with the indicated ligand (b) Ixo (green), (e) Arec (purple), (h) Pilo (pink), ACh (grey) after wash out with buffer, respectively. Positive values indicate an increase in fluorescence, negative values indicate a decrease upon agonist superfusion. Connected data points represent data retrieved from the same cell. Bars indicate means \pm sem of at least 3 independent experiments, each data point represents a single cell (see Supp Table 1). ****p < 0.0001, ***p < 0.001, **p < 0.01, according to a two-tailed paired t-test; ns, not significantly different. (c, f, i) Radar plots of mean fluorescence intensity changes $\Delta F/F_0$ in response to (c) Ixo (green), (f) Arec (purple), and (i) Pilo (pink), normalised to $\Delta F/F_0$ of 1 mM ACh in the same cell. The bold line indicates ACh (set to 100%). The direction of $\Delta F/F_0$ is indicated with arrows: increase (up), decrease (down). (f) The change of direction in fluorescence emission for M₂R⁸⁴ and M₂R¹⁷⁵ upon arecoline stimulation is highlighted in red. Constructs used: SP-M₂R^{XXXTAG}.

Moreover, to demonstrate that the reported agonist efficacies are independent of the order of ligand addition, we have performed new experiments for the critical biosensors M₂R¹⁷⁵ and M₂R¹⁸¹ to which we added all four agonists in reverse order than the one that is depicted in Figure 2. These experiments show that the relative efficacies (% of ACh) are indeed independent of the order of agonist. For instance, iperoxo and pilocarpine result in significantly larger fluorescence changes than ACh at M₂R¹⁷⁵ and M₂R¹⁸¹, respectively, also when ACh was added first to the biosensors. This demonstrates that there is minimal to no desensitization of the GPCR biosensors within the time of the experiment. This is further supported by the constant amplitudes after repeated ACh stimulation. We are grateful for the Reviewer's suggestion and have included this important control data set as **Supplementary Figure 3** in the revised version of the manuscript.

NEW Supplementary Figure 3

Supplementary Figure 3: Agonist efficacies at biosensors are independent of the order of ligand addition. Representative fluorescence intensity changes ($\Delta F/F_0$) recorded in real-time from single HEK293T cells expressing indicated M₂R biosensors (SP-M₂R^{XXXTAG}) superfused with 1mM ACh, followed by (a) 100 μ M iperoxo (Ixo), (b) 1 mM ACh, (c) 1 mM arecoline (Arec), or (d) 10 mM pilocarpine (Pilo), after washout with buffer, respectively. Application of different agonists is indicated with shaded areas in different colours. Non-shaded areas indicate buffer application. Number of individual cells (n, for Ixo, ACh, Arec, Pilo): E175 (n = 29, 54, 15, 22); F181 (n = 20, 68, 22, 14). Data obtained from at least 3 independent experiments.

Figure 3 – Minor Comment

The authors compare the response of their biosensor panel from endogenous G proteins to a “nucleotide free” G protein mimic. I think it would be more informative to compare between ligands in panels C, F and I rather than comparing to the null hypothesis.

Authors’ reply: We thank the Reviewer for the insightful comment regarding the comparison strategy used in panels C, F, and I of Figure 3. In our original figure, we chose to highlight the effect of $G_{\alpha_{OA}}$ overexpression on the agonist-induced, maximal fluorescence amplitudes of the biosensors in comparison to the endogenous G protein (i.e. ‘wild-type’) condition. We decided on this approach to uncover which sensor of the biosensor panel reports primarily on a high-efficacy receptor/G-protein complex or on a low-efficacy receptor/G-protein complex. This comparison allowed us to stratify the different biosensors into four groups which formed, together with the kinetic analysis, the basis for Figure 4. We believe this comparison is essential and we have decided to keep this analysis (revised Figure 3) in the revised version of the manuscript.

The Reviewer suggests to compare the fluorescence changes **between the different agonists** upon $G_{\alpha_{OA}}$ overexpression vs. wildtype conditions. This is indeed interesting as $G_{\alpha_{OA}}$ overexpression may alter the efficacy differences between the four agonists, if $G_{\alpha_{OA}}$ changes the basal equilibrium of different receptor states. We have, therefore, performed this comparison as suggested by the Reviewer. Interestingly, we obtained a multidimensional picture: at some biosensors, the efficacy pattern between ligands remains unchanged, at some biosensors it is indeed altered, and some biosensors cannot discriminate between the agonists’ efficacies anymore. Therefore, this comparison provides **direct evidence that the equilibrium between the different receptor/G-protein complexes is altered** due to $G_{\alpha_{OA}}$ overexpression.

We are very grateful for the Reviewer’s suggestion that has further strengthened our conclusion on conformational equilibria of GPCRs in intact cells. We have included the respective analysis as **Supplementary Figure 5** in the revised version of the manuscript.

I also think it would be informative to show individual data points on the bar graphs in C, F and I, given a lack in consensus between representative and mean data in certain panels (B and C).

Authors’ reply: Thank you for pointing this out. In the revised manuscript we now show the individual data points in the bar graphs in **Figure 3**. We have also replaced some representative traces

Revised Figure 3

Figure 3: M₂ receptor activation results in an equilibrium of distinct M₂R/G protein-signalling complexes in intact cells. Comparison of ligand-promoted fluorescence intensity changes ($\Delta F/F_0$) of the M₂R biosensors labelled with Tet-Cy3 at the indicated positions and stimulated with 1 mM ACh (grey), 100 μ M Ixo (green), 1 mM Arec (purple), or 10 mM Pilo (pink) at endogenous G-protein levels or after $G\alpha_{oA}(G203T)$ overexpression. (a,b,d,e,g,h) Representative traces of $\Delta F/F_0$ of single cells expressing the indicated M₂R biosensors at endogenous G-protein levels (left) or after $G\alpha_{oA}(G203T)$ overexpression (right). Shaded areas indicate the duration of agonist superfusion, unshaded areas represent buffer application. (c,f,i) Statistical summary of the normalised changes in $\Delta F/F_0$ obtained from experiments in (a-f). Shown are the ligand-dependent differences in $\Delta F/F_0$ after $G\alpha_{oA}(G203T)$ overexpression (G_{mut}) normalised to the mean $\Delta F/F_0$ of endogenous G-protein levels (wt, set to 0%). Negative values indicate decreases in $\Delta F/F_0$ after $G\alpha_{oA}(G203T)$ overexpression, and positive values indicate increases in $\Delta F/F_0$ after $G\alpha_{oA}(G203T)$ overexpression. Bars indicate means \pm sem of at least 3 independent experiments, each data point represents a single cell. **** $p < 0.0001$, *** $p < 0.001$, ** $p < 0.01$, * $p < 0.05$, according to an unpaired two-tailed t-test. wt, wildtype; G_{mut} , $G\alpha_{oA}(G203T)$; ns, not significantly different.

Revised Supplementary Figure 4 (former Extended Data Figure 7)

Supplementary Figure 4: Conformational changes at M₂R-N419 are independent of G-protein activity. (a) Representative traces of $\Delta F/F_0$ of single cells expressing the M₂R-N419 biosensor (SP-M₂R^{N419TAG}) at endogenous G-protein levels (left) or after $G\alpha_{oA}(G203T)$ overexpression background (center) and stimulated with 1 mM ACh (grey), 100 μ M Ixo (green), 1 mM Arec (purple), or 10 mM Pilo (pink). Shaded areas indicate the duration of agonist superfusion, unshaded areas represent buffer application. (b) Statistical summary of the normalised changes in $\Delta F/F_0$ obtained from experiments in (a). Shown are the ligand-dependent differences in $\Delta F/F_0$ after $G\alpha_{oA}(G203T)$ overexpression (G_{mut}) normalised to endogenous G-protein levels (wt, set to 0%). Negative values indicate decreases in $\Delta F/F_0$ after $G\alpha_{oA}(G203T)$ overexpression, and positive values indicate increases in $\Delta F/F_0$ after $G\alpha_{oA}(G203T)$ overexpression. The bars represent means \pm sem of at least 3 independent experiments, each data point represents a single cell. Number of cells (n): n = 13 (Ixo), n = 24 (ACh), n = 13 (Arec), n = 14 (Pilo). ***p < 0.001, according to an unpaired two-tailed t-test. wt, wildtype; mut, mutant; ns, not significantly different.

NEW Supplementary Figure 5

Supplementary Figure 5: Overexpression of Ga_{oA}(G203T) changes the basal equilibrium of receptor/G-protein complexes. Comparison of the ligand-dependent fluorescence changes ($\Delta F/F_0$) after Ga_{oA}(G203T) overexpression or at endogenous G-protein levels after activation of the indicated M₂R biosensors (*SP-M₂R^{XXXTAG}*) with the ligands 100 μ M Ixo (green), 1 mM ACh (grey), 1 mM Arec (purple), and 10 mM Pilo (pink). For each condition, the data has been normalised to the mean $\Delta F/F_0$ of ACh (set to 100%). The bars represent means \pm sem of at least 3 independent experiments, each data point represents a single cell. Number of individual cells (n, for Ixo, ACh, Arec, Pilo): n = 15, 14, 10, 20 (T84), n = 11, 10, 9, 11 (E175), n = 29, 41, 34, 20 (F181), n = 16, 26, 13, 17 (F188), n = 12, 31, 10, 16 (A414), n = 28, 21, 19, 10 (P415), n = 13, 24, 13, 14 (N419). ****p < 0.0001, ***p < 0.001, **p < 0.01, *p < 0.05 according to an one-way analysis of variance (ANOVA multiple comparisons test) with Tukey's posthoc test. ns, not significantly different.

Figure 4 – Minor Comment

The authors map the time scales of the conformational changes observed in Figures 1-3 onto a plane, capturing differences in the activation state of M2R depending on ligand efficacy. I think that this figure would be more powerful if the authors presented a conformational model (rather than the planar images) based upon the solvatochromic properties of the cyanine dyes and the structural differences observed in the extracellular domain between inactive and active state structures of M2R or other GPCRs.

Authors' reply: The Reviewer's comment reflects precisely what we envision achieving with our sensor. However, as outlined in detail above, our biosensor data do not allow to delineate detailed structural information because a still relatively long and flexible linker separates the fluorophores from the receptor backbone. This does, unfortunately, not really allow an atomistic, structural interpretation of the observed fluorescence changes impossible. Moreover, the agonist-elicited fluorescence changes of the biosensor panel are quite large, which is hardly compatible with the rather small conformational changes of the ECLs derived from superpositions of the inactive and active receptor structures (c.f. Figure 1E). It is tempting to hypothesize that the structural dynamics of the ECLs are much larger in intact cells as might be anticipated from such structural studies. Supportive evidence for this hypothesis comes from a recent study by the Prosser lab (Picard et al., *Nat. Chem. Biol.*, 2024, <https://doi.org/10.1038/s41589-024-01682-6>) that demonstrates using NMR and computational methods that ECL movements initiated by intracellular G-protein binding can be dramatically greater and more complex than receptor structures have implied.

Based on these considerations, in the revised version of the manuscript, we decided to continue to present our biosensor data in simplified 2D plane images and to avoid structural overinterpretation of our fluorescence data.

Figure 5 – Minor Comment

The authors correlate their observations of ligand efficacy-dependent conformational changes in the extracellular domain of the receptor with changes in G protein selectivity, wherein lower efficacy ligands appear to preferentially activate G protein isoforms. To make this correlation more effective, I think that the authors should explicitly include their model presented in Figure 4 with the signaling data presented in Figure 5. I think this data provides some important insight into the controversy in the field surrounding ligand efficacy versus ligand bias and merits some further elaboration in the discussion.

Author's reply: We thank the Reviewer for this suggestion. We have revised both figure Figures 4 and 5 with the intention of graphically strengthening the correlation between ligand efficacy-dependent conformational changes in the extracellular domain of the receptor (Figure 4) with changes in G-protein selectivity (Figure 5). Specifically, in Figure 4 we have implemented a new colour code to highlight the differences between the ligand-specific receptor/G-protein complexes. We have coloured changes in positions 414 and 415 (indicative for the C1 complex) in blue, changes in positions in 181 and 188 (indicative for the C2 complex) in yellow, and changes in positions 84 and 175 (indicative for the activation trajectory) in pink or mint dependent on the direction of the fluorescence signal. For each ligand-dependent R/G complex at a certain time after agonist activation, we have summed the colour intensities from all positions and labelled the respective plane in that colour. By this, the difference between the ligands becomes visually more evident, especially for arecoline which promotes a different receptor activation trajectory. In Figure 5 F, we have used this colour code for the 4 ligands to incorporate their conformational profile directly into their G-protein activation profile. Furthermore, we have elaborated on this correlation in the discussion of Figure 5. more as suggested by the Reviewer.

Revised Figure 4

Figure 4: Ligand-specific activation trajectories and equilibria of M_2R/G protein-signalling complexes. Schematic three-dimensional representation of the time-resolved, agonist-mediated formation of GPCR signalling complexes in intact cells. The planes illustrate the receptor's extracellular surface, and the numbered positions indicate the 6 biosensors that were sensitive to G-protein modulation (see Fig. 3). Agonist-promoted fluorescence changes are depicted as peaks. The mean fluorescence changes (see Fig. 2) define the height of each peak. For better visualisation, the direction of fluorescence changes is inverted (i.e. fluorescence decreases are shown as increasing peaks). Peaks from the same group of biosensors are depicted in the same colour. The colour of each plane is the sum of the colours of all peaks in that plane. This representation yields unique colours that allow to visually discriminate between ligand-specific receptor/G protein-signalling complex. For simplification the apo state is depicted in grey and does not result from the sum of peak colours. The position of the conformational equilibria at steady state are indicated by the equilibrium arrows. Indicated times are apparent on-rates of agonist-specific fluorescence changes (Ext. Data Fig. 9, Supp. Tabl. 3). C: complex; #: intermediate complex.

There also appears to be some lack of consensus between panel C and panel F, specifically for the G_{i3} and G_{15} data points. I would recommend some other form of data presentation to alleviate this.

Authors' reply: We thank the Reviewer for drawing attention to the apparent discrepancy between panels C and F in Figure 5. In response, we carefully revisited the choice of representative traces. We have now updated panel 5c to better reflect the average data and improve consistency across panels.

Revised Figure 5

Figure 5: Activation trajectories and conformational equilibria define ligand efficacy in living cells. (a) TRUPATH assay principle. The BRET-based $G\alpha\beta\gamma$ biosensors sense the conformational rearrangement during receptor-promoted G-protein activation as a decrease in BRET due to the increased distance between $G\alpha$ - and γ -subunits⁵⁴. (b-e) Representative traces of agonist-promoted changes in normalised Δ BRET (%) for all G-protein biosensors upon activation of wildtype M_2R ($SP-M_2R-wt$) with the indicated agonists. Agonists were applied (black arrows) at saturating concentrations of (b) ACh, (c) Ixo, (d) Arec, and (e) Pilo. (f) G protein-coupling selectivity profile of the M_2R . Overview of agonist-promoted Δ BRET normalised to ACh (set to 1) obtained upon agonist-promoted, M_2R -mediated stimulation of BRET-based G-protein biosensors (TRUPATH). The coloured boxes (using the colour code from the planes in Fig. 4) on top of each column represent the equilibrium of receptor/G-protein complexes stabilised by the indicated ligand (see Fig. 4). Heatmap values represent mean \pm sem obtained from ≥ 5 experiments (see Supp. Table 4). **** $p < 0.0001$, *** $p < 0.001$, ** $p < 0.01$, * $p < 0.05$, according to ANOVA multiple comparisons test with Dunnett's post-hoc test (ACh as reference). ns, not significantly different.

Referee #3 (Remarks to the Author):

In this manuscript, Thomas et al., developed biosensors for studying GPCR signaling in intact cells. They focused on the M2 muscarinic acetylcholine receptor (M2R) and utilized these newly created M2R biosensors to study the receptor's response to stimulation in living cells. While I am not an expert in GPCR biology, technically this manuscript nicely illustrates the power of genetic code expansion and click chemistry. By combining these advanced technologies, the authors were able to design a set of minimally invasive fluorescent M2R biosensors. To create these sensors, genetic code expansion was used to incorporate noncanonical amino acids (ncAAs) site-specifically into the extracellular domain of M2R. Once incorporated, the reactive ncAAs were labelled with a cell-impermeable tetrazine-conjugated fluorescent dye using a bioorthogonal click chemistry reaction. This approach allowed the authors to target various extracellularly accessible residues and study GPCR conformational changes by evaluating fluorescence intensity variations. I see significant potential and value in using these technologies to address challenges in the studies of GPCRs. However, I also see some shortcomings that should be addressed prior to publication of this manuscript:

1. This manuscript relies on genetic code expansion and click chemistry to fluorescently label different M2R variants. While these technologies have been previously used by the authors and others for fluorescent protein labelling in various applications, including studies of not only GPCRs but also a wide range of other proteins, they are still not widely adopted. For this reason, it is particularly important to provide a comprehensive description of these techniques in both the main text and the methods section. However, my impression is that the manuscript devotes insufficient attention to these aspects. Furthermore, the current version contains numerous references to "as described previously" in both the main text and methods section, which I believe is inappropriate. I will outline specific examples below:

a. It is impressive that the authors tested 72 extracellularly accessible residues for ncAA incorporation in M2R. However, the manuscript should provide a more detailed explanation of what is required to create such biosensors. For example, the authors do not seem to describe the plasmid and genetic elements which are required to incorporate ncAAs. This information is essential for readers who are not familiar with genetic code expansion but are interested in developing or utilizing these biosensors in future studies. It might help to include additional references that deal with ncAA incorporation and click chemistry fluorescent labelling in living cells to provide further context and guidance for interested researchers.

Authors' reply: We thank the Reviewer for this helpful comment. We agree that genetic code expansion (GCE) technology should still not be considered standard for the general readership. Therefore, we made the following changes, both in the main text and in the Methods section:

1) In the main text, we have written the description of the procedure how to install the label at pages 4-5 in a more extensive and clearer way for non-specialists. The section reads now:

"The least invasive way to attach probes to a GPCR at single-residue resolution in living cells is by bioorthogonal chemistry on genetically encoded chemical anchors¹²⁻

¹⁵. Briefly, a non-canonical amino acid (ncAA) carrying an anchor for rapid catalyst-free labelling is incorporated into the receptor using the genetic code expansion technology (GCE). The label is then attached post-translationally by ultra rapid Strain-Promoted Inverse Electron-Demand Diels-Alder Cycloaddition (SPIEDAC), a reaction that occurs within a couple of minutes without interfering with other functional groups naturally present in the proteins. Using this strategy, we have previously demonstrated up to quantitative labeling of GPCRs on the live cell surface⁴⁰.

We screened now the entire extracellular surface of the M₂R to identify positions at which the click-ncAA trans-cyclooct-2-ene lysine (TCO*K)⁴¹ was efficiently incorporated and yielded robust labelling with a cell-impermeable, tetrazine-conjugated cyanine dye (Tet-Cy3) (Fig. 1a).”

- 2) We have added detailed information about GCE in the Methods section. Specifically, we have included an introductory subsection that thoroughly — but in accessible terms — describes the genetic components required for ncAA incorporation, along with the plasmids used in this study. In addition, we now explicitly mention that we have deposited the plasmid for TCO*K incorporation in Addgene, where further documentation, including the full sequence, is readily available. This section refers to the recent review by Dunkelmann and Chin on the pyrrolysine system, as well as our original reference in which the plasmid used in this work was developed. The section reads as follows:

“Principle and rationale of ncAA incorporation: To genetically encode a non-canonical amino acid (ncAA), a so-called "orthogonal" aminoacyl-tRNA synthetase (AARS)/tRNA pair must be introduced into the host cell. This pair does not cross-talk with the endogenous synthetase/tRNA pairs responsible for incorporating canonical amino acids. Typically, the orthogonal pair is derived from a different organism—for example, bacterial pairs are commonly used for genetic code expansion (GCE) in eukaryotic cells. The ncAARS specifically recognizes the ncAA, while the orthogonal tRNA functions as an amber suppressor: it carries an anticodon complementary to a stop codon (usually the amber stop codon UAG) and competes with termination factors to reassign this natural nonsense codon as a sense codon. The ncAARS charges the orthogonal tRNA with the ncAA, which is then delivered to the ribosome for regular incorporation into the nascent protein. Each ncAA requires a specific AARS, although some AARSs can accommodate more than one amino acid.

In our lab, we have established a two-plasmid system to incorporate non-canonical amino acids (ncAAs) into proteins of interest (POIs). One plasmid (typically pcDNA3) carries the gene encoding the POI, in which a TAG stop codon replaces the natural codon at the position targeted for ncAA incorporation. The second plasmid is a bicistronic construct that encodes the translational machinery: the aminoacyl-tRNA synthetase (AARS) and the corresponding tRNA. To ensure proper expression and processing of the procaryotic tRNA in mammalian cells, the tRNA gene — lacking the 3'-CCA sequence — is placed under the control of external Pol III promoters (H1 or U6) and followed by an appropriate trailer. To achieve the high tRNA concentrations required to outcompete the release factor, the tRNA expression cassette is typically repeated in tandem.

The orthogonal pair used for TCO*K incorporation is derived from the system that naturally incorporates pyrrolysine (Pyl) in methanogenic archaea in response to the amber codon (UAG)⁵⁸. Specifically, the plasmid contains one copy of the *Methanococcus barkeri* (Mb) pyrrolysyl-tRNA synthetase (PylRS) under the control of

a CMV promoter, along with four tandem copies of the gene encoding the enhanced M15 tRNA for expression in mammalian cells⁵⁹. The plasmid, which was generated in our lab, is deposited in Addgene, where the complete map and additional information can be found (Plasmid #105830 <https://www.addgene.org/105830/>).”

- 3) We have revised the reference list. For genetic code expansion (GCE) and bioorthogonal labeling, we now cite three comprehensive review articles and one protocol paper: 1) **An earlier review by Wang and Schultz**, which focuses on the general concept of GCE and the early development of the technique; 2) **The landmark review by Lang and Chin**, which provides an in-depth description of GCE in mammalian cells, along with an excellent overview of bioorthogonal labeling concepts, available reactions, and suitable ncAAs; 3) **Our recent review (Coin and Sakmar)**, which emphasizes the application of GCE to membrane proteins—especially GPCRs—and includes both basic GCE concepts and a practical overview of protein labeling strategies; 4) **A protocol paper by Nikic and Lemke**, which describes in detail how to label proteins at the cell surface using different ncAAs. In addition, we cite the original publication describing the ncAA used in this study (TCO*K, Lemke 2014) as well as our own work (Serfling et al., 2019), which demonstrates quantitative bioorthogonal labeling of GPCRs using this strategy. We believe that this selection is the most appropriate for this manuscript, as it covers both general concepts and key methodological details, while allowing interested readers to explore the topic further through the references cited in these papers.

b. Similarly, the authors should describe how the ncAA was chosen. Are there any limitations to keep in mind? For example, the click reaction between TCO-Lysine and tetrazines used in this study is known to potentially undergo a click-to-release mechanism. Discussing such limitations and the rationale behind the choice of ncAA would provide valuable insights for readers, particularly those considering using this approach in their own research.*

Authors' reply: We have added a paragraph that describes the reasons for choosing SPIEDAC on TCO*K in the Methods section. We believe that a thorough discussion of the details of each possible labelling strategy goes beyond the scope of this manuscript and may require a specific methods/protocol-type publication (see above). The new paragraph is a straight-to-the point description of our experience and rationale and includes essential references, where further discussion can be found. It reads as follows: “Fluorescent labels were attached to the receptor using ultra-rapid click chemistry between dye–tetrazine derivatives and the non-canonical amino acid TCO*K. While we have experience with other labelling chemistries in our lab¹³, such as copper-catalysed azide–alkyne cycloaddition (CuAAC) on both azide- and alkyne-containing ncAAs, and strain-promoted azide–alkyne cycloaddition (SPAAC) on ncAAs bearing strained alkynes (e.g., BCNK, see ¹⁴), we have consistently achieved the best results in terms of labelling speed, efficiency, cell viability, and reproducibility using SPIEDAC on TCO*K. This ncAA carries a selected isomer of cyclooctene (trans-2-cyclooctene), which is highly reactive

with both tetrazine and methyl-tetrazine dye derivatives yet sufficiently stable over the duration of the experiment⁴¹.”

c. The authors primarily used Cy3 and they mention that Cy3 is environmentally sensitive. Is there any published work to back this up? Also, what about other fluorophores in addition to Cy3 and Cy5? Could they be used?

Authors' reply: We have included references that demonstrate the environmental sensitivity of cyanine dyes such as Cy3 (Levitus and Ranjit. *Q. Rev. Biophys.* 2011. <https://doi.org/10.1017/S0033583510000247>, Lamichhane et al. *Proc. Natl. Acad. Sci USA.* 2015. <https://doi.org/10.1073/pnas.1519626112>, Lamichhane et al. *Structure.* 2020. <https://doi.org/10.1016/j.str.2020.01.001>).

We believe that, in principle, any fluorophore that exhibits environmental sensitivity may be used for this technique. Following the suggestion by the Reviewer, we have included new data in the revised version (**Supplementary Figure 2**) of the manuscript that demonstrates that labelling with Cy5 and TAMRA results in the generation of biosensors.

NEW Supplementary Figure 2

Supplementary Figure 2: Labelling of the M₂R⁴¹⁴ construct with different dyes also gives GPCR biosensors. Representative traces of $\Delta F/F_0$ of single cells expressing the M₂R⁴¹⁴ biosensor (*SP-M₂R^{A414TAG}*) and labelled with Tetrazine-TAMRA (a) or Tetrazine-Cy5 (b) and stimulated with 1 mM ACh. Shaded areas indicate the duration of agonist superfusion, unshaded areas represent buffer application. Data obtained from 3 independent experiments. Number of cells (n): n = 31 (a), n = 27 (b).

d. For proper biosensor variants characterization, click labelled controls without ncAA should also be included.

Authors' reply: We fully acknowledge the rationale for including a labelled control without ncAA incorporation. In the course of our thorough biosensor characterisation, we never observed any detectable labelling in cells lacking a properly incorporated ncAA (so that we could not image these samples). This corroborates the specificity of our labelling approach.

e. According to the methods section, various M2R construct variants were generated, including full-length M2R protein (without additional tags) and variants with HA-tag and GFP reporter. However, it is not entirely clear which variant was used for each specific purpose. This should be better described and clearly indicated in the text and in the corresponding figure legends.

Authors' reply: We apologize for this lack of clarity in reporting the various receptor constructs. In the revised version of this manuscript, we have added to the Methods parts which receptor construct was used for which experiment. Further, we have included this information in every figure legend.

f. The Biosensor Construction section mentions the cloning of a signaling peptide, but it is not clear why this step is necessary and if it was used in all of the constructs.

Authors' reply: We thank the Reviewer for this comment. When working with modified GPCRs (e.g. that are fused to self-labeling tags or fluorescent proteins and that often become impaired in cell surface targeting), it is a common strategy in the field to add a signal peptide to the N-terminus to increase receptor translocation to the ER membrane and, thereby, assure adequate cell-surface expression. Of note, the signal peptide that we used is self-cleavable so that it does not affect receptor function. Ref. 56 of the original manuscript ([https://doi.org/10.1016/S0021-9258\(18\)41623-7](https://doi.org/10.1016/S0021-9258(18)41623-7)) includes more details on the signal peptide.

g. What makes me wonder is the expression level of these biosensors. If I understand correctly, the biosensor variants will compete with endogenous, unlabeled GPCRs. If the expression levels of the different biosensor variants are not consistent, this could potentially influence the observed responses upon stimulation. This issue should be discussed and it would help to provide some kind of comparative analysis of the expression levels of different variants.

Authors' reply: The Reviewer raises a conceptually important point. However, the cell lines that we use do not express M₂ receptors endogenously so that the biosensors do not interact with unlabeled M₂ receptors. Further, the expression levels of all biosensors have been thoroughly measured and have been plotted in Extended Data Fig. 3. From this analysis it can be seen that all biosensors are expressed at a similar level considering that we work under overexpression conditions. This is further reflected by the new data provided in Extended Data Fig. 6 and Supplementary Table 2 which demonstrate that the potencies for G-protein activation are only minimally affected despite small differences in biosensor expression.

2. Not only related to the genetic code expansion and click chemistry-based labelling, the manuscript and methods section should be carefully reviewed to replace "as previously described" references with more detailed descriptions.

Authors' reply: We thank the Reviewer for this comment. In the revised version of the manuscript, we have deleted all "as previously described" statements and provide more experimental details instead.

3. In some of their figure legends, the authors use a wording "3 independent days of experiments". At least to me, this sounds confusing.

Authors' reply: We thank the Reviewer for pointing this out. We have changed the wording consistently to "No. of independent experiments" in all figure legends.

4. *In some of their figure legends, the authors mention individual wavelengths for excitation and emission of Cy3. I do not understand why. It is enough to describe this in the methods sections. Even there, one needs to describe the laser line that was used for the excitation and the individual filter cubes (or spectral range) that were used to collect the emitted light.*

Authors' reply: We are very grateful for this comment and have deleted all statements to the individual wavelengths in the figure legends. The Methods section contains all necessary information regarding laser lines and filter cubes.

5. *More generally, it would be helpful to provide an outlook on the broader applicability of this approach. What would be required to create similar biosensors for other GPCRs? Additionally, how might these biosensors be adapted for use in other biological systems or cell types? A discussion of these possibilities would greatly enhance the potential impact and generalizability of the study.*

Authors' reply: We agree that we should provide an outlook on the broader applicability of our biosensor approach. In fact, we have already begun the transfer of the technology to other GPCRs (for data on the dopamine sensor, please see our response to Reviewer 1 (page 1 of this rebuttal letter). We have included such an outlook in the conclusions of the revised manuscript, at the very end of the main text:

“Moreover, we anticipate that our single-color conformational sensor technology will be broadly applicable to other receptors, enabling the temporal dissection of conformational changes elicited by structurally distinct ligands, ranging from small molecules to larger peptides with diverse pharmacological profiles. This unique information, derived from the live-cell context, will enhance development of GPCR drug candidates with unique signalling profiles and expand our understanding of the general principles underlying GPCR structural dynamics.”

6. *What are the main advantages and limitations of this approach, and how will the findings of this study help advance further research in the field? Furthermore, how can researchers best utilize the different biosensor variants developed here? What is their value beyond this specific study, and how might they contribute to future investigations?*

Authors' reply: We thank the Reviewer for this suggestion. We have rewritten the first part of the conclusion section and highlighted some key advantages and limitations in the frame that we think is adequate for the manuscript.

“By attaching minimal-sized fluorescent labels to a set of activation- and G-protein coupling-sensitive positions on the extracellular receptor surface, we were able to track a receptor's activation trajectory in real-time with unprecedented conformational detail, directly in the native membrane environment of an intact cell. Unlike earlier fluorescent GPCR activation biosensors that relied on fusion to large fluorescent proteins, self-labelling tags or, at best, oligomeric epitopes⁵³, our sensors report on the movement of single receptor positions labeled with a fluorophore anchored *via* an ncAA, while leaving the intracellular surface of the receptor completely untouched. Although the method does not reach the atomic-level resolution of biophysical techniques such as NMR or DEER — since fluorophores are attached to the receptor backbone via a relatively large

and flexible linker — it offers the highest spatial and temporal resolution currently achievable for tracking conformational changes in receptors within live cells.”

We feel that our new technology does indeed allow new conclusions and interpretations. Thus, at a later point in time, when similar experiments will have been done by ourselves and hopefully other groups, and possibly when more comparisons with structural data have become possible, it would be interesting to summarize the interpretations and new venues that our technology permits.

Responses to the reviewers' comments

We would like to thank all three Reviewers again for their very positive and helpful comments, which have allowed us to further improve the manuscript. Please find our specific answers to all individual comments made by the three Reviewers below.

Referee #1:

The revised manuscript by Thomas et al. represents a significant improvement to an already interesting story. The authors have responded mostly well to this and other reviewers' comments in the form of stylistic improvements as well as providing additional experimental data. However, there are some old and new issues that need to be addressed by the authors in the form of nuanced statements and better clarification (see below).

Authors' reply:

We thank the Reviewer for re-evaluating our manuscript and we are excited that the Reviewer finds it significantly improved. Below we are taking up the Reviewer's offer to reply to the additional points in the form of nuanced statements and better clarifications.

Major comments:

The following comments are listed in order of their appearance in the manuscript, rather than in order of their importance.

1. There is a question of whether there is a difference in labeling efficiency (of the fluorescent moiety) with different agonists. Is there a difference in fluorophore access to the non-canonical tRNA-encoded residue by various conformations stabilized by agonists or antagonist? The authors did show that ligands such as NMS did not interfere directly with the Cy3, either conformationally or through altering the fluorophore's spectral properties. The authors do report differences in labeling efficiency between various positions, which could be directly related to receptor conformation and possibly sensitivity to agonists of varying efficacies. For example for positions 181, 188 and 419, where labeling efficiencies are lower than other residues, could be restricted unless in the active state. Under basal conditions the GPCRs do drift into the active state.

Authors' reply:

We thank the Reviewer for this insightful comment regarding possible differences in labeling efficiencies for distinct receptor conformations.

First of all, and most importantly, we would like to stress that in our experiments, labeling is always performed **prior** to agonist addition, that non-bound fluorophore is removed before experimentation, and that agonist-induced fluorescence signals from all biosensors are normalized to basal fluorescence and to responses elicited by the endogenous agonist. Therefore, our protocol corrects for any possible variation in labeling efficiencies across conditions and, thus, none of the conclusions drawn from the normalized responses are affected by differences in labeling efficiencies.

Further, on a more general note, labeling efficiencies of genetically encoded chemical anchors primarily depend on various chemical factors – e.g. from solvent accessibility

to orientation of molecular orbitals – that are hardly controllable, as demonstrated by the fact that it is still impossible to predict in a reliable way whether a position is going to be efficiently labeled or not, even when the receptor structure is known. Indeed, we have previously shown that contiguous positions in a flexible loop, which are all expected to be well accessible, can have enormous differences in labeling efficiencies (<https://doi.org/10.1021/acscchembio.8b01115>).

Investigating agonist-dependent effects on labeling efficiency, as the Reviewer suggests, might provide an elegant, independent way of demonstrating that our new biosensor technology reports on receptor conformational changes. Although the suggested experiments would provide access to ligand-dependent labelling kinetics, these would not reflect the kinetics of ligand-mediated conformational changes, which we measured with our experimental design.

As our study in its current form, and by the various analyses presented, directly establishes the principle that the biosensors indicate receptor conformational changes, such additional experiments would not advance the conceptual insight into the underlying biology. Rather, they would constitute a complementary methodological validation that, while interesting, lies clearly beyond the main focus of the current work.

For these reasons, we have decided to not pursue a comprehensive analysis of agonist-dependent labeling efficiencies in the current manuscript.

2. The authors did state that Cy3 is environmentally sensitive but really didn't describe how and why. It would be useful for readers to understand that solvent exposure is likely the key player in fluorescence quenching. For example, movement of the fluorescent moiety into a more solvent exposed environment as part of the change in conformation would appear as a decrease in fluorescence emission. So, biosensors that display agonist-stimulated fluorescence decreases likely have their fluorophore, in their basal states, lying in a relatively hydrophobic environment like the vestibule itself. Thus, conformational changes in the ECLs following agonist addition move the fluorophore directly into solvent. Similarly, agonist-promoted increases would put the fluorophore into a more hydrophobic environment.

Authors' reply:

We thank the Reviewer for this comment, and, in the revised manuscript, we have expanded the description of this important aspect.

It is true that cyanine dyes, based on the basic principle of physics, increase their fluorescence intensities when moving from a more polar to a more hydrophobic environment. However, as pointed out in the revised manuscript and in the previous round of revisions, the post-translational modification of ncAAs – at least currently – requires a relatively long and flexible spacer between the fluorophore and the protein backbone, making it impossible to draw detailed structural conclusions from the observed fluorescence changes. This is why in the previous version of the manuscript we wrote “*We propose that the observed changes in fluorescence result from local changes in their microenvironment caused by ligand-promoted conformational changes of the receptor*”. Following the Reviewer’s suggestion, in the revised version we have expanded this statement that now reads “*...from local changes in their microenvironment (e.g. **transitions to either more hydrophobic or to more polar microenvironments**) caused by ligand-promoted conformational...*”.

3. On p. 9 the authors describe the conformational states stabilized by agonists compared to G proteins. Herein are a couple details that need to be clarified. The G203T mutant indeed has poor affinity for nucleotides, which is why it is useful for stabilizing RG complexes for structural work. Hence it is understood that the G203T mutant stabilizes an active conformation similar to agonists. Several biophysical studies have suggested that both G protein and/or agonist addition alter the distribution of conformational states, favoring those of the active states. In this manner agonist and nucleotide-free G proteins act cooperatively to stabilize the fully active state better than G proteins or agonist alone. The question is whether failure of agonists to induce changes in fluorescence with some of the sensors when G302T is present is actually due to the fact that G203T itself adopted something close to the fully active conformation. Furthermore, these ensemble measurements really can't resolve multiple active conformations. This would require single molecule spectroscopy work (also with FRET or Trp quenching) where distributions in fluorescence lifetimes can be measured.

Authors' reply:

We are grateful for the Reviewer's thorough and thoughtful engagement with the interpretation of these results. Upon the Reviewer's request, in the revised version of the manuscript we softened slightly our wording ("As a result, agonist stimulation of biosensors indicating the formation of the high-efficacy M₂R/G-protein complex **would** result in reduced or no changes in fluorescence emission intensities").

However, we do believe that our interpretation put forward in the original manuscript remains the most likely, and the biochemical properties of the G203T mutant that the Reviewer points out, are entirely consistent with and support our original reasoning. Moreover, our new data with the M₂R-Y426A mutant (**Extended Data Figure 7**) strongly reinforce our interpretation that the M₂R¹⁷⁵ biosensor reports on receptor conformational changes that involve the 'lid closure'. While we cannot fully exclude the possibility of alternative explanations, however, there is currently no direct way of testing them in living cells.

Further, we fully agree with the Reviewer that, in ensemble measurements as we have performed, a **single** biosensor cannot directly report on multiple active receptor conformations. However, it is the **combination of seven GPCR biosensors** and analysis of the **efficacy rank order** of different agonists that has the power to unequivocally infer the existence of multiple active receptor conformations in intact cells. And this is what we state in the manuscript on page 8 "*The observation that the extent of conformational changes elicited by agonist activation scales directly with agonist efficacy at some receptor positions but inversely at others suggests that the M₂R adopts multiple active states in living cells, which differ in their capacity to stimulate G-protein signalling.*" We believe that being able to generate an entire panel of biosensors by bioorthogonal labeling is a key novelty of our technology that allows a new level of conceptual insight into GPCR biology in intact cells that was inaccessible to previous GFP-based fluorescent biosensors.

Lastly, we agree with the Reviewer that single-molecule techniques in intact cells would provide an even more direct method to resolve multiple active receptor conformations. However, due to the fast lateral diffusion of receptors in membranes of intact cells, this is one of the biggest challenges in the field and developing such techniques will require several years. In fact, to the best of our knowledge, there is only a single publication

that reports on single-molecule FRET analysis of a Class C GPCR in intact cells (<https://doi.org/10.1038/s41592-021-01081-y>), and for Class A GPCRs, which we study in this manuscript, this has never been achieved.

4. In the manuscript the authors refer to the non-G protein-coupled receptor as the 'basal condition' in contrast to WT or mutant Galpha overexpression. This is confusing to readers. Basal conditions should really only refer to non-agonist/inverse agonist treated receptor activity. As stated it is not always clear whether the authors are referring to no G proteins or no agonist, or both. Comparing receptor activity in the absence or presence of G protein should be referred to as the uncoupled or coupled-state, in this context.

Authors' reply:

We are grateful that the Reviewer has spotted these inconsistencies. In the previous version of the manuscript, we have referred to the "basal" conditions in cases where experiments were done in cells without overexpression of the G-protein mutant $G_{\alpha O A}(G203T)$, i.e. under the endogenous G-protein repertoire of the cell. We understand that referring to this as "basal" can be misleading to readers. Therefore, in the revised manuscript, we now always explicitly state "endogenous G-protein expression/repertoire" for experiments that were performed without overexpression of the $G_{\alpha O A}(G203T)$ mutant.

5. On p. 12 the authors describe the kinetics of fluorescence changes due to agonist treatment on biosensors 181 and 188 and how they are slower than with other biosensors. Is this due to steric affects of the bulky fluorophores on agonist entry into the orthosteric site? Have the authors tried looking at [3H]NMS on-rates to 181 and 188 compared to other biosensors? This is important since the authors on p. 13 are arguing that this is naturally a slower conformational step that follows other conformational changes, indicated by other biosensors.

[REDACTED]

[REDACTED]

[Redacted text block]

[Redacted text block]

[Redacted text block]

[Redacted text block]

[Redacted text block]

Minor comments:
The description of the actions of PTX in p. 10 is a little misleading. The major effects of PTX-mediated ADP ribosylation of the C-terminus of Galphai/o prevents coupling through sterically occluding access to the receptor core. This is why it affects nucleotide exchange, especially for the use of PTX in this study.

Authors' reply:

We thank the Reviewer for this comment. We have revised the respective section in the manuscript which now reads “By ADP-ribosylation of $G_{\alpha_{i/o}}$ -subunits, PTX locks the α -subunits of endogenous $G_{i/o}$ proteins into an inactive, GDP-bound state⁵⁰, **which hampers productive coupling between receptors and G proteins**”. Along this line we would like to note that even in the presence of PTX, G-proteins can still assemble (unproductively) with receptors, i.e. they can physically contact receptors (<https://doi.org/10.1124/mol.106.030304>, <https://doi.org/10.1096/fj.10-154997>, <https://doi.org/10.3389/fendo.2012.00082>).

The first use of the term non-canonical amino acid (ncAA) in the introduction should also include ‘as known as unnatural amino acid incorporation’ or something of the sort.

Authors’ reply:

We have revised the respective section in the manuscript which now reads “Briefly, a non-canonical amino acid (ncAA), **also referred to as unnatural amino acid**, carrying an anchor for rapid catalyst-free labelling is incorporated into the receptor using the genetic code expansion technology (GCE).”

Referee #2:

The authors’ reluctance to provide a more structural interpretation of biosensor readout are fair, and I think that this viewpoint is clear in the data interpretation throughout the manuscript. However, I think that both sets of experimental data provided in the response letter (Reviewer Figures 2 and 3) provide important context for the study. The demonstration with LY2119620 in Reviewer Figure 2 is important for the significance of this study. For this technology to be transformative for GPCR drug discovery down the line, it is paramount for it to provide some insight as to how drugs influence the dynamics of the extracellular domain of the receptor, regardless of a structural interpretation of the readout. To this end, demonstrating how these biosensors report on an allosteric compound that directly interacts with the extracellular loops serves this purpose.

Furthermore, the experimental data in Reviewer Figure 3 (and interpretation provided by the authors) also provide essential physical context for the biosensor readout, especially given that the biosensor experiments are performed at (near) saturating concentrations that appear to still be able to activate G proteins downstream of the compromised biosensor. I would be curious as to how the chosen mutation influences G protein activation downstream of a non-biosensor version of M2R, which is relevant to my next point.

Authors’ reply:

We are very grateful for the Reviewer’s fair and really supportive evaluation of our manuscript. Upon the Reviewer’s request, we have included the two mentioned Figures into the revised manuscript as **Extended Data Figs. 3 and 7**. Integration of the two new Figures has resulted in the following slight changes to the main text:

Page 6: “Importantly, all seven biosensors retain the ability to respond to the positive allosteric modulator LY2119620 which binds to the allosteric binding site located at the extracellular side of the receptor (Extended Data Fig. 3).”

Page 9: “In support of this conclusion, Ala mutation of Y426, a key residue involved in this ‘lid closure’, led to a complete loss of ACh-induced fluorescence changes in the M_2R^{175} biosensor (Extended Data Fig. 7).”

We have not performed G-protein activation experiments with a non-biosensor version of the Y426A mutant. However, there are data in the literature (<https://doi.org/10.1124/molpharm.122.000661>) that show that mutation of M₂R-Y426 to phenylalanine reduces both the affinity and potency of ACh and iperoxo approximately by ≈2 orders of magnitude.

*I maintain that there is a correlation between the response of the Gi13 FRET sensor in Extended Figure 3, and the potency of the biosensor response to acetylcholine in Extended Data Figure 5. For clarity, here is a plot of this comparison, pulled from the source data provided by the authors. This correlation suggests that the more potent the biosensor fluorescence change is to the reference ligand, the larger the impact there is on G protein activation. This correlation between biosensor readout and function implies that **the biosensors could be influencing the activation states occupied by the receptor**, instead of passively reporting on the conformational states occupied by the receptor. To this end, there is also a convincing trend between another functional readout, the TRUPATH EC50 of the biosensor for acetylcholine (Extended Data Figure 6), and the ability of a sensor to differentiate between high and low efficacy signaling complexes (Figure 3). It is important for the reader to understand these implications, especially as this technology is translated to other receptor systems.*

Authors' reply:

We truly appreciate the Reviewer's rigorousness in assessing our data and deeply engaging with both our novel technology and the unprecedented mechanistic insights it provides into GPCR biology. This has been extremely helpful during both rounds of revisions.

The correlations provided by the Reviewer (now included in **Extended Data Fig. 4i**) imply that biosensors at which ACh has higher potency and smaller amplitudes (e.g. the M₂R¹⁷⁵ biosensor) may potentially be more prone to activation, e.g. due to higher degrees of receptor/G-protein pre-coupling or spontaneous activity. Therefore, the Reviewer's remarks have prompted us to test experimentally whether the modification introduced in the receptor to obtain the biosensors may affect its basal activity. Accordingly, we have performed some preliminary experiments in which we looked at the effect of the inverse agonist N-methylscopolamine (NMS) on the M₂R^wt and biosensor-mediated activation of G_{oA} relative to ACh. And indeed, our preliminary data suggest that some biosensors have higher (175, 414), some lower (188, 419), and some (181) no spontaneous activity or R/G pre-coupling relative to wildtype receptors.

Upon further reflection this may actually not be too surprising and point to the structural basis for why these specific positions represent GPCR activation sensors whereas other positions do not. It is reasonable to assume that the positions that yield GPCR biosensors are directly reporting on allosteric communication networks between the ligand binding site and the G protein-coupling interface. Thus, our screen for GPCR biosensors might have unexpectedly proven itself as a straight-forward way of detecting key reporters in allosteric communications pathways within a GPCR.

Most importantly, however, and even though some biosensors appear to have different levels of constitutive activity, all of them, without any exception, readily distinguish between the potencies and efficacies of different agonists like wild-type receptors. This is specifically reflected by the following aspects:

- 1) The rank order of agonist potencies is exactly the same at all biosensors and at wild-type receptors: iperoxo is always the most potent agonist, whereas pilocarpine is always the least potent.
- 2) All biosensors, irrespective of their activity in the apo state, retain the ability to distinguish between full and partial agonists.

Based on the Reviewer's comment, we have toned down the wording in the manuscript which now reads on page 7: "All agonists stimulated G-protein activation at all seven biosensors in a concentration-dependent manner and **similarly** as at wild-type M₂ receptors (Extended Data Fig. 5). **We noted a trend that high agonist potency is negatively correlated with the dynamic range of the functional response (Extended Data Fig. 4i). This would be compatible with the notion that some biosensors have slightly different spontaneous activity than wild-type M₂ receptors.**

I will also note that in performing these analyses, I found discrepancies between the reported EC₅₀ values in the legend of Extended Data Figure 5 and the source data file. There are also discrepancies between Figure 3i and the source data file. I would advise a careful review of the source data file for any other discrepancies.

Authors' reply:

We are very grateful that the Reviewer has spotted these discrepancies, a fact that all authors deeply regret. The legend of Extended Data Figure 4 (former Extended Data Figure 5) has been corrected. Further, the current revised source data file has been reviewed independently by multiple co-authors and we are confident that now all data plotted in the Figure correspond correctly to the source data file.

Referee #3:

I appreciate the effort the authors have put into revising their manuscript. They now provide a detailed description of the underlying GCE technology and bioorthogonal labelling, even more detailed than expected. Overall, I believe this will make the methodology more broadly applicable to other researchers. In this regard, I also recommend that the plasmids be made more freely available to the community, since (if feasible) this would be of great benefit.

Authors' reply:

We are very grateful that the Reviewer appreciates the effort we have put into the revision and that we have exceeded the Reviewer's expectations. We will aim to distribute all plasmids *via* Addgene upon publication of this manuscript.

I have a few additional minor comments on this version:

1. Choice of ncAA: The authors write: "which is highly reactive with both tetrazine and methyl-tetrazine dye derivatives yet sufficiently stable over the duration of the experiment⁴¹." Please specify in that section what the duration of these experiments was, as this remains a critical question when working with TCO(2-en)-Lys.

Authors' reply:

We thank the Reviewer for spotting this point. In the revised version of the manuscript, we have added the duration of the experiment to this sentence which now reads “*which is highly reactive with both tetrazine and methyl-tetrazine dye derivatives yet sufficiently stable over the duration of the experiment (max. 2 min)*⁴¹.”

2. *Control without ncAA incorporation: The authors write: “We fully acknowledge the rationale for including a labelled control without ncAA incorporation. In the course of our thorough biosensor characterisation, we never observed any detectable labelling in cells lacking a properly incorporated ncAA (so that we could not image these samples). This corroborates the specificity of our labelling approach.” I still believe it would be useful to show the controls without ncAAs. While I understand that doing so may be more difficult at this stage, the authors should at least clarify in the respective section or figure legends that these control experiments were performed. This clarification would also be important for other researchers intending to apply this technology to other GPCR types.*

Authors’ reply:

In the revised version of the manuscript, we have included this experiment as Supplementary Figure 2 and added a sentence to the main text on page 5: “As a control, we performed labelling in the absence of TCO*K and did not observe any fluorescence (**Supplementary Fig. 2**).”

3. *“Up to” phrasing: On p. 5, the authors write: “we have previously demonstrated up to quantitative labelling...” I assume that “up to” is redundant here, or else the sentence needs to be reformulated.*

Authors’ reply:

We have rephrased this sentence which now reads “...we have previously demonstrated quantitative labeling...”.

4. *Plasmid ratios: In the Methods section, when describing different plasmid ratios (e.g., 1:1:1, 10:10:1, 5:5:1), does the total amount of DNA remain the same as defined at the beginning of that Methods section? Please clarify.*

Authors’ reply:

Correct. To clarify this aspect, we have added the total amount of DNA to the respective sections mentioned by the Reviewer.

5. *Plate format for DNA transfection: The authors mention transfection with 4 µg of total DNA, but it is not clear for which plate format this was done. For example, for determining the cell-surface expression of M2R biosensors with an ELISA, the cells were supplemented with TCO*K as described above and transfected with 10 µl Lipofectamine 2000, using a cDNA ratio of 1:1 (SP-HA-M2R^{XXX}TAG or SP-HA-M2R-wt, MbPyIRSAF/4xtRNAM15) of 4 µg total plasmid 24 h after seeding. Please specify the plate format in this context.*

Authors’ reply:

We have added the plate format for DNA transfection to the respective section which now reads “For determining the cell-surface expression of M₂R biosensors with an ELISA, the cells were supplemented with TCO*K as described above and transfected with 10 µl Lipofectamine 2000, using a cDNA ratio of 1:1 (SP-HA-M₂R^{XXX}TAG or SP-HA-

M₂R-wt, MbPyIRS^{AF}/4xtRNA^{M15}) of 4 µg total plasmid 24 h after seeding in a T25 cell culture flask“

6. *Ext. Data Fig. 1 legend: Are those epifluorescence images from living cells? I assume so, since the manuscript otherwise seems to show only live-cell images. If not, please specify in the figure legends and the Methods section.*

Authors' reply:

We are grateful for the Reviewer's comment. The respective images are confocal images and we have corrected the Figure legend accordingly.

7. *TAMRA-tetrazine source: In the Methods section, TAMRA-tetrazine is also mentioned. Please specify its source.*

Authors' reply:

The source for TAMRA-tetrazine has been added to the Methods section of the revised manuscript.